# RasGRP1 promotes the acute inflammatory response and restricts inflammation-associated cancer cell growth

Cong Wang [1,5], Xue Li [1,5], Binbin Xue [1,5], Changping Yu [1,5], Luoling Wang [1,5], Rilin Deng [1], Hui Liu [1], Zihao Chen [1], Yingdan Zhang [1], Suping Fan [2,3], Chaohui Zuo[4], Hungyu Sun[1], Haizhen Zhu [1] ✉, Jianli Wang [2,3] ✉ & Songqing Tang [1] ✉

An acute inflammatory response needs to be properly regulated to promote the elimination of pathogens and prevent the risk of tumorigenesis, but the relevant regulatory mechanism has not been fully elucidated. Here, we report that Ras guanine nucleotide-releasing protein 1 (RasGRP1) is a bifunctional regulator that promotes acute inflammation and inhibits inflammation-associated cancer. At the mRNA level, *Rasgrp1* activates the inflammatory response by functioning as a competing endogenous RNA to specifically promote IL-6 expression by sponging let-7a. In vivo overexpression of the *Rasgrp1* 3' untranslated region enhances lipopolysaccharide-induced systemic inflammation and dextran sulphate sodium-induced colitis in *Il6*[+/+] mice but not in *Il6*[-/-] mice. At the protein level, RasGRP1 overexpression significantly inhibits the tumour-promoting effect of IL-6 in hepatocellular carcinoma progenitor cell-like spheroids. Examination of the EGFR signalling pathway shows that RasGRP1 inhibits inflammation-associated cancer cell growth by disrupting the EGFR-SOS1-Ras-AKT signalling pathway. Tumour patients with high RasGRP1 expression have better clinical outcomes than those with low RasGRP1 expression. Considering that acute inflammation rarely leads to tumorigenesis, this study suggests that RasGRP1 may be an important bifunctional regulator of the acute inflammatory response and tumour growth.

IL-6 is an important inflammatory cytokine that is expressed during infection and cancer and was identified as a prominent target for clinical intervention[1–4]. Almost all stromal cells and immune cells produce IL-6. Cytokines, bacteria and viruses are major inducers of IL-6 production[5,6]. IL-6 expression is relatively low in resting innate immune cells but is rapidly elevated in innate immune cells activated by microbial components such as lipopolysaccharide (LPS)[7]. During the maintenance of IL-6 level under physiological conditions, the microRNA (miRNA) let-7, which is abundantly expressed in resting innate immune cells, represses IL-6 expression by binding to its 3'

[1]Institute of Pathogen Biology and Immunology, Department of Pharmacy, College of Biology, Hunan Provincial Key Laboratory of Medical Virology, Hunan University, Changsha 410082, China. [2]Institute of Immunology, and Bone Marrow Transplantation Center of the First Affiliated Hospital, Zhejiang University School of Medicine, Hangzhou 310058, China. [3]Institute of Hematology, Zhejiang University & Zhejiang Engineering Laboratory for Stem Cell and Immunotherapy, Hangzhou 310058, China. [4]Department of Gastroduodenal and Pancreatic Surgery, Translational Medicine Research Center of Liver Cancer, Hunan Cancer Hospital, Changsha 410013, China. [5]These authors contributed equally: Cong Wang, Xue Li, Binbin Xue, Changping Yu, Luoling Wang. ✉e-mail: zhuhaizhen69@yahoo.com; jlwang@zju.edu.cn; tangsq@hnu.edu.cn

untranslated region (UTR)[8–10]. However, let-7 expression is only slightly downregulated in activated innate immune cells[10]. Thus, the rapid inhibitory effect of let-7 on IL-6 in activated innate immune cells, especially during the acute inflammatory response, maybe a potential druggable target in inflammatory diseases.

miRNAs constitute a class of small noncoding RNAs that modulate gene expression by promoting mRNA degradation or inhibiting translation[11]. An mRNA can sometimes sponge many of its targeted miRNAs via its 3′ UTR. As miRNA sponges, RNA transcripts engage in crosstalk and regulate each other by competing for their shared miR-NAs, known as competing endogenous RNAs (ceRNAs)[12,13]. Based on competition for targets, ceRNAs exhibit positively correlated and temporal, spatial and disease-specific expression patterns[13]. Recent reports have indicated that mRNAs, long ncRNAs (lncRNAS), pseudo-genes and circular RNAs (cirRNAs) can act as ceRNAs to affect the progression of cancer or neurological diseases[14–19]. Interestingly, sponging of an miRNA is competitive after a threshold ceRNA level is reached[20]. For example, it was recently demonstrated that an miRNA target was derepressed immediately after a threshold of added ceRNA was exceeded[21]. In addition, the expression of inflammatory cytokines, such as IL-6, is substantially upregulated during the acute inflammatory response. However, the crosstalk between ceRNAs and the expression of inflammatory cytokines during the acute inflammatory response remains to be further investigated.

Ras guanine nucleotide-releasing protein 1 (RasGRP1), a member of the RasGRP family, is a guanine nucleotide exchange factor associated with Ras activation[22,23]. The catalytic region of RasGRP1 consists of a Ras exchange motif (REM) and a cell division cycle 25 (CDC25) domain, and the regulatory region of RasGRP1 contains two EF-hands (calcium-binding domains) and a C1 domain (a DAG-binding domain)[22,24]. RasGRP1 is highly expressed in T cells and can also be detected in B cells, mast cells, natural killer cells, neuronal cells and activated macrophages[25–30]. Previous studies have shown that RasGRP1 is critical for mediating Ras/Erk activation, thus promoting the positive and negative selection of T cells[31–33]. Recently, increasing evidence has shown that RasGRP1 is involved in many human diseases, especially inflammatory diseases such as systemic lupus erythematosus, type 2 diabetes, and cancer[34–38]. Although the function of RasGRP1 has been extensively investigated in T cells, its role in innate immune cells, especially in macrophages, is still unclear. Elucidating the function of RasGRP1 in innate immunity will provide further insight into the pathophysiology and underlying mechanisms of inflammatory diseases.

Here, we show that RasGRP1 is an important regulator of acute inflammation that promotes the production of the proinflammatory cytokine IL-6 by competing with let-7a and decreases the probability of inflammation-associated cancer in a protein-independent and protein-dependent manner, which may explain why acute inflammation rarely leads to tumorigenesis.

## Results

### The expression of both RasGRP1 and IL-6 is regulated by let-7a
Previous studies have shown that let-7 can directly inhibit the expression of IL-6 by binding the *IL6* 3′ UTR[8,9]. We examined the expression profiles of let-7a, let-7b, let-7c, let-7d, let-7e, let-7f, let-7g, let-7i and let-7k in mouse peritoneal macrophages and found that let-7a showed the highest expression level in these cells (Fig. 1a). In addition, the results of RNA immunoprecipitation (RIP) experiments in which miRNA was used to pull down mRNA showed that let-7a binds *Il6* mRNA in a dose-dependent manner (Fig. 1b), suggesting that let-7a may be the major let-7 family member that regulates IL-6 expression in peritoneal macrophages. To confirm that let-7a inhibits IL-6 expression, we cotransfected *Il6* or *Tnf* expression plasmids and synthetic let-7a mimics and/or let-7a inhibitors into HEK293 cells. The results showed that the let-7a mimics inhibited the expression of IL-6 and TNF

by binding the *Il6* 3′ UTR and that this effect was reversed by the let-7a inhibitor (Fig. 1c and Supplementary Fig. 1a, b). To investigate whether let-7a is a critical mediator for inhibiting IL-6 expression in macro-phages, we overexpressed or silenced let-7a in macrophages by transfecting the cells with synthetic let-7a mimics or a let-7a inhibitor. Quantitative RT–PCR analysis showed that *Il6* mRNA levels exhibited negligibly changes in LPS-stimulated macrophages transfected with let-7a mimics or a let-7a inhibitor (Fig. 1d). After stimulation of the cells with LPS, IL-6 protein levels decreased in the let-7a mimic-transfected cells and increased in the let-7a inhibitor-transfected cells compared with the control cells (Fig. 1e). To explore the mechanism that let-7a regulates IL-6 production, luciferase reporter plasmids containing the wild-type *Il6* 3′ UTR or a mutant *Il6* 3′ UTR were constructed (Supplementary Fig. 1c). Luciferase reporter system analysis indicated that let-7a mimics disrupted *Il6* 3′ UTR reporter gene activity but did not disrupt the luciferase activity of the mutant *Il6* 3′ UTR reporter gene (Fig. 1f). These results suggested that IL-6 expression in macrophages was inhibited by let-7a via translational inhibition but not via mRNA degradation.

To identify the possible targets of let-7a that might compete with *Il6* during the innate immune response, the gene expression profiles of mouse peritoneal macrophages stimulated with LPS were further analysed. We focused on the expression of 1046 putative let-7 target genes predicted by the TargetScan algorithm (http://www.targetscan.org), and 23 putative let-7 target genes, including 10 genes with upregulated expression, showed an expression change of 5-fold or greater compared to LPS-unstimulated macrophages (Fig. 1g). To verify the accuracy of the gene expression profiles, the expression of 10 upregulated genes was further quantified by qPCR, and the results showed that among these 10 genes, *Rasgrp1* exhibited the most strongly upregulated expression (Fig. 1h), indicating that *Rasgrp1* might effectively sponge let-7a. To confirm that *Rasgrp1* was regulated by let-7a, we cotransfected Rasgrp1 expression plasmids and synthetic let-7a mimics and/or a let-7a inhibitor into HEK293 cells and found that the let-7a mimics inhibited the expression of RasGRP1 by binding the *Rasgrp1* 3′ UTR and that the let-7a inhibitor reversed the effect of the let-7a mimics (Fig. 1i, j and Supplementary Fig. 1d). Luciferase reporter analysis also showed that the let-7a mimics reduced *Rasgrp1* 3′ UTR reporter gene activity but did not reduce the luciferase activity of the mutant *Rasgrp1* 3′ UTR reporter gene (Fig. 1k and Supplementary Fig. 1e). In summary, these results indicated that both IL-6 and RasGRP1 were regulated by let-7a.

### *Rasgrp1* is coexpressed with *Il6*, and *RasGRP1* silencing selectively inhibits IL-6 protein expression during the acute inflammatory response
To explore the relationship of *Rasgrp1* and *Il6* during the acute inflammatory response, we treated mouse peritoneal macrophages with a range of doses of LPS and observed that the expression of *Rasgrp1* and *Il6* was significantly upregulated in a dose-dependent manner (Fig. 2a). A similar phenomenon was observed when peritoneal macrophages were stimulated by a range of doses of polyinosinic-polycytidylic acid (poly(I:C)) (Fig. 2b) or CpG oligodeoxynucleotide (ODN) (Fig. 2c). In addition, we stimulated mouse bone marrow-derived macrophages with a range of doses of LPS, poly (I:C) or CpG ODN. We found that the expression of *Rasgrp1* and *Il6* was significantly upregulated in a dose-dependent manner (Supplementary Fig. 2a–c). These results suggested that *Rasgrp1* may have been coexpressed with *Il6*.

To confirm the coexpression of *Rasgrp1* and *Il6*, we treated mouse peritoneal macrophages with 100 ng/ml LPS for different durations and found that the expression levels of both *Rasgrp1* and *Il6* were quickly and significantly upregulated in a time-dependent manner (Fig. 2d). Similar results were obtained when peritoneal macrophages were stimulated with 10 μg/ml poly (I:C) (Fig. 2e) or 5 μg/ml CpG ODN

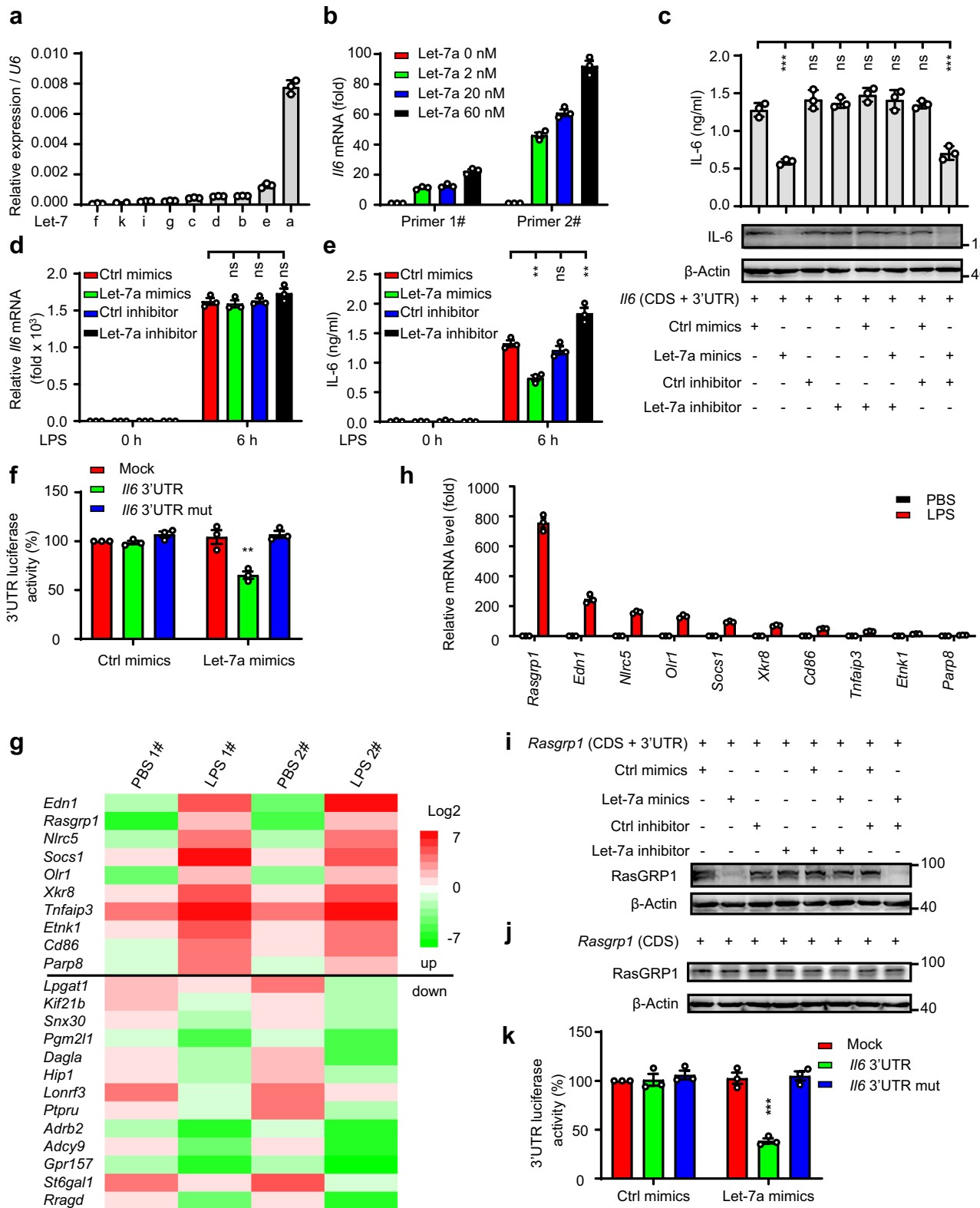

(Fig. 2f) for different durations. In addition, we stimulated mouse bone marrow-derived macrophages with 100 ng/ml LPS, 10 μg/ml poly (I:C) or 5 μg/ml CpG ODN for different durations. We observed that the *Rasgrp1* and *Il6* expression levels were quickly and significantly upregulated in a time-dependent manner (Supplementary Fig. 2d–f).

Next, we examined the coexpression of *Rasgrp1* and *Il6* in intestinal epithelial cells isolated from inflamed colons of C57B/6 mice. We found that *Rasgrp1* and *Il6* expression was significantly upregulated in

these cells in the days after the mice were fed 2.5% dextran sulfate disodium (DSS; Fig. 2g). We also examined the coexpression of *Rasgrp1* and *Il6* in cells isolated from the lung, kidney, colon, heart and liver of C57B/6 mice challenged with LPS. We observed that the expression levels of both *Rasgrp1* and *Il6* were significantly upregulated during LPS-induced systemic inflammation (Fig. 2h). Together, these data indicated that *Rasgrp1* is coexpressed with *Il6* in acute inflammation and inflammation-associated diseases.

**Fig. 1 | The expression of both RasGRP1 and IL-6 is regulated by let-7a. a** qPCR analysis of microRNA (miRNA) Let-7 isoform expression in peritoneal macrophages (means ± SEM, *n* = three independent experiments). **b** RNA immunoprecipitation (RIP) of the miRNA let-7a binding *Il6* mRNA. (means ± SEM, *n* = three independent experiments). **c** ELISA (**top**) of IL-6 in the supernatants of HEK293 cells transfected with the indicated molecules for 24 h (means ± SEM, *n* = three independent experiments, One-way ANOVA Tukey test, ***p = 0.0001, ***p = 0.0001 in sequence). Immunoblot analysis (**down**) of IL-6 in lysates of HEK293 cells of (**c**) (the image shown is a representative of three independent experiments). **d** qPCR analysis of *Il6* in peritoneal macrophages transfected with the indicated molecules (means ± SEM, *n* = three independent experiments, One-way ANOVA Tukey test). **e** ELISA of IL-6 in the supernatants of peritoneal macrophages treated as in (**d**) (means ± SEM, *n* = three independent experiments, One-way ANOVA Tukey test, **p = 0.002, **p = 0.003). **f** Luciferase reporter analysis of *Il6* 3' UTR reporter gene activity in HEK293 cells treated with the let-7a mimics or a matched control (means ± SEM, *n* = three independent experiments, One-way ANOVA LSD test, **p = 0.002). **g** Microarray analysis of mouse peritoneal macrophages treated with 100 ng/ml LPS. **h** qPCR analysis of predicted of the miRNA let-7-targeted genes in mouse peritoneal macrophages (means ± SEM, *n* = three independent experiments). **i** Immunoblot analysis of RasGRP1 and β-actin in lysates of HEK293 cells transfected with the indicated molecules for 24 h. **j** Immunoblot analysis of RasGRP1 and β-actin in lysates of HEK293 cells transfected as described in (**i**), but with *Rasgrp1* (CDS + 3' UTR) replacing *Rasgrp1* (CDS). **k** Luciferase reporter analysis of *Rasgrp1* 3' UTR reporter gene activity in HEK293 cells (means ± SEM, *n* = three independent experiments, One-way ANOVA LSD test, ***p = 0.0001). The data shown in **i** and **j** are from one representative experiment of three independent experiments. ns, not significant; *p < 0.05, ** p < 0.01, ***p < 0.001.

To ascertain the function of *Rasgrp1* in the acute inflammatory response, we silenced *Rasgrp1* expression in mouse peritoneal macrophages (Fig. 2i) and found that short interfering RNA (siRNA) knockdown of *Rasgrp1* did not affect the mRNA expression of cytokines or chemokines in peritoneal macrophages stimulated with LPS, poly (I:C) or CpG ODN (Fig. 2j–l and Supplementary Fig. 3a–o). Consistent with the mRNA expression of cytokines and chemokines, the activation of ERK, JNK and P38 MAPKs, as well as that of the IKKα/β-IκBα pathway, was not affected by *Rasgrp1* expression knockdown after LPS treatment (Supplementary Fig. 3p). Interestingly, IL-6 secretion was significantly decreased when we stimulated *Rasgrp1*-silenced peritoneal macrophages with LPS, poly (I:C) or CpG ODN (Fig. 2m–o). In contrast, no difference in TNF or IL-1β secretion was observed between the *Rasgrp1*-silenced macrophages and the control macrophages (Supplementary Fig. 3q–v). Taken together, these results showed that *Rasgrp1* selectively increased IL-6 protein levels during the acute inflammatory response.

We also examined the coexpression of *RasGRP1* and *IL6* in human peripheral blood monocyte-derived macrophages (MDMs). We found that *RasGRP1* and *IL6* expression levels were also significantly upregulated in MDMs stimulated with a range of doses of LPS (Supplementary Fig. 4a), poly (I:C) (Supplementary Fig. 4b) or CpG ODN (Supplementary Fig. 4c). Similar results were obtained when MDMs were stimulated with 100 ng/ml LPS (Supplementary Fig. 4d), 10 μg/ml poly (I:C) (Supplementary Fig. 4e) or 5 μg/ml CpG ODN (Supplementary Fig. 4f) for different durations.

Next, we silenced *RasGRP1* expression in human MDMs (Supplementary Fig. 4g) and found that siRNA knockdown of *RasGRP1* did not affect the mRNA expression of cytokines or chemokines in human MDMs stimulated with LPS, poly (I:C) or CpG ODN (Supplementary Fig. 4h–m). Consistent with the effects observed in mouse peritoneal macrophages, IL-6 secretion was significantly decreased when we silenced *RasGRP1* expression in human MDMs stimulated with LPS, poly (I:C) or CpG ODN (Supplementary Fig. 4n). However, TNF secretion was not affected (Supplementary Fig. 4o). Taken together, these results showed that *RasGRP1* also selectively promoted IL-6 protein levels in human MDMs during the acute inflammatory response.

### The *Rasgrp1* 3' UTR enhances IL-6 protein expression levels by sponging let-7a

To verify that *Rasgrp1* competes with *Il6* for let-7a binding, we cotransfected an *Il6* expression vector and let-7a mimics with the *Rasgrp1* coding sequence (CDS) and/or a 3' UTR expression vector into HEK293 cells and found that the *Rasgrp1* 3' UTR, but not the *Rasgrp1* CDS, promoted IL-6 expression by competing with let-7a for *Il6* binding (Fig. 3a). RIP results also showed that the *Rasgrp1* 3' UTR, but not the mutant *Rasgrp1* 3' UTR, competed with *Il6* for binding to let-7a (Fig. 3b). To identify the sequence in the *Rasgrp1* 3' UTR that binds to let-7a, three candidate let-7a-sponging RNAs that were each 30 bp in length were synthesised on the basis of the sequence proximal to the potential let-7 binding site in the *Rasgrp1* 3' UTR (Fig. 3c). Then, the effects of the three let-7a-sponging RNA candidates were examined by quantitative RT–PCR, and the results showed that the let-7a-sponging RNAs had no significant effect on *Il6* mRNA levels in LPS-stimulated macrophages in the presence or absence of the let-7a mimic (Fig. 3d and Supplementary Fig. 4p). ELISA showed that only the sponging RNA candidate containing a sequence complementary to the seed sequence (Supplementary Fig. 5a), which is important for the binding of an miRNA to a mRNA, increased IL-6 protein levels by competing with *Il6* in macrophages stimulated with LPS (Fig. 3e). Confocal microscopy showed that only the sponging RNA candidate containing a sequence complementary to the seed sequence bound let-7a (Fig. 3f and Supplementary Fig. 6). RIP results also showed that the sponging RNA candidate containing a sequence complementary to the seed sequence competed with *Il6* for let-7a binding and that this competition was dose-dependent (Fig. 3g). These data suggested that the sponging RNA candidate containing a sequence complementary to the seed sequence positively regulated the production of the IL-6 protein in macrophages during an acute inflammatory response.

To provide further evidence, two sgRNAs and three genotyping primers were designed to deplete the let-7a-binding site in the 3' UTR of *Rasgrp1* in RAW264.7 cells via a genome-editing technique (Supplementary Fig. 7a). Five candidate *Rasgrp1* 3' UTR mutant clones in RAW264.7 cells were confirmed by PCR screening (Supplementary Fig. 7b). DNA sequencing confirmed two positive clones among the five candidate mutant clones (Fig. 3h and Supplementary Fig. 8). In the two confirmed clones with mutations, we observed that the *Rasgrp1* and *Il6* mRNA levels were not affected by mutation of the *Rasgrp1* 3' UTR (Fig. 3i and Supplementary Fig. 9), but the IL-6 protein levels were significantly decreased after LPS, poly (I:C) or CpG ODN treatment (Fig. 3j). To exclude off-target effects, we transfected the sponging RNA candidate containing a sequence complementary to the seed sequence into the positive clones. We found that the sponging RNA candidate containing a sequence complementary to the seed sequence reversed the effect of *Rasgrp1* 3' UTR mutation on IL-6 protein levels after TLR3/4/9 agonist treatments (Fig. 3k). Together, these results suggested that the *Rasgrp1* 3' UTR sponges let-7a to increase IL-6 protein levels during an acute inflammatory response.

### The *Rasgrp1* mRNA 3' UTR aggravates LPS-induced systemic inflammation in *Il6*[+/+] mice

To investigate the role of the *Rasgrp1* 3' UTR in vivo, we synthesised cholesterol-conjugated let-7a-sponging RNA and let-7a mimics. After intraperitoneal injection of the cholesterol-conjugated let-7a-sponging RNA and let-7a mimics, we found that the sponging RNA candidate containing a sequence complementary to the seed sequence effectively bound to the let-7a mimics in vivo (Fig. 4a, b). The mice that had been intravenously injected with the cholesterol-conjugated let-7a mimics (Supplementary Fig. 10a) produced less IL-6 in response to LPS than did the control mice, and the sponging RNA candidate containing

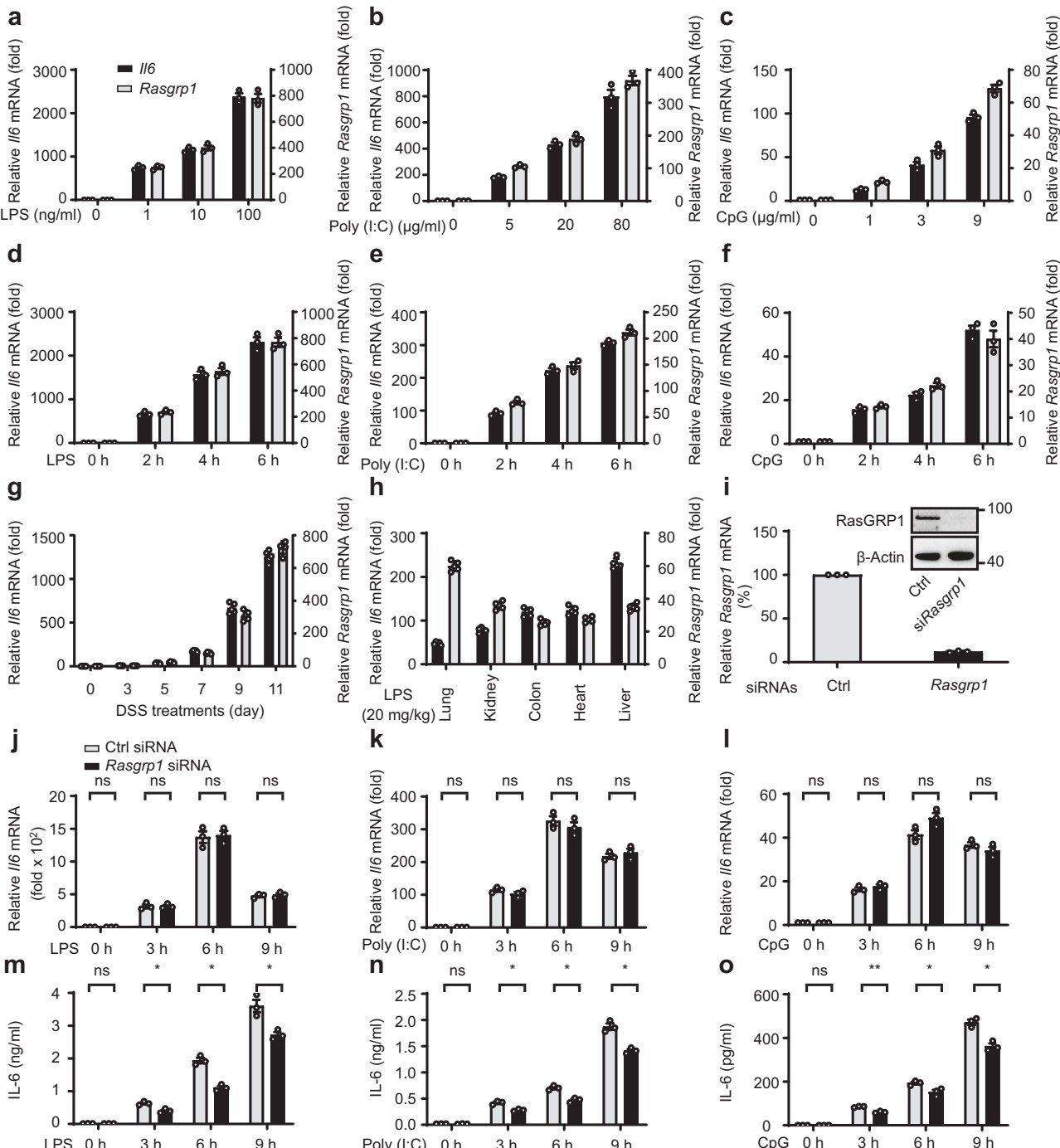

**Fig. 2 | *Rasgrp1* is coexpressed with *Il6*, and *Rasgrp1* silencing selectively inhibits IL-6 protein expression in an acute inflammatory response.**
**a–c** Quantitative PCR (Q-PCR) analysis of *Il6* and *Rasgrp1* mRNA expression in peritoneal macrophages treated with different doses of lipopolysaccharide (LPS; 0, 1, 10 or 100 ng/ml) (**a**), polyinosinic-polycytidylic acid (poly(I:C); 0, 5, 20 or 80 μg/ml) (**b**), or CpG oligodeoxynucleotide (0, 1, 3 or 9 μg/ml) (**c**) for 6 h. **d–f** qPCR analysis of *Il6* and *Rasgrp1* mRNA expression in peritoneal macrophages treated with 100 ng/ml LPS (**d**), 10 μg/ml poly (I:C) (**e**), or 5 μg/ml CpG ODN (**f**) for the indicated times (hours). **g** qPCR analysis of *Il6* and *Rasgrp1* mRNA expression in intestinal epithelial cells isolated from dextran sulphate sodium (DSS)-treated mice on the indicated days. **h** qPCR analysis of *Il6* and *Rasgrp1* mRNA expression in organs from mice intraperitoneally injected with LPS (15 mg per kg body weight) for 6 h. **i** qPCR analysis of *Rasgrp1* mRNA expression in peritoneal macrophages 48 h after transfection with *Rasgrp1* short interfering RNA (siRNA). RasGRP1 protein was examined by Western blotting (embedded image). **j–l** qPCR analysis of *Il6* mRNA expression in peritoneal macrophages transfected as described in (**a**) and treated 48 h later with 100 ng/ml LPS (**j**), 10 mg/ml poly (I:C) (**k**) or 5 mM of CpG ODN (**l**) for the indicated times (hours). **m–o** ELISA and quantification of IL-6 in the supernatant of macrophages treated as described in (**j–l**). The values plotted are the means ± SEM of three independent experiments in all figures, paired t test, two-sided, for **m**, *p = 0.0352, *p = 0.0112, *p = 0.0124 in sequence; for **n**, *p = 0.0445, *p = 0.0488, *p = 0.0394 in sequence, for **o**, *p = 0.0099, *p = 0.0258, *p = 0.049 in sequence. ns, not significant.

a sequence complementary to the seed sequence reversed the effects of the let-7a mimics (Fig. 4c). After lethal challenge with LPS, the *Il6*[+/+] mice but not the *Il6*[-/-] mice that had been intravenously injected with cholesterol-conjugated let-7a exhibited substantially prolonged survival, whereas the *Il6*[+/+] mice treated with let-7a-sponging RNA exhibited profoundly reduced survival (Fig. 4d, e). Furthermore, after intraperitoneal injection of the cholesterol-conjugated let-7a mimics into mice, we observed milder inflammatory infiltration in the lungs,

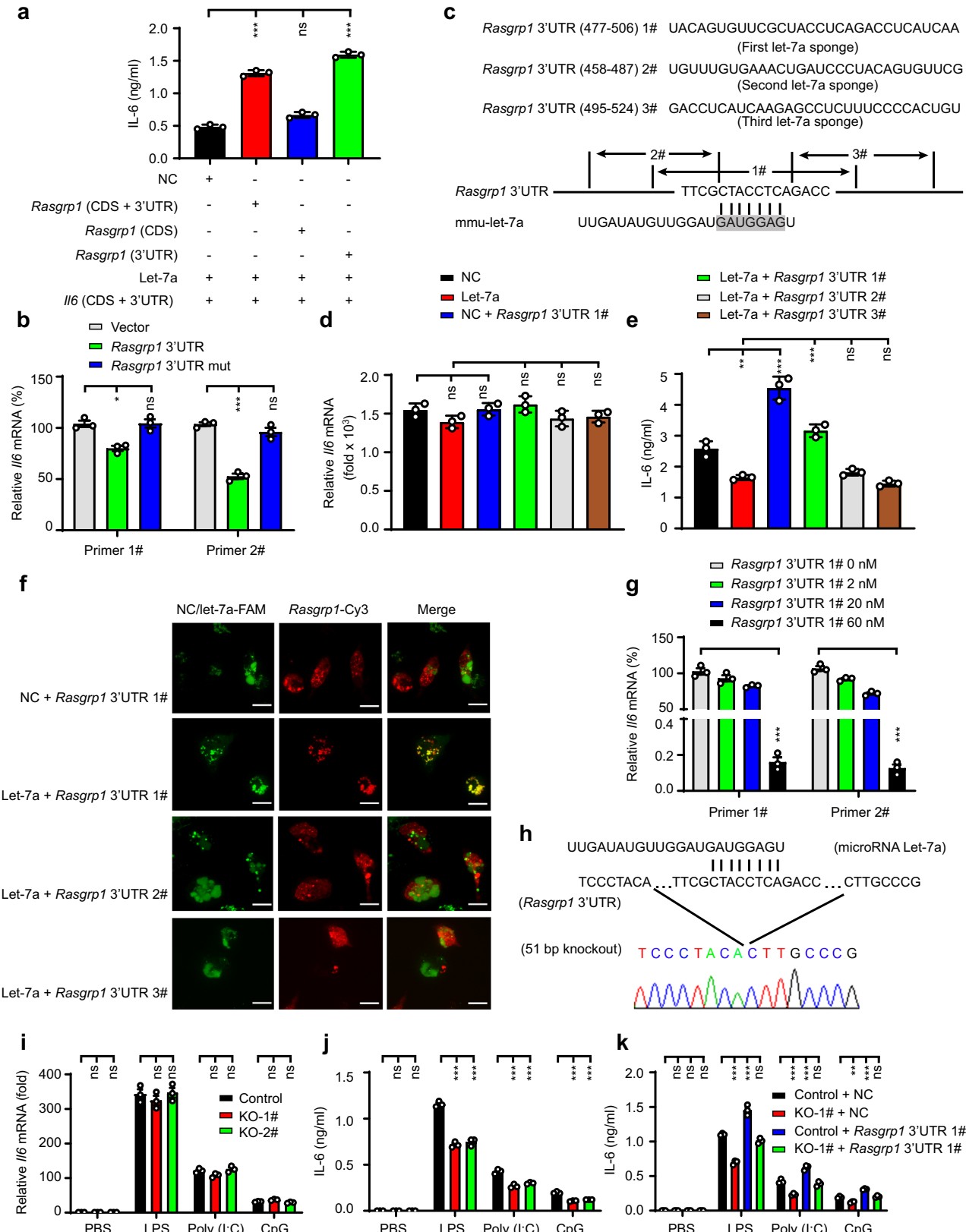

but intraperitoneal injection of the cholesterol-conjugated let-7a-sponging RNA caused more severe inflammatory infiltration in the lungs, and the sponging RNA candidate containing a sequence complementary to the seed sequence reversed the effects of the let-7a mimics (Fig. 4f). Immunohistochemistry (IHC) and immunofluorescence (IF) staining showed that the let-7a mimics inhibited IL-6

production (Fig. 4g–j and Supplementary Fig. 10) in the lungs of mice, but the sponging RNA candidate containing the sequence complementary to the seed sequence promoted IL-6 production. All these results suggested that the *Rasgrp1* 3'UTR promoted LPS-induced systemic inflammation by competing with *Il6* for let-7a binding, which led to enhanced IL-6 production. Considering that the uptake of the

**Fig. 3 | The 3′ UTR of *Rasgrp1* increases IL-6 protein levels by sponging let-7a.**
**a** ELISA of IL-6 in HEK293 cells transfected with plasmids expressing the CDS and/or 3′ UTR of *Rasgrp1* and *Il6* and transfected with let-7a mimics for 24 h (means ± SEM, One-way ANOVA Tukey test, ***$p$ = 0.0001, ***$p$ = 0.0001 in sequence, $n$ = three independent experiments). **b** RNA immunoprecipitation (RIP) analysis of the miRNA let-7a binding *Il6* mRNA (means ± SEM, One-way ANOVA LSD test, **$p$ = 0.002, ***$p$ = 0.0001 in sequence, $n$ = three independent experiments). **c** Schematic diagram showing the sequence of a sponge RNA. **d** qPCR of *Il6* in peritoneal macrophages transfected with indicated molecular and treated 24 h later with 100 ng/ml lipopolysaccharide (LPS) for 6 h (means ± SEM, $n$ = three independent experiments, One-way ANOVA Tukey test). **e** ELISA of IL-6 in the supernatants of peritoneal macrophages treated as described in (**d**) (means ± SEM, One-way ANOVA Tukey test, **$p$ = 0.002, ***$p$ = 0.0001, ***$p$ = 0.0001 in sequence, $n$ = 3 independent experiments). **f** Representative confocal microscopic image showing peritoneal macrophages cotransfected with indicated molecular (Scale bars represent 20 μm, the image shown is a representative of three independent experiments). **g** RIP analysis of the combination of let-7a and *Il6* mRNA. (means ± SEM, One-way ANOVA Tukey test, ***$p$ = 0.0001, ***$p$ = 0.0001 in sequence, $n$ = three independent experiments) **h** DNA sequencing of PCR products confirmed the targeted mutation in the RAW264.7 macrophage clone. **i** qPCR of *Il6* in the RAW264.7 macrophage clones KO-1# or KO-2# (means ± SEM, $n$ = three independent experiments, One-way ANOVA LSD test). **j** ELISA of IL-6 in the supernatants of the RAW264.7 macrophages treated as described in (**i**) (means ± SEM, One-way ANOVA LSD test, ***$p$ = 0.0001, ***$p$ = 0.0001, ***$p$ = 0.0001, ***$p$ = 0.0001, ***$p$ = 0.0001, ***$p$ = 0.0001 in sequence, $n$ = three independent experiments). **k** ELISA of IL-6 in the supernatants of the RAW264.7 macrophage clone KO-1# transfected and treated with the indicated molecular (means ± SEM, One-way ANOVA Tukey test, ***$p$ = 0.0001, ***$p$ = 0.0001, ***$p$ = 0.0001, ***$p$ = 0.0001, **$p$ = 0.001, ***$p$ = 0.0001 in sequence, $n$ = three independent experiments).

---

*Rasgrp1*−3′ UTR in vivo might not be very efficient, the observed in vivo effects might partly be due to the induction of an interferon response.

## The *Rasgrp1* 3′ UTR aggravates DSS-induced colitis in *Il6*[+/+] mice

Our data indicated that the *Rasgrp1* 3′ UTR promoted IL-6 protein production during TLR-induced inflammatory responses both in vitro and in vivo. To further investigate the pathophysiological significance of these observations, we analysed the role of *Rasgrp1* 3′ UTR in the development of DSS-induced colitis. After subjecting mice to DSS treatment, we found that the *Il6*[+/+] mice but not the *Il6*[-/-] mice that had been intravenously injected with cholesterol-conjugated let-7a, exhibited significantly less weight loss (Fig. 5a, b). However, the rate of body weight loss was significantly higher in the *Il6*[+/+] mice intravenously injected with cholesterol-conjugated let-7a-sponging RNA (Fig. 5a and Supplementary Fig. 11a). Moreover, the mice intravenously injected with cholesterol-conjugated let-7a mimic showed less severe rectal bleeding and colon shortening (Fig. 5c, d), lower serum IL-6 levels (Fig. 5e, g, h, j, k), less bowel wall destruction and greater bowel wall tissue proliferation, and less inflammatory cell infiltration than the control mice (Fig. 5f, i and Supplementary Fig. 11b). In contrast, the mice intravenously injected with cholesterol-conjugated let-7a-sponging RNA showed more severe rectal bleeding and greater colon shortening (Fig. 5c, d), higher serum IL-6 levels (Fig. 5e, g, j, k), greater bowel wall destruction and higher inflammatory cell infiltration than the control mice (Fig. 5f, i). When the cholesterol-conjugated let-7a mimic and let-7a-sponging RNA were simultaneously injected intravenously, the mice acquired a phenotype that was similar to that of the control mice (Fig. 5). Collectively, our data suggested that the *Rasgrp1* 3′ UTR aggravates the development of colitis by promoting IL-6 protein production.

## RasGRP1 inhibits the growth of inflammation-associated cancer cells

A previous study showed that IL-6 signalling contributes to the malignant progression of liver cancer progenitors[39]. Therefore, we induced HCC progenitor cell-like spheroid formation by growing Huh7 cells on low-attachment plates and found that IL-6 significantly promoted the malignant progression of liver cancer progenitors (Supplementary Fig. 12). An acute inflammatory response usually leads to high IL-6 production, but it rarely leads to tumorigenesis. How is the tumour-promoting effect of IL-6 counteracted during the acute inflammatory response? In our experiments, we observed that RasGRP1 expression was significantly upregulated, but the function of the RasGRP1 protein in the acute inflammatory response is largely unknown. To ascertain the role of RasGRP1, we used lentivirus to overexpress RasGRP1 in Huh 7 cells and found that RasGRP1 overexpression significantly inhibited the tumour-promoting effect of IL-6 (Fig. 6a, b and Supplementary Fig. 13). A recent study has reported that upregulation of RasGRP1 expression is associated with a better

prognosis in colorectal cancer patients because it limits proliferative EGFR-SOS1-Ras-ERK signals[40]. Therefore, we speculated that RasGRP1 may inhibit the development of liver cancer by inhibiting EGFR-SOS1-Ras-ERK signalling pathway activation to suppress the tumour-promoting effect of IL-6 during the acute inflammatory response. To test our hypothesis, we evaluated the expression levels of *RasGRP1*, *SOS1*, *SOS2*, *EGF* and *EGFR* in Huh7 and HepG2 cells and found that *EGF* and *EGFR* exhibited higher expression in Huh7 cells than in HepG2 cells (Fig. 6c). Then, we explored the role of RasGRP1 in Huh7 cells treated with 10 ng/ml EGF and found that RasGRP1 inhibited Huh7 cell proliferation (Fig. 6d–f), mainly by limiting EGFR-SOS1-Ras-AKT signalling (Fig. 6g, h). Next, we evaluated the specificity of the anti-RasGRP1 antibodies (Fig. 7a and Supplementary Fig. 14) and performed IHC staining for RasGRP1 in a human liver cancer tissue microarray. Representative images of the liver cancer tissue samples with high, moderate or weak (according to the IHC score in which weak≤1; 1<moderate≤2; and 2<high≤3) RasGRP1 expression are shown in Fig. 7b. Interestingly, only 22.2% of the liver cancer tissue samples exhibited high RasGRP1 expression, with a greater percentage of samples showing moderate and low RasGRP1 expression (40% and 37.8%, respectively) (Fig. 7c). However, 50% of the liver cancer tissue samples showed upregulated RasGRP1 expression compared to that in matched adjacent liver tissues (Fig. 7d). The RasGRP1 IHC score for the tumour samples with upregulated RasGRP1 expression was higher than that in tumour samples with downregulated RasGRP1 expression (Fig. 7e). The liver cancer patients with upregulated RasGRP1 expression exhibited significantly smaller tumour sizes (Fig. 7f), lower γ-glutamyl transferase levels (Fig. 7g) and longer total survival (Fig. 7h) than the liver cancer patients with downregulated RasGRP1 expression. Taken together, our results indicated that the RasGRP1 protein can inhibit liver cancer by limiting EGFR-SOS1-Ras-AKT signalling.

To investigate whether the RasGRP1 protein exerts an inhibitory effect on other cancers, we examined the effect of RasGRP1 gene expression on the survival of cancer patients with one of the multiple types of cancer. Specifically, we accessed the Kaplan–Meier Plotter database (http://kmplot.com/analysis/), and our analysis of the database samples indicated that high RasGRP1 expression was positively correlated (Fig. 8a–l) or not significantly correlated (Supplementary Fig. 15a–k) with higher overall survival for cancer patients with a specific type of cancer. Notably, high RasGRP1 expression was positively correlated with lower overall survival only for patients with kidney renal papillary cell carcinoma (Supplementary Fig. 15l). These results suggested that the RasGRP1 protein may inhibit the growth of multiple kinds of cancer cells.

## Discussion

In this study, we showed that RasGRP1 has an important function in regulating the acute inflammatory response and cancer. An acute inflammatory response is crucial for the efficient clearance of

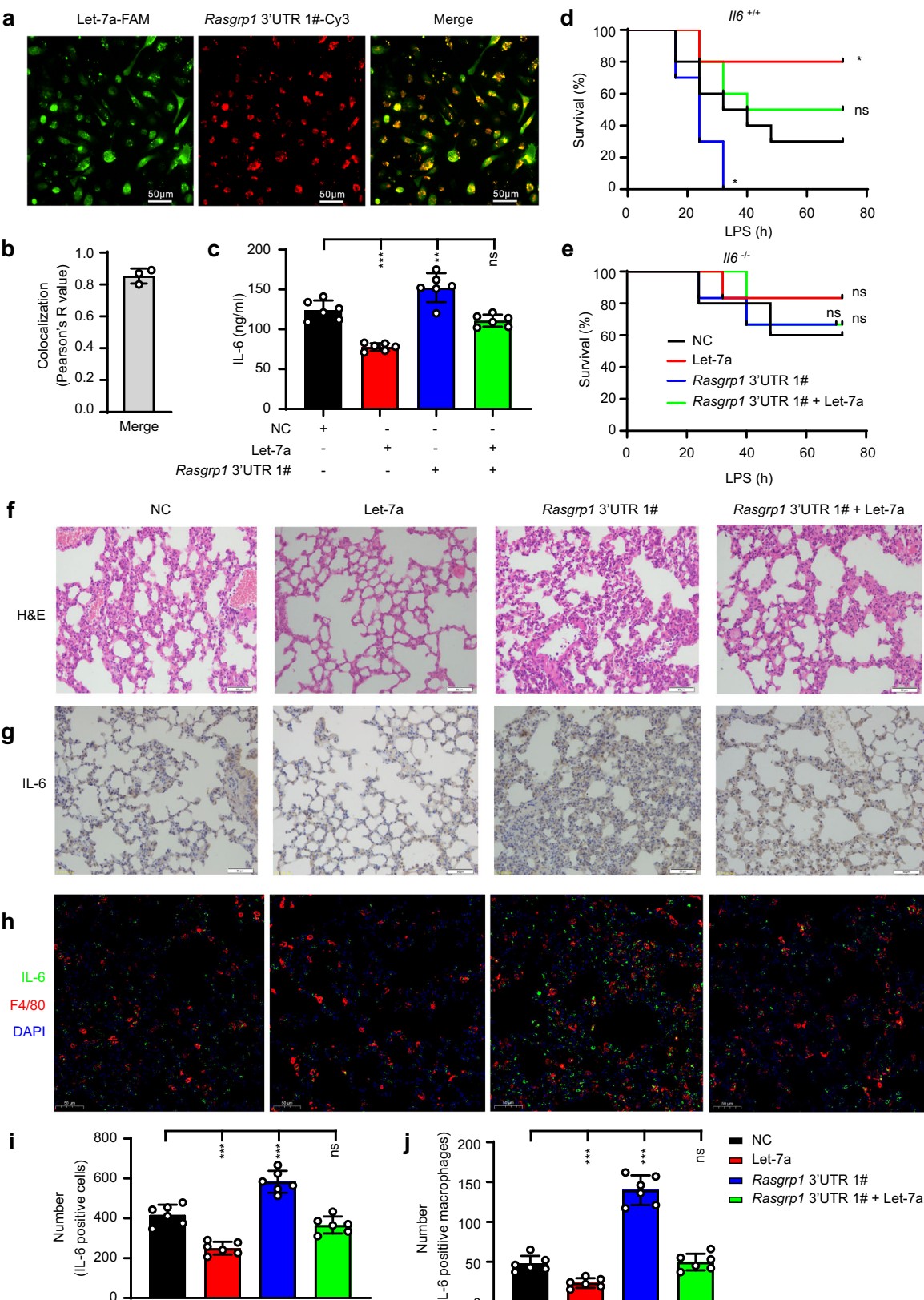

pathogens but needs to be strictly regulated to prevent deleterious side effects, such as the development of tumours and autoimmune diseases[9,41]. Our study demonstrated that RasGRP1 is a key regulator that promotes production of the proinflammtory cytokine IL-6 and decreases the probability of inflammation-associated cancer developing during acute inflammation (Fig. 8m).

Let-7 was one of the first miRNAs identified, and to uncover its biological role, extensive research has focused on let-7 target prediction[42]. Let-7a family members are highly conserved across animal species, particularly with respect to seed sequences, and have been reported to be important regulators of tumours and immunity, with specific roles in repressing the production of IL-6, a key

**Fig. 4 | The 3' UTR of *Rasgrp1* aggravates lipopolysaccharide (LPS)-induced systemic inflammation in *Il6*[+/+] mice. a** Representative confocal microscopic image showing peritoneal macrophages isolated from mice treated after intra-peritoneal injection of cholesterol-conjugated microRNA (miRNA) Let7a-FAM (green) and its sponge RNA *Rasgrp1* 3' UTR 1#-Cy3 (red). **b** Colocalization analysis (**a**) using the Coloc 2 plug-in program. (The values plotted are the means ± SEM of three independent experiments.) **c** ELISA of IL-6 level in serum obtained from *Il6*[+/+] mice after intravenous injection of cholesterol-conjugated miRNA let-7a and/or its sponge RNA *Rasgrp1* 3' UTR 1#, and injected intraperitoneally with LPS 24 h later; the analysis was performed 2 h after the LPS treatment (means ± SEM, *n* = 6, One-way ANOVA Tukey test, ***p* = 0.0001, **p* = 0.003, in sequence, *n* = 6 mice per group). **d, e** Survival of *Il6*[+/+] mice (**d**) and *Il6*[-/-] mice (**e**) treated as described in (**c**) (log-rank test, for **d**: **p* = 0.0293, **p* = 0.0333, in sequence, *n* = 10 mice per group;

for **e**: *n* = 6 mice per group). **f** Haematoxylin and eosin (H&E) staining of the lungs of *Il6*[+/+] mice treated as described in (**c**). **g** Immunohistochemistry (IHC) staining for IL-6 protein in the lungs of *Il6*[+/+] mice treated as described in (**f**). **h** Immunofluorescence costaining for macrophages and IL-6 in the lungs of *Il6*[+/+] mice treated as described in (**f**). **i** The number of IL-6-positive cells per 1000 cells shown in (**g**) was determined with Image-Pro Plus 6.0 software (means ± SEM, Mann–Whitney U, Two-side, ***p* = 0.0001, ***p* = 0.0001, in sequence, *n* = 6 field per group). **j** The number of IL-6-positive macrophages in (**h**) was determined with Image-Pro Plus 6.0 software (the means ± SEM, *n* = 6, Mann–Whitney U, two-side, ***p* = 0.0001, ***p* = 0.0001, in sequence, *n* = 6 field per group). (the image shown in **a**, **f**, **g** and **h** is a representative of three independent experiments; Scale bars in **a**, **f**, **g** and **h** represent 50 µm). ns, not significant; **p* < 0.05, ***p* < 0.01, *** *p* < 0.001.

proinflammatory cytokine in acute inflammatory responses[8]. We found that macrophages preferentially express let-7a to inhibit IL-6 production during the acute inflammatory response. A key question regarding the acute inflammatory response remains: How do macrophages rapidly release the inhibitory effect of let-7 on IL-6 during pathogenic infection. Our analysis of the gene expression profiles of mouse peritoneal macrophages stimulated with LPS showed that *Rasgrp1* mRNA may be the most important ceRNA for *Il6* binding because it sponges sufficient let-7a molecules to derepress the inhibitory effect of let-7 on IL-6 production. To determine whether the other known targets of let-7, such as *Edn1*, *Socs1*, *Olr1* and *Tnfaip3*[43–47], can synergistically regulate IL-6 expression with *Rasgrp1* during an acute inflammatory response, further investigation is needed. The reason why *Rasgrp1* mRNA did not affect *Il6* mRNA levels via let-7a was not discerned, which is a limitation of this study; further investigation is needed. Upon phosphorylation, Ago2 controls the activation of macrophages by reversing miRNA-mediated repression during the initial transient activation of macrophages[48]. Whether *Rasgrp1* and Ago2 exert a synergistic effect on IL-6 expression during macrophage activation also requires further investigation.

RasGRP1 is a Ras guanine nucleotide-releasing factor that activates Ras and its downstream ERK and AKT signalling pathways[49,50]. However, the function of RasGRP1 in tumorigenesis has been debated[51]. The EGFR signalling-driven ERK and AKT signalling cascades are critical for the development and tumorigenesis of HCC, lung cancer, breast cancer and other cancers[52–54]. A recent study reported that upregulation of RasGRP1 expression is associated with a better prognosis in colorectal cancer patients because it inhibits proliferative EGFR-SOS1-Ras-ERK signalling[40]. We found that overexpression of RasGRP1 suppressed liver cancer cell growth by inhibiting proliferative EGFR-SOS1-Ras-AKT signalling. However, further investigation may be required to understand whether RasGRP1-mediated EGFR-SOS1-Ras-ERK or EGFR-SOS1-Ras-AKT signalling inhibition is a general mechanism that affects other inflammation-associated cancers.

In summary, our study demonstrated that RasGRP1 is an essential regulator that promotes the inflammatory response by increasing IL-6 production and suppressing liver cancer cell growth via inhibition of EGFR-SOS1-Ras-AKT signalling pathway activation. Under physiological conditions, especially during an acute inflammatory response, RasGRP1 may be important in regulating the inflammatory response and decreasing the probability that inflammation-associated cancer will develop.

## Methods
### Ethics statement
All animal experiments were conducted in accordance with the principles of the Declaration of Helsinki and were approved by the Animal Care and Use Committee of the School of Medicine, Zhejiang University. Human tissue specimens were obtained with informed consent and approved by the institutional review boards of Hunan Cancer

Hospital, and the experiments were conducted in accordance with the principles of the Declaration of Helsinki.

### Mice, cells and reagents
Wild-type C57BL/6 mice (male, 6–8 weeks of age) were purchased from Joint Ventures Sipper BK Experimental Animals (Shanghai, China). *Il6*[-/-] mice (C57BL/6, 6–8 weeks of age) were purchased from the Jackson Laboratory. The mice were bred under pathogen-free conditions and housed in groups of three to five per cage on a 12 h light/dark cycle at 23 °C and 50% relative humidity. All animal experiments were conducted in accordance with the principles of the Declaration of Helsinki and were approved by the Medical Ethics Committee of Zhejiang University School of Medicine. HEK293 cells, HEK293T cells, Huh7 cells and RAW264.7 cells were obtained from the American Type Culture Collection (ATCC, Manassas, VA) and cultured according to ATCC-recommended protocols. Mouse peritoneal macrophages and human peripheral blood MDMs were prepared and cultured as described previously[30]. LPS (0111:B4) and poly (I:C) were purchased from Sigma (St. Louis, MO, USA). CpG ODN was obtained from Invitrogen (San Diego, California). Antibodies against RasGRP1 (ab37927, 1:1000 for WB, 1:100 for IHC) and Cy3 (ab52060, 1:100) and GAPDH (ab128915, 1:1000) were purchased from Abcam, Inc. (Cambridge, MA). Antibodies against IL-6 (12912, 1:100), F4/80 (30325, 1:200), Ki-67 (9129, 1:200), JNK1/2 (9252, 1:1000), phosphorylated (p)-JNK1/2 (Thr183/Tyr185) (4668, 1:1000), ERK1/2 (9101, 1:1000), p-ERK1/2 (Thr202/Tyr204) (4695, 1:1000), p38(9215, 1:1000), p-p38 (Thr180/Tyr182) (9212, 1:1000), AKT (9272, 1:1000), p-AKT (Ser473) (4060, 1:1000), IkBα (4814, 1:1000), p-IkBα (Ser32/36) (9246, 1:1000) and p-IKKα/β (2697, 1:1000) were purchased from Cell Signalling Technology (Beverly, MA). An antibody against β-actin (A1978, 1:2000) was purchased from Sigma–Aldrich.

### MiRNA mimics, sponges and inhibitors
Let-7a mimics (cholesterol-modified oligonucleotides labelled with FAM), let-7a sponges #1, #2, and #3 (cholesterol modified oligo-nucleotides labelled with Cy3) and let-7a inhibitors (dsRNA oligo-nucleotides) obtained from GenePharma (Shanghai, China) were used for the overexpression and inhibition of let-7a. Macrophages were transfected with miRNAs to establish a final concentration of 20 nM.

### Plasmid construction and transfection
The CDSs and 3' UTRs of *Rasgrp1* and *Il6* were obtained from mouse macrophage cDNA, and each was cloned into a pcDNA3.1 vector. The CDS of *Tnf* and the 3' UTR of *Il6* were obtained and cloned into a pcDNA3.1 vector. The wild-type *Il6* 3'UTR or a mutant *Il6* 3' UTR was cloned into the pMIR-Report vector (Ambion, Austin, TX). The wild-type *Rasgrp1* 3'UTR or a mutant *Rasgrp1* 3' UTR was cloned into the pcDNA3.1 vector or pMIR-Report vector (Ambion, Austin, TX). Each constructed plasmid was confirmed by sequencing. The

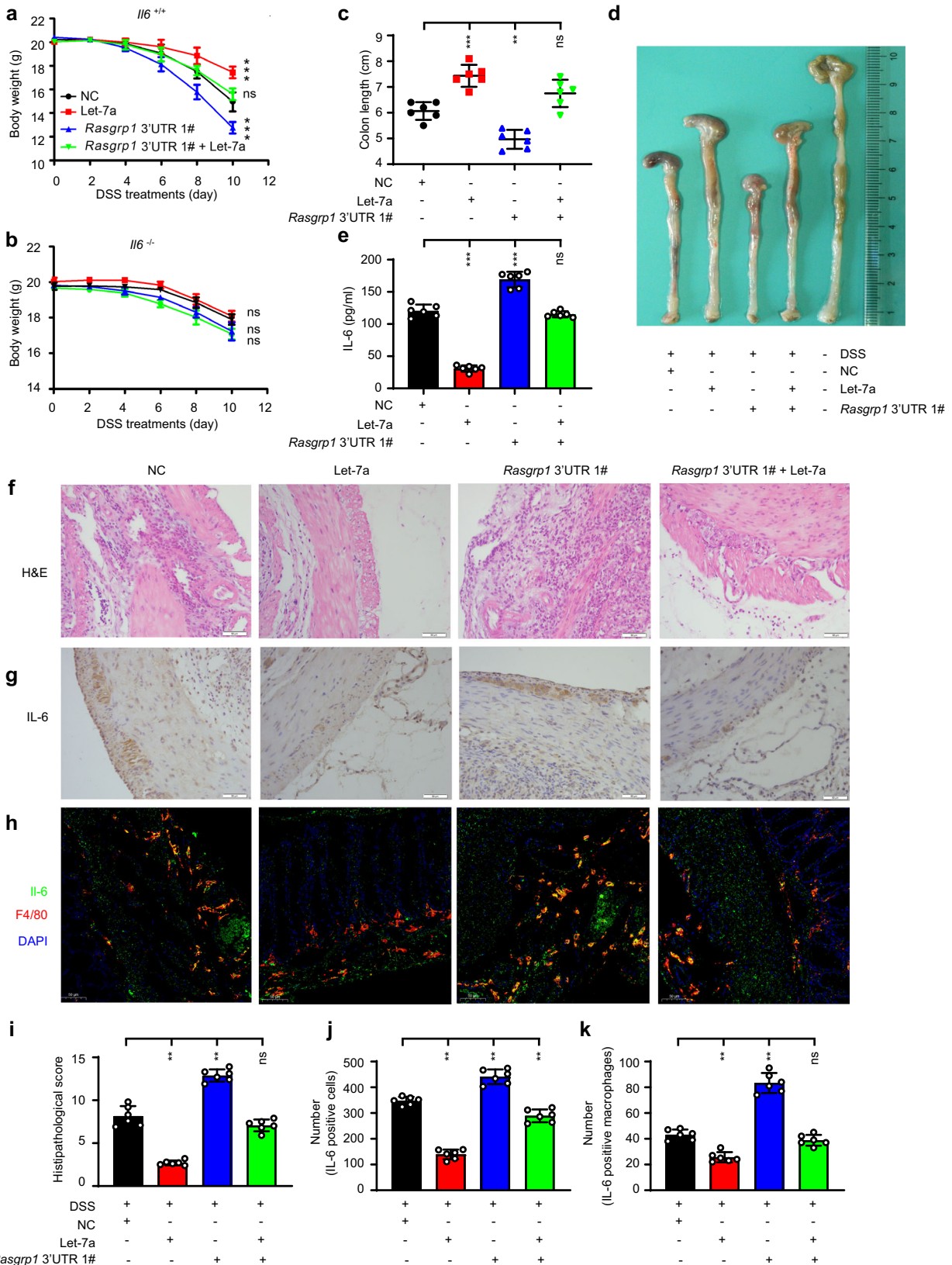

f H&E — NC, Let-7a, *Rasgrp1* 3'UTR 1#, *Rasgrp1* 3'UTR 1# + Let-7a

g IL-6

h Il-6 / F4/80 / DAPI

corresponding primers are listed in Supplementary information Table S1. JetPEI transfection reagents (Polyplus Transfection, Illkirch, France. Cat: 409-10) were used for the cotransfection of plasmids and miRNAs into HEK293T cells. Gene-specific primer pairs for plasmid construction are listed in Supplementary Table 1.

**RNA interference**

siRNA duplexes[30] were synthesised for knocking down mouse *Rasgrp1* and human *RasGRP1* expression (Shanghai GenePharma Co., Shanghai, China). The siRNA duplexes of 5'-GCUCCAUCUAUUCCAAGCUTT-3' (sense) and 5'-AGCUUGGAAUAGAUGGAGCTT-3' (antisense) were used

**Fig. 5 | The 3' UTR of *Rasgrp1* aggravates dextran sulphate sodium (DSS)-induced colitis in *Il6*[+/+] mice. a, b** Weight changes over time of *Il6*[+/+] mice (**a**) and *Il6*[−/−] mice (**b**) after intraperitoneal injection of the indicated molecular (means ± SEM, One-way ANOVA Tukey test, for **a**: ***$p = 0.0001$, ***$p = 0.0001$, in sequence, $n = 6$ per group; for **b**: $n = 6$ per group). **c** The length of the colons of the *Il6*[+/+] mice treated as described in (**a**) (means ± SEM, One-way ANOVA Tukey test, ***$p = 0.0001$, **$p = 0.001$, in sequence, n = 6 per group). **d** Gross appearance of representative colons collected from *Il6*[+/+] mice ($n = 6$ per group) treated as described in (**a**). **e** ELISA of IL-6 in serum of *Il6*[+/+] mice treated as described in (**a**) (means ± SEM, One-way ANOVA Tukey test, ***$p = 0.0001$, ***$p = 0.0001$, in sequence, $n = 6$ per group). **f** Haematoxylin and eosin (H&E) staining of the colons of the *Il6*[+/+] mice ($n = 6$ per group) treated as described in (**a**).

**g** Immunohistochemistry (IHC) staining for IL-6 protein in the colons of the *Il6*[+/+] mice ($n = 6$ per group). **h** Immunofluorescence costaining for macrophages and IL-6 in the colons of the *Il6*[+/+] mice ($n = 6$ per group) treated as described in (**a**). **i** The histopathological score in (**f**) (the means ± SEM, $n = 6$, Mann-Whitney U, two-side, **$p = 0.002$, **$p = 0.002$, in sequence, $n = 6$ field per group). **j** The number of IL-6 positive cells per 1000 cells shown in (**g**) was determined with Image-Pro Plus 6.0 software (the means ± SEM, $n = 6$, Mann−Whitney U, two-side, **$p = 0.002$, **$p = 0.002$, **$p = 0.002$, in sequence, $n = 6$ field per group). **k** The number of IL-6-positive macrophages shown in (**h**) (means ± SEM, $n = 6$, Mann−Whitney U, two-side, **$p = 0.0001$, **$p = 0.0001$, in sequence, $n = 6$ field per group). Scale bars represent 50 μm (bottom). All experiments were repeated three times. ns, not significant; *$p < 0.05$, **$p < 0.01$, ***$p < 0.001$.

for mouse *Rasgrp1* expression knockdown, whereas siRNA duplexes of 5'-GCGGGAUGAACUGUCACAATT-3' (sense) and 5'-UUGUGACAGUU-CAUCCCGCTT-3' (antisense), were used for human *RasGRP1* expression knockdown. The siRNA duplexes of 5'-UUCUCCGAACGUG UCACGUTT-3' (sense) and 5'-ACGUGACACGUUCGGAGAATT-3' (antisense) were synthesised as RNA interference controls (Ctrls). The siRNA duplexes were transfected into macrophages using INTERFERin according to the manufacturer's protocol (Polyplus-Transfection, Illkirch, France).

### Quantitative PCR (qRCR)
Total cellular RNA was extracted using TRIzol reagent, and complementary DNA was synthesised using a reverse transcription kit. Quantitative real-time RT–PCR was performed to analyse the levels of the indicated molecules by using a SYBR RT–PCR kit. Data obtained for each molecule were obtained by evaluating the threshold cycle (Ct) of the target molecule after normalisation against the Ct value of β-actin. Gene-specific primer pairs for qPCR are listed in Supplementary Table 2.

### Enzyme-linked immunosorbent assays (ELISAs) for cytokines
ELISA kits for human and mouse TNF (human, Cat: DTA00D; mouse, Cat: MTA00B), IL-1β (mouse, Cat: MLB00C) and IL-6 (human, Cat: D6050; mouse, Cat: SM6000B) were obtained from R&D Systems. The concentrations of TNF, IL-1β and IL-6 in the culture supernatants or serum were measured by ELISA. Briefly, 50 μl of diluted culture supernatant and 50 μl of assay diluent were added to each well and incubated for 2 h at room temperature, and the wells were washed three times with washing buffer. Then, 100 μl of anti-IL-6, anti-IL-1β or anti-TNF antibody was added to each well, and the plate was incubated for 2 h at room temperature and washed three times with washing buffer. After adding 100 μl per well of horseradish peroxidase-conjugated secondary antibody and incubating the plate in the dark for 1 h at room temperature, 50 μl of substrate per well was added to the extensively washed plate and incubated in the dark for 30 min at room temperature. Then, 50 μl of stopping buffer was added to each well to stop the reaction. Finally, the optical intensity of each sample was measured at 450 nm. The concentration of cytokines was calculated with a standard curve of standard samples.

### 3' UTR luciferase reporter
The wild-type and mutated *Il6* or *Rasgrp1* 3'UTR luciferase reporter construct was prepared by amplifying the mouse wild-type and mutated *Il6* or *Rasgrp1* mRNA 3'UTR and cloning into the pGL3-promoter vector (Promega). Eighty nanograms of luciferase reporter plasmid, 10 ng of pRL-TK-Renilla-luciferase plasmid and 50 nM let-7a mimics or matched control were cotransfected into HEK293 cells. After 24 h, luciferase activities were measured using the Dual-Luciferase Reporter Assay System (Promega). Data were normalized by dividing the firefly luciferase activity by the Renilla luciferase activity. Primer pairs for 3'UTR luciferase reporter vector construction are listed in Supplementary Table 3.

### Western blotting
Total cell lysates were prepared in cell lysis buffer (Cell Signaling Technology) containing phosphatase inhibitor cocktail (Sigma) as described previously[30]. Cell extracts were subjected to SDS–PAGE, and the proteins were transferred onto nitrocellulose membranes and subjected to immunoblot analysis. The primary antibodies and secondary antibodies were used at a 1:1000 dilution. Full-sized scans of western blots are provided in the Source Data file.

### Gene microarray analysis
RNA was extracted from macrophages or LPS-stimulated macrophages using an RNA extraction kit (Fastagen Biotech Co., Shanghai, China. Cat: 220011). Gene expression microarray analysis was performed using a Mouse OneArray Plus® Microarray (Hsinchu, Taiwan) according to the manufacturer's instructions. The changes in the let-7a target genes predicted by TargetScan (http://www.targetscan.org/vert_72) are presented in a heatmap on the basis of the microarray results.

### RIP analysis
Biotin-tagged let-7a and streptomycin tagged magnetic beads were used to pull down *Il6* mRNA for RIP analysis as previously described[55]. Briefly, HEK293 cells were cotransfected with *Il6* mRNA plasmids and/or *Rasgrp1* 3'UTR plasmids for 24 h. Total cellular RNA was extracted using TRIzol reagent, and the target mRNAs of let-7a were captured by biotin-tagged let-7a and streptomycin-tagged magnetic beads, washed, and extracted using TRIzol reagent again. The RIP RNAs were reverse transcribed and detected by qPCR assay. The primers used are listed in Supplementary Table 3. Gene-specific primer pairs for RIP analysis are listed in Supplementary Table 4.

### H&E staining
Mouse colons and lung tissues were embedded in paraffin, followed by slicing and dewaxing. The slices were then stained with haematoxylin (BOSTER, Wuhan, China) for 5 min and counterstained with eosin (BOSTER) for 1 min. Images were captured using an inverted microscope.

### Immunofluorescence
Mouse colons and lung tissues were embedded in paraffin, followed by slicing and deparaffinization. After antigen retrieval by boiling in antigen retrieval buffer (sodium citrate, pH 6.0), the tissue sections were blocked with goat serum for 1 h at room temperature. The slides were rinsed with PBS and then incubated with anti-F4/80 and anti-IL-6 or anti-Cy3 antibodies overnight at 4 °C, followed by incubation with fluorescent dye-conjugated secondary antibodies at room temperature for 1 h. Nuclei were counterstained with DAPI for 5 min. Images were captured using a confocal microscope.

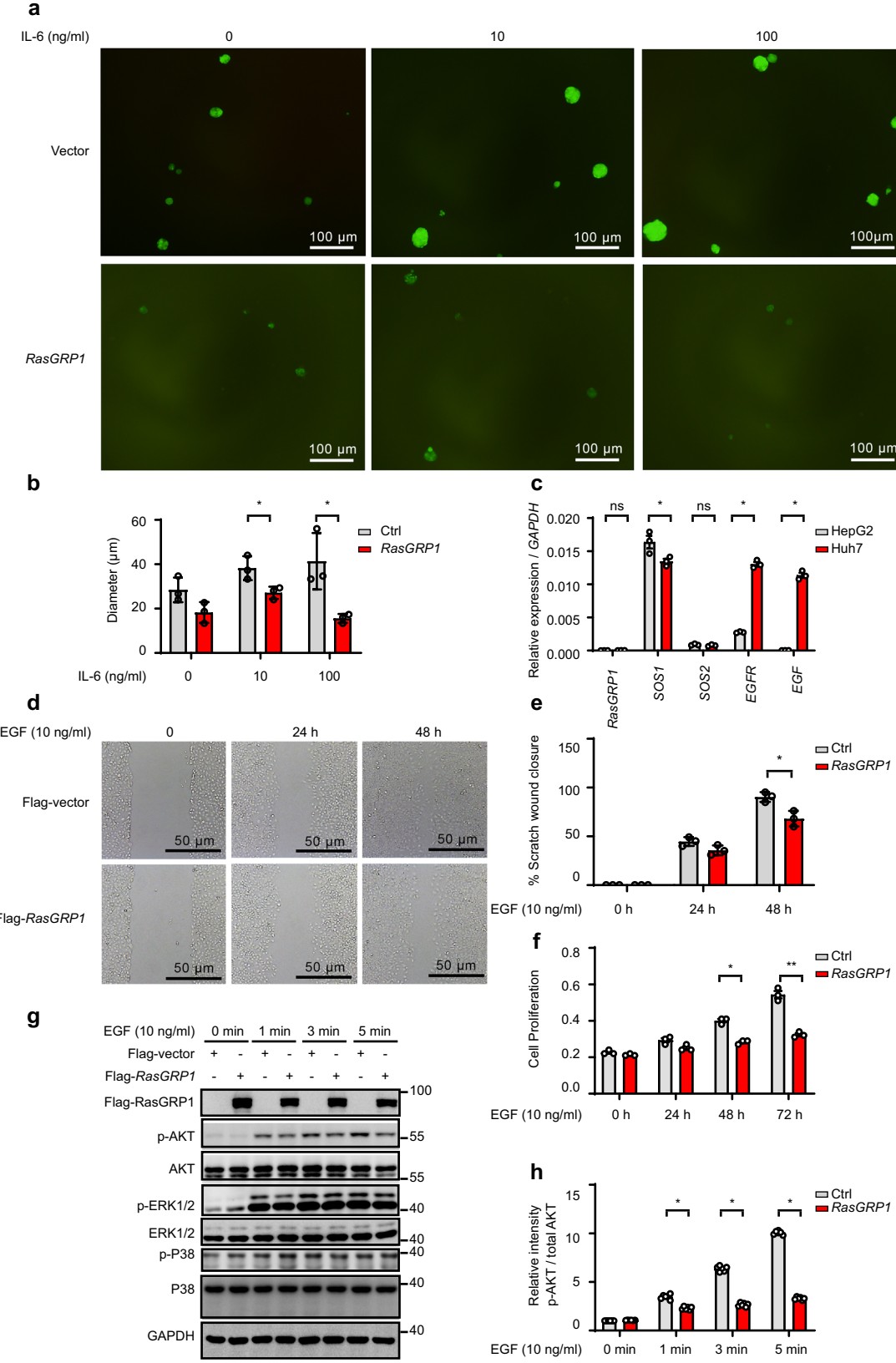

## Cell scratching test

Huh7 cells stably transfected with RasGRP1 or not were seeded in a six-well plate at 60% confluence and cultivated at 37 °C and 5% $CO_2$ for one day. Then, the cells were incubated with serum-free medium for another day. A scratch was made in the cell monolayer using a 10 µl pipette tip. The cells were then cultivated in serum-free medium for 24–48 h. Images of the scratch-healing process at 24 h and 48 h were obtained using an inverted microscope.

## Cultivation of HCC progenitor cell-like spheroids

Huh7 cells were cultivated on low-attachment plates in DMEM supplemented with EGF (20 ng/ml), FGF (20 ng/ml) and B27 (2%) as

**Fig. 6 | RasGRP1 affects the growth of liver cancer cells. a** Huh7 cells with or without overexpressing lentivirus-delivered RasGRPP1 were cultured with 20 ng/ml EGF, 20 ng/ml FGF, B27 (2%) and the indicated concentration of IL-6 on low-attachment plates. (A representative image of three independent experiments is shown, Scalebar, 100 μm). **b** Diameter of hepatocellular carcinoma (HCC) progenitor cell-like spheroids shown in (**a**). (means ± SEM, paired t test, two-sided, *p = 0.03, *p = 0.0462 in sequence, n = 3 HCC progenitor cell-like spheroids). **c** qPCR analysis of *RasGRP1, SOS1, SOS2, EGFR* and *EGF* expression in HepG2 and Huh7 cells (means ± SEM, unpaired t test, two-sided, *p = 0.0482, *p = 0.0001, *p = 0.0001 in sequence, n = three independent experiments). **d** Analysis of Huh7 cell migration by scratch wound assays. Images were acquired at the indicated times (hours) (A representative image of three independent experiments is shown). **e** Quantitative analysis of analysis shown in (**d**) (means ± SEM, paired t test, one-

side, *p = 0.0313, n = three independent experiments). **f** Cell Counting Kit-8 (CCK-8) assays were used to detect the role of RasGRP1 in Huh7 cells treated with 10 ng/ml EGF (means ± SEM, paired t test, two-side, *p = 0.0172, *p = 0.0033 in sequence, n = three independent experiments). **g** Immunoblot analysis of the indicated molecules in the lysates of Huh7 cells treated with 10 ng/ml EGF for the indicated times (minutes) (the image shown in **g** is a representative of five independent experiments). **h** The activation of AKT signalling as shown in (**e**) was quantified by determining the band intensity and was calculated as the ratio of the phosphorylated AKT level (Ser473) to total AKT level and normalised to that in unstimulated samples. (means ± SEM, Wilcoxon signed-rank test, two-sided, *p = 0.028, *p = 0.028, *p = 0.028 in sequence, n = five independent experiments). ns, not significant; *p < 0.05, **p < 0.01, ***p < 0.001.

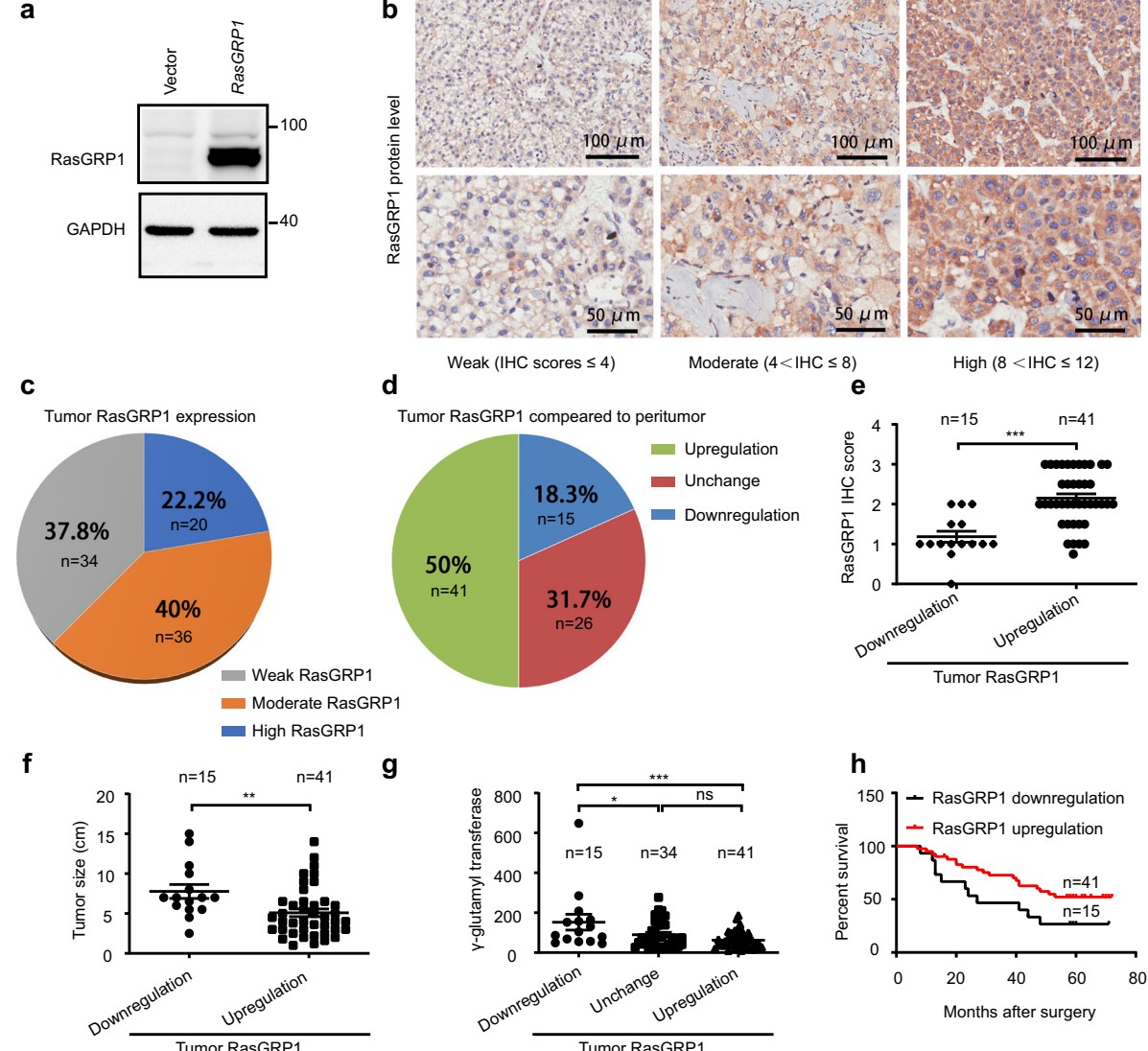

**Fig. 7 | RasGRP1 affects the prognosis of liver cancer patients. a** Western blot analysis with specific anti-RasGRP1 antibodies (the image shown in (**a**) is a representative of three independent experiments. **b** Representative images showing immunohistochemistry (IHC) staining (Primary antibody: anti-RasGRP1, 1:100 dilution, Abcam, ab37927; Second antibody: HRP-labelled goat anti-rabbit IgG, 1:1000 dilution, Sigma-Aldrich, AP132P) indicating high, moderate and weak staining intensity of RasGRP1 protein in a human liver cancer tissue microarray. **c** Pie chart showing the IHC-determined RasGRP1 expression levels in tumour tissues according to the findings shown in (**b**). **d** Demographics of the liver cancer patients exhibiting RasGRP1 expression changes in tumour tissues compared those

in paired adjacent liver tissues. **e** The expression levels of RasGRP1 in liver cancer tissues from patients with RasGRP1 expression that was upregulated (n = 41 patients) or downregulated (n = 15 patients) were compared (means ± SEM, Unpaired t test, Two-side, ***p = 0.0001). **f**, **g** The association between RasGRP1 and tumour size (**f**) or γ-glutamyl transferase (**g**) was analysed (means ± SEM, unpaired t test for **f**, Two-side, **p = 0.0075; and means ± SEM, One-way ANOVA LSD test for **g**, *p = 0.013, ***p = 0.0001 in sequence). **h** Kaplan–Meier analysis of RasGRP1 expression and overall survival of liver cancer patients (log-rank test, *p = 0.0437). ns, not significant; *p < 0.05, **p < 0.01, ***p < 0.001.

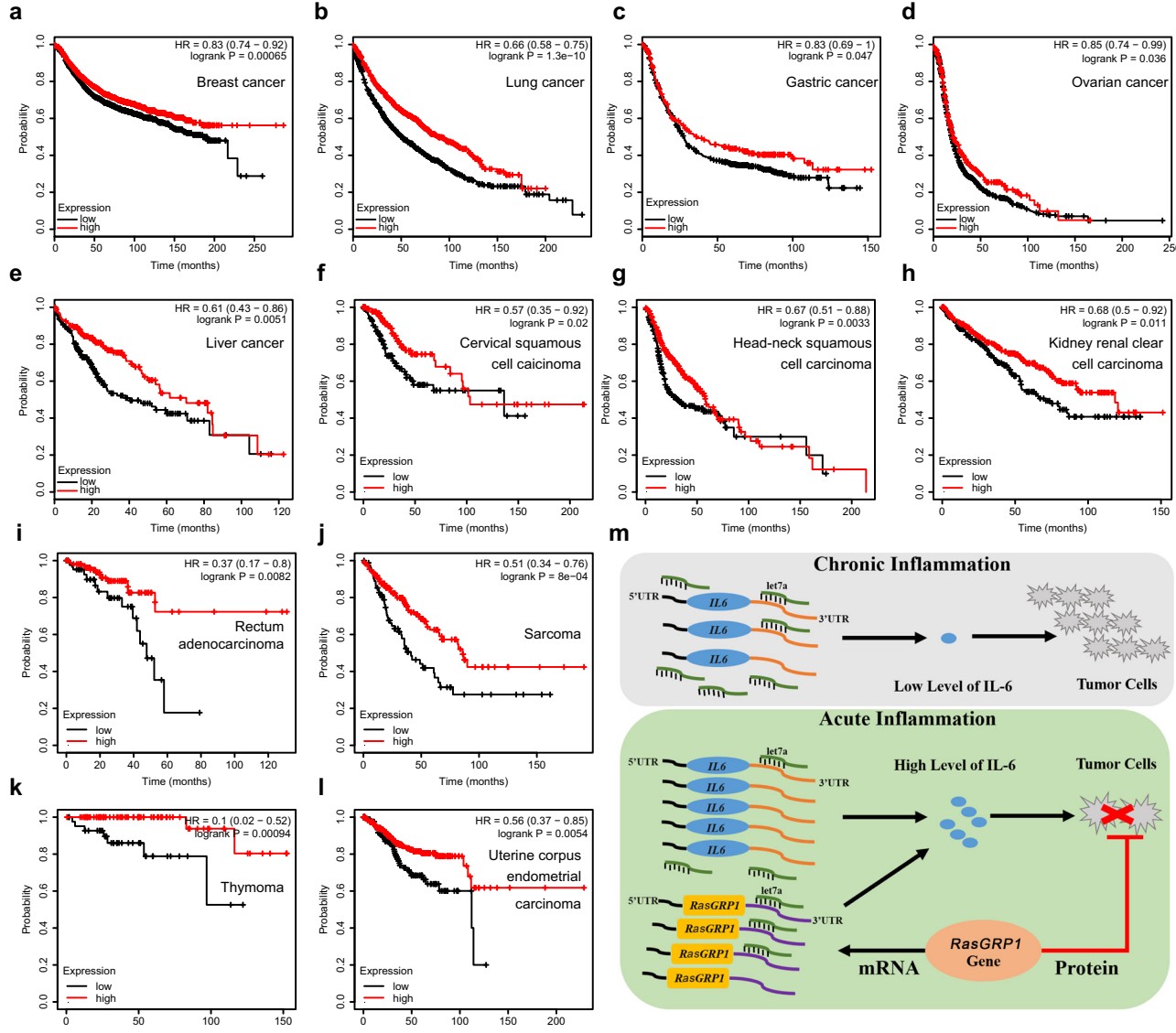

**Fig. 8 | RasGRP1 expression correlates with the overall survival of cancer patients. a–l** The survival curves determined through Kaplan–Meier plot profiling (http://kmplot.com/analysis/) of cancer patients stratified according to high and low RasGRP1 expression. **m** Schematic representation showing RasGRP1 effects in promoting proinflammatory cytokine IL-6 production and decreasing the probability of inflammation-associated cancer in an mRNA-dependent or protein-dependent manner.

previously described[56]. Briefly, Huh7 cells with or without over-expression of lentivirus-delivered RasGRPP1 were cultured with 20 ng/ml EGF, 20 ng/ml FGF, B27 (2%) and the indicated concentration of IL-6 on low-attachment plates and cultivated at 37 °C and 5% $CO_2$ for one week. Images of the HCC progenitor cell-like spheroids were captured under an inverted fluorescence microscope.

### Tumour samples
All fresh tumour tissues used for immunohistochemical staining were excised from patients hospitalised at Hunan Cancer Hospital. All experiments were undertaken in accordance with the approval of the institutional review boards of Hunan Cancer Hospital and Zhejiang University School of Medicine and were performed in accordance with the Declaration of Helsinki.

### Immunohistochemistry (IHC)
Colons and gastric tissues were subjected IHC staining as previously described[56]. Briefly, the tissues were embedded in paraffin, followed by slicing and deparaffination. After antigen retrieval by boiling in antigen retrieval buffer (sodium citrate, PH 6.0), the tissue sections were incubated in 2% hydrogen peroxide for 30 minutes and blocked with goat serum for 30 minutes at room temperature. The slides were incubated with an RasGRP1 antibody (Abcam, ab37927) overnight at 4 °C, followed by incubation with an HRP-labelled secondary antibody (Sigma–Aldrich, AP132P) for 50 min at room temperature and staining with DAB for 5 min. The slides were then counterstained with haematoxylin, dehydrated, and cover slipped using neutral resin.

### Liver cancer tissue microarray
A liver cancer tissue microarray containing 90 carcinoma tissues and 90 matched adjacent tissues was purchased from Shanghai Outdo Biotech Co. (Shanghai, China). IHC analysis was performed using an anti-RasGRP1-specific antibody (Abcam, Inc. Cambridge, MA) according to the manufacturer's instructions. Briefly, the tissue sections were blocked with goat serum and then incubated overnight with an anti-RasGRP1 antibody at 4 °C. The sections were stained with 3,3-diaminobenzidine and counterstained with

haematoxylin after incubation with a secondary antibody. PBS was used as the negative control. The staining intensity was evaluated independently by two pathologists without knowledge of the patients' clinical records, and the samples were categorized into negative, weakly positive, moderately positive and strongly positive groups according to the whole visual field of the tissue samples under a low-power microscope. The percentage of positively stained cells was defined as the average number of positive cells among 100 random cells in each of three visual fields with different staining intensities of each tissue sample under a high-power microscope. The IHC score (0–12) was calculated by multiplying the scores for the intensity of positive staining (negative = 0, weak = 1, moderate = 2, and strong = 3) and the percentage of positively stained cells (0–25% = 1, 26–50% = 2, 51–75% = 3, and 75–100% = 4). IHC scores ≤4 indicated weak RasGRP1 expression, IHC scores > 4 but ≤8 indicated moderate RasGRP1 expression, and IHC scores > 8<but ≤12 indicated high RasGRP1 expression.

### CRISPR–Cas9-mediated depletion of the *Rasgrp1* 3′ UTR
The *Rasgrp1* 3′ UTR was knocked out using a CRISPR–Cas9 system as described previously[30]. sgRNA sequences were designed at http://crispr.mit.edu as follows: sgRNA 1, 5′-TCTGAGGTAGCGAACACTGT-3′; sgRNA 2, 5′-GGAAGGAGGCGGGCAAGTGA-3′. These sequences were subcloned into a pLentiCRISPR V2 plasmid and cotransfected into RAW264.7 cells. After induction with 2 µg/ml puromycin for 7 days, the clones propagated from single cells were selected. Depletion of the *Rasgrp1* 3′UTR was identified by PCR and confirmed by DNA sequencing.

### Dextran sulfate sodium (DSS)-induced colitis and histology
To establish models of DSS-induced colitis in *Il6*⁻/⁻ mice and in wild-type mice, all mice (8 weeks of age) received water containing 2.5% DSS for 7 days followed by normal water. On the indicated days, the body weight of each mouse was measured. Ten days after the final DSS treatments, mouse sera were collected for use in the IL-6 ELISA, the colon length was measured, and the colons were subjected to hae-moxylin & eosin (H&E) staining and analysis to evaluate the severity of inflammation[30].

### Survival curves
The survival curves of the cancer patients were generated by Kaplan–Meier plot profiling (http://kmplot.com/analysis/).

### Statistical analysis
The data were analysed using SPSS software. The results are reported as the means ± SMEs or mean ± S.D.s of at least three independent experiments. The significance of differences between two groups was assessed by Student's *t* test, one-way ANOVA with the LSD test, one-way ANOVA with the Tukey test, the Mann-Whitney U-test and the Wil-coxon signed-rank test. Survival curves were generated using the Kaplan–Meier method and compared with the log-rank test. Statistical significance was defined as follow: *$p$ value < 0.05, **$p$ value < 0.01 or ***$p$ value < 0.001.

### Reporting summary
Further information on research design is available in the Nature Portfolio Reporting Summary linked to this article.

## Data availability
The microarray data generated in this study has been deposited in the NCBI Gene Expression Omnibus database[57] and is accessible through GEO with the primary accession number GSE165800. All other data generated in this study are provided in the article and the Supple-mentary Information. Source data are provided with this paper.

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

## Acknowledgements

We thank Prof. Long fei Mao (Hunan University) for the help of statistical test. This study was supported by grants from the National Natural Science Foundation of China (81971498, 81601383, 81730064, 81902069 and 31670877), the National Science and Technology Major Project (2017ZX10202201), the China Post-doctoral Science Foundation (2015M571888), the Fundamental Research Funds for the Central Universities and the Hunan Natural Science Foundation (2018JJ3091).

## Author contributions

S.T., J.W. and H.Z. designed the experiments and supervised the project; C.W., X.L., B.X., C.Y., L.W., R.D., H.L., Z.C., Y.Z., S.F. and C.Z. performed the experiments; S.T., H.S., J.W. and H.Z. analysed the results and wrote the manuscript.

## Competing interests

The authors declare no competing interests.
