## [Peer Review File · Nature Communications]

RasGRP1 promotes the acute inflammatory response and restricts inflammation-associated cancer cell growthREVIEWER COMMENTS

Reviewer #1 (Remarks to the Author):

The authors studied the effects of RASGRP1 3'UTR on IL-6 expression, mainly in macrophages and cell lines. They are showing the co-expression of RASGRP1 and IL-6 (mRNA) upon, for example, Toll-Like Receptor stimulation, and in different tissue cells from epithelial origin in mice. They show that Let7a transfection affects expression of IL-6, and that co-transfecting specific RASGRP1 3'UTR sequences may affect this. They hypothesize that a specific RASGRP1 3'UTR region is a sponge for let7a, and then move on to in vivo models of LPS- induced systemic inflammation, and DSS intestinal inflammation, showing that i.p. injection of RasGRP1 3'UTR promotes inflammation in IL-6 proficient mice. The concept is interesting. However, the data are currently not supporting the author's claims. Overall, I have multiple severe concerns about experimental controls, procedures, and analyses that should be addressed. Currently, the manuscript is showing many different models and pieces of data, that are not coherent. An improved coherence of the story-line would benefit the manuscript.

Major concerns:

1)

a) All figures: In the legends of figure 1 the authors describe to be showing 1 representative experiment of 3 separate experiments. For figures 1a,b,c,d, f, all experimental data should be included in the graphs. This goes for all figures that are summarizing graphs. Only for western blot and representative microscope images it is acceptable-with the notion that all raw data could be made available in the supplements.

b) Showing individual values is required to display variation in experiments.

c) statistical test used should be indicated in the legends.

d) It is not clear why in some cases SEM is shown instead of S.D.

2) In Figure 1b the authors perform an overexpression assay in HEK293 to measure IL-6 production upon co-expression of Let-7a mimics. The authors claim this provides proof that Let-7a directly binds to the IL-6 UTR. This is not proven here: it's a 24 hr assay, so many other processes could affect IL-6 production. This figure is missing some essential controls to determine whether this effect of Let-7a mimics is actually due to binding to the IL-6 UTR. I recommend including the following experiments:

a) transfect cells with IL-6 CDS+mutated UTR (mutate the site complementary to Let-7a) to show that this UTR is actually required for the Let-7a effect.

b) Use cells transfected with another cytokine/protein CDS+IL-6 UTR, and measure if that production is changed.

c) show the effect of Let-7a on IL-6 production of sham /un-transfected cells.

Now it is not clear whether HEK293 cells produce any IL-6 endogenously.

d) The above comments are similar for figure 1g, which shows the same assay for RASGRP1.

3) Fig 1,2,3:

a) The authors assess changes in secreted IL-6 upon LPS stimulation, and transfections.

The specificity of the Let-7a regulation for IL-6 production should be demonstrated by comparing the effect of Let-7a on other cytokines: TNFa, IL1b, which are produced upon macrophage LPS stimulation. It would be even stronger if they included a cytokine that does not show increased production by these macrophages upon LPS stimulation.

b) After transfection of primary cells, viability of the cells should be demonstrated to be equal and reasonable at the time of experimental readout. (let-7a, 3'UTR, and RASGRP1 siRNA)

4) Fig.1b,d,and Fig.2m,n,o: The authors use ELISA to determine protein expression for IL-6.

However, ELISA only shows secreted protein. The differences in protein expression, and thus an effect on translation which is claimed here, instead of cytokine secretion in the cells can be shown in a western blot, or for example by intracellular flow cytometry staining.

5) Figure S1p should include RasGRP1 expression in the same samples of this western blot to demonstrate effect on protein knockdown between conditions.

6) Figure 3c, and Fig 4a: Please provide quantification of co-localization of multiple cells.

7) Fig 4: Please indicate numbers of samples included in each figure.

-IL-6 stainings of FFPE: The authors should indicate the antibody clone or number in the methods, and, importantly, provide evidence that the antibody is specific in this protocol. In the current images the staining seems positive all over. It is impossible to judge from this whether these antibodies specifically bind to mouse IL-6, by eg including tissue of the IL-6 KO mice.

8) Fig 5. Though RasGRP1 is also important in epithelial, and T cells signaling and differentiation in DSS induced inflammation. However, The authors start the story with macrophages, and could here focus on the macrophages specifically to maintain a clear story on the function in macrophage driven IL-6 responses in acute inflammation.

9) Fig. 6. RasGRP1 antibodies for human FFPE staining are often aspecific. RasGRP1 antibody product number (methods) and specificity controls (figures) should be provided. A major conclusion is now based on a rather blunt quantification of signal by IHC, which has no proof of specificity. Moreover, scoring of intensity method is not described, and should be carefully done, not visually. This part of the story seems disconnected, and the data is not convincing.

Minor comments:

1) Figure 1e: show which values correspond to which color ('high', and 'low' are subjective)

2) Text:

a) The written text requires revision. It currently contains typo's and grammatical errors.

b) The abstract should describe how the protein level effect was studied (model) and the patient tumor types that are studied here.

c) In the introduction 'sponging' should be introduced to the reader very clearly.

Reviewer #2 (Remarks to the Author):

In the manuscript by Wang et al., the authors have tried to explain the let-7a sponging effect of Ras guanine nucleotide exchange factor RasGRP1 mRNA to explain the upregulation of let-7a target IL-6 that are getting controlled in activated macrophages by let-7a. They have also identified an miRNA controller independent role of RasGRP1 in controlling cancer cell growth. This is an interesting hypothesis but the data presented is not convincing enough to support the let-7a regulatory role of RasGRP1 in mammalian macrophages.

Major concern:

1. It is surprising to find that IL-6 mRNA don't respond to changed let-7a levels but rather the protein expression is only getting altered in presence of RasGRP1. Why? Are the mRNAs not getting engaged with polysomes? How they are escaping miRNA-mediated degradation?

2. It has been shown that in murine macrophage the initial activation phases the miRNAs undergoes a global derepression for initial time window of 1-6h (Mazumder et al 2013, EMBO rep). Therefore, the IL6 expression control in initial time window as shown in macrophage cells could have been nothing to do with RasGRP1-mediated sponging of let-7a rather due to global miRNA derepression.

3. Most of the RasGAP1 data on sponging has been done in HEK293 which is not a natural immune cells to express IL-6. The effect may be obtained with high expression of RasGAP1 mRNA/UTR but the concentration of the mimic or construct may not be in physiological levels in HEK293 cells compared to what we observed in macrophage cells where it is more relevant.

4. It would be good to test a point mutant of RasGap1 with mutation in predicted let-7a binding sites to confirm that the effect is mediated by let-7a and not by any other protein /RNA interacting with the deleted domain to in the deletion construct used to coregulate the IL-6 but independent of let-7a

activity change.

5. Let-7a expression is low in hepatic Huh7 and HepG2 cells (Pillai et al 2005, Science), therefore the regulation of RasGAP1 on oncogenic pathways is nothing to do let-7a sponging!

Reviewer #3 (Remarks to the Author):

In their manuscript "Ras guanine nucleotide exchange factor RasGRP1 promotes acute inflammatory response and restricts inflammation-contributed cancer cell growth" Cong Wang and colleagues investigate the molecular basis for TLR ligand-dependent rapid upregulation of IL-6 protein in macrophages. They describe a mechanism in which the miRNA Let7a represses IL6 mRNA translation. This inhibitory effect is released upon stimulation through transcriptional up-regulation of RasGRP1 mRNA that serves as a sponge RNA to scavenge Let-7a. They demonstrate that Let-7a binds to the 3' UTR of both, IL6 and Rasgrp1 mRNA and sole expression of Rasgrp1 3' UTR is sufficient to block Let7a-mediated repression of IL6.

The overall concept of the manuscript is intellectually appealing and most of the presented experiments are of high quality. The obtained in vivo data in Fig. 4 and 5. are very interesting. However, there are several major concerns that prevents the manuscript in its current form from being published in Nature communications.

General issues:

(1) All in vitro data that are presented are of one single experiment. Standard deviation and statistics are based on technical replicates and not on biological repetitions! Albeit the authors ensure in each Figure that the experiments are representative for three similar and independent experiments, this is not the scientific standard that should apply. In some experiments the biological difference is rather modest and might be due to biological variations. It is therefore absolutely mandatory to show the mean of at least three independent experiments in each Figure.

(2) Along the same line, for all in vitro data indication of the number of replicates and the statistical test that was applied is lacking. In the M&M section the authors indicate the usage of Student's t-test or Mann-Whitney U test. However, this is inappropriate when testing whether the mean of a population is different from a specific value (e.g. fold increase / 1). Use one sample t-test or Wilcoxon signed-rank test instead in this case.

(3) In their previous publication Tang et al. 2014, the authors claimed that RasGRP1 is barely, if at all expressed in murine peritoneal macrophages and human MDMs. Expression of RasGRP1 in peritoneal macrophages rather decreased upon LPS stimulation. Now they show the opposite with a 800 fold increase. How do these two findings fit together?

(5) The concept of RNA sponging is intellectually appealing. However, the specificity of RasGRP1 as "Let-7a sponge" is not evident to me.

How do the authors explain that Let-7a preferentially binds to RasGRP1? Do other potential Let-7a targets, e.g. Socs1 also serve as Let-7a sponge? An experiment using e.g. Socs1 3' UTR as presented in Fig. 3 a would be informative.

Along the same line, are protein levels of other Let-7a targets other than IL-6 also increased? Does the increase in RasGRP1 also lead to an increase in RasGRP1 translation? And if yes, does translation of RasGRP1 mRNA interfere with its "sponging function"? Or does binding of Let-7a prevent RasGRP1 translation?

(6) Albeit the data presented in Fig. 6 and 7 might be interesting they do not connect to the rest of the manuscript that focuses on regulation of IL-6 via RasGRP1. Therefore the title is also misleading as the authors do not demonstrate that RasGRP1 promotes inflammation in a murine tumour model or correlates with tumour-associated inflammation in human tumour samples.

If the authors would like to transfer their findings on IL-6 regulation into tumour cells, they should focus on HCC progenitor cells (He et al., 2013, Cell), given the fact that here Let-7a regulates an autocrine IL-6 loop. It would be interesting to see, if RasGRP1 contributes to the upregulation of IL-6 in these cells.

Alternatively, the authors might address the impact of RasGRP1 on EGF-dependent IL-6 expression/release in tumour-associated macrophages (TAMs) (see Lanaya et al. 2014, Nat Cell Biol) in HCC.

(6) The authors present expression data as fold increase. However, they do not further specify in relation to what. This is in particular intriguing for Fig. 1 a, which quantifies Let7 RNA species in untreated macrophages. Please specify accordingly in the Figure legend.

Specific issues:

Figure 1:

Fig. 1 g: In order to strengthen their concept, the authors should include a control in which RasGRP1 CDS w/o 3' UTR (or mutated UTR) is co-expressed with Let-7a mimic. As a consequence this control should not be influenced by the Let-7a mimic.

Figure 3:

Fig. 3 a: It would be interesting to analyse, if other Let-7a targets are also increasing. Furthermore, by titrating RasGRP1 (3' UTR) the authors should be able to demonstrate a dose-dependent effect. That would emphasize the "sponge-hypothesis".

Fig. 3 c: This is an interesting and convincing assay. However, punctae are difficult to see. Size of the images should be increased and scale bar should be integrated. The authors should also quantify co-localisation as it is difficult to draw conclusions from 2 single cells.

Fig. 3 f: Longer reads to cover also sgRNA sites should be included in the supplement.

Fig. 3 g-i: The authors should perform confocal images (incl. quantification) as in c) in order to prove that Let-7a binding to RasGRP1 RNA is lost. Furthermore, the authors should also perform qPCR for Rasgrp1 in order to demonstrate that RNA levels of Rasgrp1 are not altered due to the change in 3' UTR in their KO cells.

Figure 4+5:

The authors should demonstrate if injection of their conjugated RNA induces an IFN-response in the animals, as this might also alter translation.

Figure 4:

The main text is misleading. The authors talk about intravenous and intraperitoneal injections. If the authors used i.v. injections for b-f, they should demonstrate that also under these conditions, macrophages take up RasGRP1-3'UTR. In principle it should be possible to detect their conjugated RNA in macrophages present in the lung tissue e.g. by immunohistochemistry using anti-Cy3 antibodies.

Fig. 4a: Please increase image size and add scale bar. Co-localisation should be quantified.

Fig. 4b: What are the kinetics of IL-6 regulation after RasGRP1-3'UTR injection? Do the authors observe an altered acute-phase response in these animals?

Fig. 4f: Please show higher magnification. IL-6/F4/80 co-staining along with pSTAT3 staining would strengthen the authors' statement.

Figure 5:

The authors should present a disease index and histology score. As IL-6 has also a regenerative role in colitis, the authors should also investigate IEC proliferation (Ki67/BrdU) and survival (cl.Casp3).

How do the authors ensure that the conjugated RNA is still present in macrophages over that long period of time (see also above)?

Fig. 5f: Please show higher magnification. IL-6/F4/80 co-staining along with pSTAT3 staining would strengthen the authors' statement.

Figure 6:

As indicated above, these data do not fit to the rest of the manuscript. The effect of RasGRP1 on EGFR signalling seems to be independent of its "sponging effect". Furthermore, the role of EGFR in HCC is a bit more complicated. While EGFR in hepatocytes controls DNA damage repair (Geiger-Maor, 2015 Cancer Res; Guedj et al., 2020 Gut; Lanaya et al. 2014 Nat Cell Biol), it is rather EGFR in TAMs that promotes tumorigenesis through induction of IL-6 secretion. In order to link to the rest of their manuscript, the authors might want to correlate RasGRP1 expression in TAMs with IL-6 expression/secretion in TAMs.

Or focus on HcPCs as outlined above. To this end, the authors might want to grow Huh7 and HepG2 on low attachment plates to induce HcPC-like spheroids that do up-regulate IL-6.

Fig. 6c+d: How many independent experiments have been analysed? This should be indicated in the Figure legend. I assume the authors normalised to the respective unstimulated samples. This should be indicated.

Minor points:

Fig. 2: the authors should carefully check y-axis labeling.

Fig. S5: Labeling of the Figure (W, K, Q) does not correlate to the Figure legend (W, P, Q). Please correct.

Fig. 6b: Labeling should be improved and match the Figure legend. Please spell out CCK8 in the legend. This type of proliferation assay might not be familiar to every reader.

Fig. 6 j: The labeling and the Figure legend are misleading. What is really presented here?

Point-to-point responses to Reviewers (NCOMMS-21-07341)

Response to Reviewer #1

Many thanks for the insightful comments and suggestions. We have revised our manuscript as suggestions and performed new experiments which are required to address the concerns. We hope we have addressed your concerns in our revised manuscript. Thank you for your kind consideration. If any questions exist, inform us and we will discuss and perform required experiments.

Reviewer #1 The authors studied the effects of RASGRP1 3'UTR on IL-6 expression, mainly in macrophages and cell lines. They are showing the co-expression of RASGRP1 and IL-6 (mRNA) upon, for example, Toll-Like Receptor stimulation, and in different tissue cells from epithelial origin in mice.

They show that Let7a transfection affects expression of IL-6, and that co-transfecting specific RASGRP1 3'UTR sequences may affect this. They hypothesize that a specific RASGRP1 3'UTR region is a sponge for let7a, and then move on to in vivo models of LPS- induced systemic inflammation, and DSS intestinal inflammation, showing that i.p. injection of RasGRP1 3'UTR promotes inflammation in IL-6 proficient mice. The concept is interesting. However, the data are currently not supporting the author's claims. Overall, I have multiple severe concerns about experimental controls, procedures, and analyses that should be addressed. Currently, the manuscript is showing many different models and pieces of data, that are not coherent. An improved coherence of the storyline would benefit the manuscript.

Response: To address the concerns clearly, we performed Luciferase reporter system analysis and RNA-binding miRNA immunoprecipitation (RIP) analysis (new Fig.1b, f, k, and Fig.3b, 3g). Luciferase reporter system analysis results show that let-7a mimics could impair IL-6 or RasGRP1 3'UTR reporter gene activity but could not impair the luciferase activity of the mutant IL-6 or RasGRP1 3'UTR reporter gene (Fig.1f and k). RIP analysis results show that biotin tagged let-7a could pull down IL-6 mRNA in a let-7 dose dependent manner (Fig.1b). Overexpression of RasGRP1 3'UTR, but not the mutant RasGRP1 3'UTR could prevent the combination of let-7a and IL-6 (Fig.3b). The let-7a target sequence of RasGRP1 3'UTR could compete let-7a with IL-6 in a dose dependent manner (Fig. 3g). These results suggest that RASGRP1 3'UTR region is a sponge for let7a. We also performed the cultivation of HCC progenitor cells-like spheroids (Fig. 6a and Fig. S12). The results show that IL-6 significantly promote the growth of HCC progenitor cells-like spheroids in a dose dependent manner (Fig. S12). But RasGRP1 significantly inhibits the tumor-promotion effect of IL-6 in HCC progenitor cells-like spheroids (Fig. 6a). Taken together, our results indicated that RasGRP1 is an important regulator for acute inflammation to promote the production of the proinflammatory cytokine IL-6 and restrict IL-6-contributed cancer cell growth in a protein-independent or protein-dependent manner

Major concerns:

a) All figures: In the legends of figure 1 the authors describe to be showing 1 representative experiment of 3 separate experiments. For figures 1a,b,c,d, f, all

experimental data should be included in the graphs. This goes for all figures that are summarizing graphs. Only for western blot and representative microscope images it is acceptable-with the notion that all raw data could be made available in the supplements.

Response: We thank the reviewer for this suggestion and all experimental data were included in the graphs (**New Fig.1-7**).

b) Showing individual values is required to display variation in experiments.

c) statistical test used should be indicated in the legends.

Response: In the revised manuscript, the individual value of three independent experiments is showed in the summarized graphs and the statistical test is indicated in the legends.

d) It is not clear why in some cases SEM is shown instead of S.D.

Response: We apologize for this oversight. In the revised manuscript, we show means \pm SEM for three independent experiments and show means \pm S.D. for three repeat samples.

In Figure 1b the authors perform an overexpression assay in HEK293 to measure IL-6 production upon co-expression of Let-7a mimics. The authors claim this provides proof that Let-7a directly binds to the IL-6 UTR. This is not proven here: it's a 24 hr assay, so many other processes could affect IL-6 production. This figure is missing

some essential controls to determine whether this effect of Let-7a mimics is actually due to binding to the IL-6 UTR. I recommend including the following experiments:

a) transfect cells with IL-6 CDS+mutated UTR (mutate the site complementary to Let-7a) to show that this UTR is actually required for the Let-7a effect.

Response: We provide new data showing that IL-6 3'UTR, but not mutant IL-6 3'UTR, is actually required for the let-7a effect. **(New Fig. 1f)**

b) Use cells transfected with another cytokine/protein CDS+IL-6 UTR, and measure if that production is changed.

Response: TNF- α CDS + IL-6 3'UTR expression plasmids was constructed and we found that the expression of TNF- α is regulated by let-7a. **(New Fig. S1)**

c) show the effect of Let-7a on IL-6 production of sham /un-transfected cells.

Now it is not clear whether HEK293 cells produce any IL-6 endogenously.

d) The above comments are similar for figure 1g, which shows the same assay for RASGRP1.

Response: We analysis the effect of Let-7a on the production of IL-6 and RasGRP1 in un-transfected cells, but we found that HEK293 cells can't produce IL-6 and RasGRP1 endogenously (data was not shown).

We also provide new data showing that RasGRP1 3'UTR, but not mutant RasGRP1 3'UTR, is actually required for the let-7a effect. **(New Fig. 1j and k)**

3) Fig 1,2,3:

a) The authors assess changes in secreted IL-6 upon LPS stimulation, and transfections.

The specificity of the Let-7a regulation for IL-6 production should be demonstrated by comparing the effect of Let-7a on other cytokines: TNF α , IL1 β , which are produced upon macrophage LPS stimulation. It would be even stronger if they included a cytokine that does not show increased production by these macrophages upon LPS stimulation.

Response: We provide new data showing that the protein level of TNF- α and IL-1 β are not affected by let-7a in macrophages treated by LPS, Poly (I:C) or CpG. (New Fig. S1q-v)

b) After transfection of primary cells, viability of the cells should be demonstrated to be equal and reasonable at the time of experimental readout. (let-7a, 3'UTR, and RASGRP1 siRNA)

Response: We detected the viability of primary macrophages transfected let-7a, 3'UTR or RasGRP1 siRNA for 3 days and found that the viability of cells are very fine. (Data was not show in MS).

4) Fig.1b,d,and Fig.2m,n,o: The authors use ELISA to determine protein expression for IL-6. However, ELISA only shows secreted protein. The differences in protein expression, and thus an effect on translation which is claimed here, instead of

cytokine secretion in the cells can be shown in a western blot, or for example by intracellular flow cytometry staining.

Response: We appreciate the reviewer's suggestion. We detected the effect of Let-7a on IL-6 production by western blot. (New Fig. 1c)

5) Figure S1p should include RasGRP1 expression in the same samples of this western blot to demonstrate effect on protein knockdown between conditions.

Response: We analyzed RasGRP1 expression in figure S2p samples by western blot, but we can't detect it (data was not shown). Because RasGRP1 is barely expressed in unactivated murine peritoneal macrophages and the protein level of RasGRP1 is very low in murine peritoneal macrophages treated with LPS for 2 hours.

We demonstrated the effect of protein knockdown of RasGRP1 in macrophages treated with LPS for 24 hours by western blot. (New Fig. 2i)

6)Figure 3c, and Fig 4a: Please provide quantification of co-localization of multiple cells.

Response: Thanks for your helpful suggestion. We provided quantification of co-localization of multiple cells for Figure 3c, and Fig 4a. (New Fig. 4b and Fig. S6)

7) Fig 4: Please indicate numbers of samples included in each figure.

Response: We appreciate the reviewer's comment and provide the numbers of samples in the figure legends. (New Fig. 4)

-IL-6 stainings of FFPE: The authors should indicate the antibody clone or number in the methods, and, importantly, provide evidence that the antibody is specific in this protocol. In the current images the staining seems positive all over. It is impossible to judge from this whether these antibodies specifically bind to mouse IL-6, by eg including tissue of the IL-6 KO mice.

Response: We have indicated the antibody number in the methods (**page 18, line 391-395**). We provide new result to proof the specificity of IL-6 antibodies by using IL-6 KO mice. (**New Fig. S10b**)

8) Fig 5. Though RasGRP1 is also important in epithelial, and T cells signaling and differentiation in DSS induced inflammation. However, the authors start the story with macrophages, and could here focus on the macrophages specifically to maintain a clear story on the function in macrophage driven IL-6 responses in acute inflammation.

Response: Thanks for your constructive suggestion. We co-stained IL-6 and F4/80 to observe the production of IL-6 in macrophage. (**New Fig.5h**)

9) Fig. 6. RasGRP1 antibodies for human FFPE staining are often aspecific. RasGRP1 antibody product number (methods) and specificity controls (figures) should be provided. A major conclusion is now based on a rather blunt quantification of signal by IHC, which has no proof of specificity. Moreover, scoring of intensity method is

not described, and should be carefully done, not visually. This part of the story seems disconnected, and the data is not convincing.

Response: Many thanks for your insightful comments. We have provided RasGRP1 antibody product number (**ab37927, Lot: GR243514-1**) in the methods and provided specificity controls of RasGRP1 antibody. (**New Fig.7a**)

We described the method of scoring of RasGRP1 intensity in the methods. (**page 22, line 465-469**)

To make this part connect to the rest of the manuscript more tightly, we performed new experiments and found that RasGRP1 inhibits the tumour-promotion effect of IL-6 in HCC progenitor cells-like spheroids. (**New Fig.6a**)

Although RasGRP1 is a Ras guanine nucleotide releasing factor for the activation of oncogene Ras, but the function of RasGRP1 is not only promote proliferation and tumorigenesis. A study reported that RasGRP1 and RasGRP3-dependent Ca^{2+} -Erk signaling is pro-apoptotic in the negative selection of B cell, and disrupting Ca^{2+} -dependent activation of RasGRP1/3 and Erk is sufficient to inhibit antigen receptor-induced apoptosis (*Nat Immunol.* 2011; 12(5): 425–433). Another study reported that RasGRP1 opposes proliferative EGFR-SOS1-Ras signals and restricts intestinal epithelial cell growth (*Nat Cell Biol.* 2015;17(6):804-815). Our study found that RasGRP1 inhibits the tumour-promotion effect of IL-6 or EGF in Huh7 cells. It seems that the function of RasGRP1 is related to the stimulus and its strength. So, I think the data is reasonable.

Minor comments:

1) Figure 1e: show which values correspond to which color ('high', and 'low' are subjective)

Response: We provide the values of the color. (New Fig.1g)

2) Text:

a) The written text requires revision. It currently contains typo's and grammatical errors.

Response: We apologize for the typo's and grammatical errors, and we have checked carefully to corrected them.

b) The abstract should describe how the protein level effect was studied (model) and the patient tumor types that are studied here.

Response: We revised the abstract according to the suggestions. (page 2, line 32-35)

c) In the introduction 'sponging' should be introduced to the reader very clearly.

Response: Thanks for your nice suggestion. We revised the introduction according to your suggestion. (page 3, line 57-59)

Response to Reviewer #2

Many thanks for the insightful comments and suggestions. We have revised our manuscript as suggestions and performed new experiments which are required to address the concerns. We hope we have addressed your concerns in our revised manuscript. Thank you for your kind consideration. If any questions exist, inform us and we will discuss and perform required experiments.

Reviewer #2 In the manuscript by Wang et al., the authors have tried to explain the let-7a sponging effect of Ras guanine nucleotide exchange factor RasGRP1 mRNA to explain the upregulation of let-7a target IL-6 that are getting controlled in activated macrophages by let-7a. They have also identified a miRNA controller independent role of RasGRP1 in controlling cancer cell growth. This is an interesting hypothesis but the data presented in not convincing enough to support the let-7a regulatory role of RasGRP1 in mammalian macrophages.

Response: To address the concerns clearly, we performed Luciferase reporter system analysis and RNA-binding miRNA immunoprecipitation (RIP) analysis (new Fig.1b, f, k, and Fig.3b, 3g). Luciferase reporter system analysis results show that let-7a mimics could impair IL-6 or RasGRP1 3'UTR reporter gene activity but could not impair the luciferase activity of the mutant IL-6 or RasGRP1 3'UTR reporter gene (Fig.1f and k). RIP analysis results show that biotin tagged let-7a could pull down IL-6 mRNA in a let-7 dose dependent manner (Fig.1b). Overexpression of RasGRP1 3'UTR, but not the mutant RasGRP1 3'UTR could prevent the combination of let-7a and IL-6 (Fig.3b). The let-7a target sequence of RasGRP1 3'UTR could compete let-7a with

IL-6 in a dose dependent manner (Fig. 3g). We also performed the experiments of co-staining IL-6 and F4/80 in the lungs (Fig. 4h) or colons (Fig. 5h) to confirm the let-7a regulatory role of RasGRP1 in mammalian macrophages.

1. It is surprising to find that IL-6 mRNA don't respond to changed let-7a levels but rather the protein expression is only getting altered in presence of RasGRP1. Why ? Are the mRNAs not getting engaged with polysomes? How they are escaping miRNA-mediated degradation?

Response: We appreciate the reviewer's comment. Maybe most animal miRNAs form imperfect complementarity with mRNA to regulate protein synthesis, and plant miRNAs usually form perfect complementarity with mRNA to regulate mRNA degradation. In this study, IL-6 mRNA forms imperfect complementarity with Let-7a, which may help IL-6 mRNA escaping miRNA-mediated degradation. Further mechanistic experiments to explore the reason in more detail are beyond the scope of the current study.

2. It has been shown that in murine macrophage the initial activation phases the miRNAs undergoes a global derepression for initial time window of 1-6h (Mazumder et al 2013, EMBO rep). Therefore, the IL6 expression control in initial time window as shown in macrophage cells could have been nothing to do with RasGRP1-mediated sponging of let-7a rather due to global miRNA derepression.

Response: We have cited this paper in the discussion of our revised MS. We think the

global miRNA derepression and RasGRP1 3'UTR-mediated sponging effect on let-7a may work well together to regulate the expression of IL-6 to make sure the acute inflammatory response works smoothly. Because the mRNA level of IL-6 and the phosphorylation level of Ago2 are dynamic in the acute inflammatory response. When the phosphorylation level of Ago2 begin decline, the mRNA level of IL-6 still increasing. So, I think RasGRP1-mediated sponging effect on let-7a and Ago2-mediated global miRNA derepression may work well together to regulate the expression of IL-6.

3. Most of the RasGAP1 data on sponging has been done in HEK293 which is not a natural immune cells to express IL-6. The effect may be obtained with high expression of RasGAP1 mRNA/UTR but the concentration of the mimic or construct may not be in physiological levels in HEK293 cells compared to what we observed in macrophage cells where it is more relevant.

Response: Because HEK293 cell is not a natural cell to express IL-6 and RasGRP1, we can easily study the mechanism of RasGAP1 mRNA/UTR on the production of IL-6 by over-expression of IL-6 or RasGRP1 in HEK293 cell. Another reason is that HEK293 cell is a good cell for the study of miRNA and reporter gene. So, we firstly study the function of RasGRP1 on regulating the production of IL-6 in HEK293 cells and then confirm this function in macrophages or in vivo experiments.

4. It would be good to test a point mutant of RasGap1 with mutantation in predicted

let-7a binding sites to confirm that the effect is mediated by let-7a and not by any other protein /RNA interacting with the deleted domain to in the deletion construct used to coregulate the IL-6 but independent of let-7a activity change.

Response: We really appreciate the reviewer's good suggestion. We provide new experiments using a mutant RasGRP1 3'UTR in our revised MS. (New **Fig.1j and k**)

5. Let-7a expression is low in hepatic Huh7 and HepG2 cells (Pillai et al 2005, Science), therefore the regulation of RasGAP1 on oncogenic pathways is nothing to do let-7a sponging!

Response: We agree with the comment that Let-7a expression is very low in Huh7. To make the interpretation clearly, we performed a new experiment to study the function of RasGRP1 protein in inhibiting the tumour-promotion effect of IL-6 in HCC progenitor cells-like spheroids (New **Fig.6a**). Because IL-6 signaling contributes to the malignant progression of liver cancer progenitors. Acute inflammatory response usually produces a great quantity of IL-6 but rarely leads to tumorigenesis. We are interested in the question that how to prevent the tumor promoting effect of IL-6 during acute inflammatory response. Previous studies have reported that RasGRP1 and RasGRP3-dependent Ca^{2+} -Erk signaling is pro-apoptotic in the negative selection of B cell, and disrupting Ca^{2+} -dependent activation of RasGRP1/3 and Erk is sufficient to inhibit antigen receptor-induced apoptosis (*Nat Immunol.* 2011; 12(5): 425–433). Another study also reported that RasGRP1 opposes proliferative EGFR-SOS1-Ras signals and restricts intestinal epithelial cell growth (*Nat Cell*

Biol. 2015;17(6):804-815). So, we wanted to know whether RasGRP1 has a role in preventing tumorigenesis during acute inflammatory response. Our study find that RasGRP1 inhibits the tumour-promotion effect of IL-6 or EGF in Huh7 cells, which dependent on RasGRP1 protein but not dependent on RasGRP1 mRNA or Let-7a.

Response to Reviewer #3

Reviewer #3 In their manuscript “Ras guanine nucleotide exchange factor RasGRP1 promotes acute inflammatory response and restricts inflammation-contributed cancer cell growth” Cong Wang and colleagues investigate the molecular basis for TLR ligand-dependent rapid upregulation of IL-6 protein in macrophages. They describe a mechanism in which the miRNA *Let7a* represses IL6 mRNA translation. This inhibitory effect is released upon stimulation through transcriptional up-regulation of RasGRP1 mRNA that serves as a sponge RNA to scavenge *Let-7a*. They demonstrate that *Let-7a* binds to the 3' UTR of both, IL6 and *Rasgrp1* mRNA and sole expression of *Rasgrp1* 3' UTR is sufficient to block *Let7a*-mediated repression of IL6.

The overall concept of the manuscript is intellectually appealing and most of the presented experiments are of high quality. The obtained in vivo data in Fig. 4 and 5. are very interesting. However, there are several major concerns that prevents the manuscript in its current form from being published in Nature communications.

Response: Many thanks for the insightful comments and suggestions. We have revised our manuscript as suggestions and performed new experiments which are required to address the concerns. We hope we have addressed your concerns in our revised manuscript. Thank you for your kind consideration. If any questions exist, inform us and we will discuss and perform required experiments.

General issues:

(1) All in vitro data that are presented are of one single experiment. Standard deviation and statistics are based on technical replicates and not on biological

repetitions! Albeit the authors ensure in each Figure that the experiments are representative for three similar and independent experiments, this is not the scientific standard that should apply. In some experiments the biological difference is rather modest and might be due to biological variations. It is therefore absolutely mandatory to show the mean of at least three independent experiments in each Figure.

Response: Thanks for your good suggestion. I agree with you absolutely. In the revised manuscript, we show the mean of three independent experiments in the graphs.

(New Fig.1-7).

(2) Along the same line, for all in vitro data indication of the number of replicates and the statistical test that was applied is lacking. In the M&M section the authors indicate the usage of Student's t-test or Mann-Whitney U test. However, this is inappropriate when testing whether the mean of a population is different from a specific value (e.g. fold increase / 1). Use one sample t-test or Wilcoxon signed-rank test instead in this case.

Response: Thanks for your recommendations. We have indicated the number of replicates and indicated the statistical test in the figure legend for all in vitro data, and revised the M&M section about the usage of statistical test.

(3) In their previous publication Tang et al. 2014, the authors claimed that RasGRP1 is barely, if at all expressed in murine peritoneal macrophages and human MDMs. Expression of RasGRP1 in peritoneal macrophages rather decreased upon LPS

stimulation. Now they show the opposite with a 800 fold increase. How do these two findings fit together?

Response: We apologize for expressing not very clearly. In our previous publication in 2014, we found that RasGRP1 is barely expressed in murine peritoneal macrophages and human MDMs. The expression of RasGRP2-4 decreased in peritoneal macrophages treated with LPS, but the expression of RasGRP1 increased, which shown in the Supplementary Figure 1a. So, these two findings fit together.

(*Nat Commun.* 2014; 5:4657) Supplementary Figure 1. RasGRP1, RasGRP2 and RasGRP4 expression levels in mouse macrophages. Peritoneal macrophages derived from C57BL/6 wild type mice were treated with or without 100 ng/ml LPS (a, d, g),

10 µg/ml Poly (I:C) (b, e, h) or 5 µM CpG ODN (c, f, i) as indicated. The mRNA levels of RasGRP1 (a-c), RasGRP2 (d-f) and RasGRP4 (g-i) were examined by Q-PCR. Data are presented as mean±s.d. of triplicate samples and are representative of three independent experiments.

(4) The concept of RNA sponging is intellectually appealing. However, the specificity of RasGRP1 as “Let-7a sponge” is not evident to me.

How do the authors explain that Let-7a preferentially binds to RasGRP1? Do other potential Let-7a targets, e.g. Socs1 also serve as Let-7a sponge? An experiment using e.g. Socs1 3' UTR as presented in Fig. 3 a would be informative.

Response: Thanks for your comment. In my opinion, RasGRP1 as let-7a sponging mainly depends on the expression level of RasGRP1 mRNA (**New Fig. 3g**).

In another study, we analyzed the sponge effect of IL-10 3'UTR on let-7a, we found that IL-10 3'UTR also serves as let-7a sponge to regulate the expression of IL-6 (data was not shown). We know that IL-6 and IL-10 exhibit different temporal expression patterns. Maybe in some disease-specific expression patterns, IL-10 will serve as let-7a sponge to regulate the expression of IL-6.

Along the same line, are protein levels of other Let-7a targets other than IL-6 also increased? Does the increase in RasGRP1 also lead to an increase in RasGRP1 translation? And if yes, does translation of RasGRP1 mRNA interfere with its “sponging function”? Or does binding of Let-7a prevent RasGRP1 translation?

Response: Thanks for your interesting questions. we observed that both IL-6 and RasGRP1 increased at protein levels in acute inflammatory response. We think that the translation of RasGRP1 mRNA will interfere with its “sponging function” and binding of Let-7a will prevent RasGRP1 mRNA translation. When the mRNA of RasGRP1 increase, a part of RasGRP1 mRNA will as a sponge to bind Let-7a and another part of RasGRP1 mRNA will be translated to protein, which depends on the translation efficiency and sponging efficiency. So, the increase in RasGRP1 mRNA also lead to an increase in RasGRP1 translation.

(5) Albeit the data presented in Fig. 6 and 7 might be interesting they do not connect to the rest of the manuscript that focuses on regulation of IL-6 via RasGRP1. Therefore the title is also misleading as the authors do not demonstrate that RasGRP1 promotes inflammation in a murine tumour model or correlates with tumour-associated inflammation in human tumour samples.

If the authors would like to transfer their findings on IL-6 regulation into tumour cells, they should focus on HCC progenitor cells (He et al., 2013, Cell), given the fact that here Let-7a regulates an autocrine IL-6 loop. It would be interesting to see, if RasGRP1 contributes to the upregulation of IL-6 in these cells.

Alternatively, the authors might address the impact of RasGRP1 on EGF-dependent IL-6 expression/release in tumour-associated macrophages (TAMs) (see Lanaya et al. 2014, Nat Cell Biol) in HCC.

Response: We really appreciate the reviewer’s constructive comments. We focus on

HCC progenitor cells and performed a new experiment and found that RasGRP1 protein inhibits the tumor-promotion effect of IL-6 in Huh7 induced HCC progenitor cells-like spheroids (New **Fig.6a and Fig S12**).

(6) The authors present expression data as fold increase. However, they do not further specify in relation to what. This is in particular intriguing for Fig. 1 a, which quantifies Let7 RNA species in untreated macrophages. Please specify accordingly in the Figure legend.

Response: We appreciate the reviewer's suggestion. we have revised the Figure and Figure label. (**Fig.1a and Fig 6b**).

Specific issues:

Figure 1:

Fig. 1 g: In order to strengthen their concept, the authors should include a control in which RasGRP1 CDS w/o 3' UTR (or mutated UTR) is co-expressed with Let-7a mimic. As a consequence, this control should not be influenced by the Let-7a mimic.

Response: Thanks for your good suggestion. We have performed this control experiment. (**New Fig. 1j**).

Figure 3:

Fig. 3 a: It would be interesting to analyse, if other Let-7a targets are also increasing. Furthermore, by titrating RasGRP1 (3' UTR) the authors should be able to

demonstrate a dose-dependent effect. That would emphasize the “sponge-hypothesis”.

Response: We performed the dose-dependent experiment in **new fig. 3g**.

Fig. 3 c: This is an interesting and convincing assay. However, punctae are difficult to see. Size of the images should be increased and scale bar should be integrated. The authors should also quantify co-localisation as it is difficult to draw conclusions from 2 single cells.

Response: We appreciate the reviewer’s suggestion. We have increased the size of the images and integrated the scale bar and quantify the co-localization. (**New Fig. S6 and Fig 3f**).

Fig. 3 f: Longer reads to cover also sgRNA sites should be included in the supplement.

Response: We provide the longer reads in the supplemental figure. (**New Fig. S8**).

Fig. 3 g-i: The authors should perform confocal images (incl. quantification) as in c) in order to proof that Let-7a binding to RasGRP1 RNA is lost. Furthermore, the authors should also perform qPCR for Rasgrp1 in order to demonstrate that RNA levels of Rasgrp1 are not altered due to the change in 3’ UTR in their KO cells.

Response: We performed qPCR for Rasgrp1 in 51 bp KO cells and find that RNA levels of Rasgrp1 are not altered in these cells. (**New Fig. S9**).

But we couldn't perform confocal images to observe Let-7a binding to RasGRP1 RNA is lost in 51 bp bases KO cells as in Fig. 3 c. Because we couldn't label the target site of let-7a on RasGRP1 3'UTR in 51 bp bases KO cells.

Figure 4+5:

The authors should demonstrate if injection of their conjugated RNA induces an IFN-response in the animals, as this might also alter translation.

Response: We had analyzed IFN- β in serum of mice injected with the conjugated RNA by ELISA, but the level of IFN- β in serum is very low, which we couldn't detect it (data is not show).

Figure 4:

The main text is misleading. The authors talk about intravenous and intraperitoneal injections. If the authors used i.v. injections for b-f, they should demonstrate that also under these conditions, macrophages take up RasGRP1-3'UTR. In principle it should be possible to detect their conjugated RNA in macrophages present in the lung tissue e.g. by immunohistochemistry using anti-Cy3 antibodies.

Response: We apologize for this oversight. To eliminate the misleading, we co-stained F4/80 and Cy3 in the lung tissue. (**New Fig. S10a**).

Fig. 4a: Please increase image size and add scale bar. Co-localisation should be quantified.

Response: We have increased the size of the images and integrated the scale bar and quantified the co-localization. (**New Fig. 4b**).

Fig. 4b: What are the kinetics of IL-6 regulation after RasGRP1-3'UTR injection? Do the authors observe an altered acute-phase response in these animals?

Response: We detected the level of IL-6 in serum of mice at 12h, 24h, and 36h after RasGRP1-3'UTR injection. But we did not detect the changes of IL-6 (Data was not show) and did not observe an altered acute-phase response in these animals until treated the mice with LPS.

Fig. 4f: Please show higher magnification. IL-6/F4/80 co-staining along with pSTAT3 staining would strengthen the authors' statement.

Response: We have shown higher magnification and stained IL-6, F4/80 and pSTAT3 by green, red and pink respectively. but the pink is very weak. Maybe pSTAT3 is very difficult to stain or only a few of STAT3 has been phosphorylated. (**New Fig. 4h**).

Figure 5:

The authors should present a disease index and histology score. As IL-6 has also a regenerative role in colitis, the authors should also investigate IEC proliferation (Ki67/BrdU) and survival (cl. Casp3).

Response: we provide new experimental data showing histology score, IEC proliferation and survival. (**New Fig. 5i and Fig S11b**).

How do the authors ensure that the conjugated RNA is still present in macrophages over that long period of time (see also above)?

Response: We intravenously injected mice with the conjugated RNA every three days to make sure the conjugated RNA present for a long time. we provide new experimental data showing co-staining F4/80 and Cy3 in the intestinal tissue. (**New Fig S11a**).

Fig. 5f: Please show higher magnification. IL-6/F4/80 co-staining along with pSTAT3 staining would strengthen the authors' statement.

Response: We provide higher magnification and co-stained IL-6, F4/80 and pSTAT3 by green, red and pink respectively. But the pink is very weak. Maybe pSTAT3 is very difficult to stain or only a few of STAT3 has been phosphorylated. (**New Fig. 5h**).

Figure 6:

As indicated above, these data do not fit to the rest of the manuscript. The effect of RasGRP1 on EGFR signalling seems to be independent of its “sponging effect”. Furthermore, the role of EGFR in HCC is a bit more complicated. While EGFR in hepatocytes controls DNA damage repair (Geiger-Maor, 2015 Cancer Res; Guedj et al., 2020 Gut; Lanaya et al. 2014 Nat Cell Biol), it is rather EGFR in TAMs that promotes tumorigenesis through induction of IL-6 secretion. In order to link to the rest of their manuscript, the authors might want to correlate RasGRP1 expression in

TAMs with IL-6 expression/secretion in TAMs.

Or focus on HcPCs as outlined above. To this end, the authors might want to grow Huh7 and HepG2 on low attachment plates to induce HcPC-like spheroids that do up-regulate IL-6.

Response: We appreciate the reviewer's comment. We grow Huh7 on low attachment plates to induce HcPC-like spheroids. We detected the expression of IL-6 and RasGRP1 in HcPC-like spheroids and found that the expression of IL-6 and RasGRP1 were very low. But we found that IL-6 promotes the growth of HcPC-like spheroids significantly and RasGRP1 inhibits the tumor-promotion effect of IL-6 on HcPC-like spheroids. (New Fig.6a and Fig S12).

Fig. 6c+d: How many independent experiments have been analysed? This should be indicated in the Figure legend. I assume the authors normalised to the respective unstimulated samples. This should be indicated.

Response: We appreciate the reviewer's suggestion, and we have provided this information in the Figure legend of our revised MS.

Minor points:

Fig. 2: the authors should carefully check y-axis labeling.

Response: Apologies we have checked carefully and corrected the y-axis labeling in the revised MS.

Fig. S5: Labeling of the Figure (W, K, Q) does not correlate to the Figure legend (W, P, Q). Please correct.

Response: Apologies we have corrected the labeling of the figure in the revised MS.

Fig. 6b: Labeling should be improved and match the Figure legend. Please spell out CCK8 in the legend. This type of proliferation assay might not be familiar to every reader.

Response: we have improved the labeling and spell out CCK8 in the legend in the revised MS.

Fig. 6 j: The labeling and the Figure legend are misleading. What is really presented here?

Response: Apologies we have corrected the figure legend in the revised MS.

REVIEWER COMMENTS

Reviewer #1 (Remarks to the Author):

I would like to thank the authors for their efforts and revised version of their manuscript. Some questions remain to be addressed and I will indicate them below:

Please include correct spelling for RasGRP1 (human protein), RASGRP1 (italics, human gene), Rasgrp1(mouse) and so on in the manuscript to avoid confusion about what it is you are showing.

2.a) thank you for including the experiments (fig. 1f). To allow interpretation of solidity and reasoning behind this assay, please include a description of how this mutant is generated and why it was included in the text., and in the methods please clearly describe the used reporter gene included the IL-6 3'UTR sequence wt/mutant.

b) Fig S1: please include a clear description of used plasmids in the methods section.

c/d) please include untx/sham controls HEK293 for IL-6/RASGRP1 in the figures.
please include detailed description of RASGRP1 WT/mutant 3'UTR in the methods section

3.a) Fig S1q-v: I don't understand where to find this figure: I have fig S1 with graph only. fig s2 has only the RasGRP1 siRNA vs control, not the let-7a mimick experiments? Please address this.

3.b) Please show these viability data in a supplementary figure.

4) New fig. 1C: please adjust your legend, and indicate how often you have performed the western blot experiment.

5) Thanks, protein turnover takes longer, so I understand your additional longer timepoint of 24 hrs. Please indicate clearly in the legend: now the WB experiment is not mentioned at all in the legend. It strengthens your point, that effect is due to mRNA level changes and not protein.

6) Thanks. Could you please explain in the legends and/or methods how many regions of what size you analyzed (or how many cells).

Why are there 4 datapoints per condition in fig s6 when there are 3 indep. experiments indicated in the legend? Please address.

9) The western blot is not helpful at all for FFPE specificity. The only way is to use different control tissues known to be negative for RasGRP1 vs positive control (you can use proteinatlas to find proper control tissues). Please address.

Please provide additional information in the description of scoring (methods section): Have you used specific cut-off values for intensities (median intensity of a specific size area?o how do you get your intensity values?)

Regarding the cell percentages: how have you counted cells? What is the cut-off value for positive qualification?

Regarding the last tumor related RasGRP1 protein part reply:

Indeed RasGRP1 protein has different function in different receptor responses, also depending on presence and function of other GEFs.

The reasoning that an effect of RasGRP1 on T and B cell selection affects tumor induction is weak and suggestive.

RasGRP1 ko mice and humans lacking functional RASGRP1 have severe immune deficiency (salzer et al Nat. Immunology (2016), Baars eji 2021)- and overexpression has been associated to T-ALL (hartzell et al. Science Signal 2013). this is a selection/development error. Partial decreased

RasGRP1 expression/function has been observed in autoimmunity (daley, eLife 2013, baars eji 2021)

The suggestion made here by the authors has no foundation: We are here looking at tumors, and acute inflammatory tissue responses, not at T and B cell selection during development. I recommend removing this part from the text completely, and only refer to the epithelial cancer citation (depeille), which is of relevance here.

Please check/correct legends of figure 6 some of the numbers seem swapped, and please include for the wound healing assay experiment repeat numbers/quantification.

One last recommendation when re-reading the manuscript: The story is quite complex and a graphical model/ abstract summarizing the mode of action of RASGRP1 mRNA versus protein, and the contexts, would help a reader to understand the manuscript better.

Reviewer #2 (Remarks to the Author):

In the revised version the authors have addressed most of my concerns either experimentally or have provided explanation. Regarding the unaltered mRNA levels of IL-6 in presence of let-7a has not been well explained and should be cited as a limitation of the study in the discussion.

Reviewer #3 (Remarks to the Author):

The authors addressed all issues raised by the reviewer. The manuscript has been improved. However, there are still some points that need attention:

(1) Statistics need revision by a statistician. It does not seem that statistical testing has been applied correctly. The authors did not test for data distribution and used only t-test throughout the manuscript. In many subfigures the authors used paired testing which does not seem to be appropriate. Furthermore, when testing against a normalised value, i.e. against 1, t-test does not apply but Wilcoxon signed-rank test should be used (see e.g. Fig. 6f). Not all Figure legends are consistent in indicating the statistical test that was used.

(2) The authors now indicate that the in vitro experiments have been performed three times. However, the mean of data that is displayed does not vary from the previous data representation. Also the standard deviation is very little. This is a little bit surprising given the natural variation of biological systems.

(3) Fig. 4: While there is a clear in vivo effect, it is not evident that macrophages take up RasGRP1-3'UTR for long time. There is hardly any co-staining visible in Figures 4h, S11a. It is therefore still possible that their in vivo effect is due to IFN induction.

(4) Fig. 6+ S12: as suggested by the reviewer, the authors now present the growth of spheroids. While the images in Fig. S12 are convincing, images in Fig. 6 are not. Bright field images would be better to see that these are spheroids. Furthermore, why does the size of spheroid differ so much between Fig. S12 and Fig. 6? A quantification of spheroid size would strengthen the author's statements.

(5) Language needs revision. There are also several typos in the new text.

Point-to-point responses to Reviewers (NCOMMS-21-07341A)

Response to Reviewer #1

Many thanks for the insightful comments and suggestions. We have revised our manuscript as suggestions and performed new experiments which are required to address the concerns. We hope we have addressed your concerns in our revised manuscript. Thank you for your kind consideration. If any questions exist, inform us and we will discuss and perform required experiments.

I would like to thank the authors for their efforts and revised version of their manuscript.

Some questions remain to be addressed and I will indicate them below:

Please include correct spelling for RasGRP1 (human protein), RASGRP1 (itallics, human gene), Rasgrp1(mouse) and so on in the manuscript to avoid confusion about what it is you are showing.

Response: We thank the reviewer's good suggestion and apologies for this confusion. we have corrected these spelling in revised MS.

2.a) thank you for including the experiments (fig. 1f). To allow interpretation of solidity and reasoning behind this assay, please include a description of how this mutant is generated and why it was included in the text., and in the methods please clearly describe the used reporter gene included the IL-6 3'UTR sequence wt/mutant.

Response: We thank the reviewer's good suggestion. we have described how this mutant generated (**new Fig S1c**) and explained why it was included in the text (**page 6**,

line 120-123).

we provide description about *IL-6* 3'UTR reporter gene in the methods section (**page 20, line 428-430**).

b) Fig S1: please include a clear description of used plasmids in the methods section.

Response: we provide description about the plasmids used in Fig S1 at the methods section (**page 20, line 427-428**).

c/d) please include untx/sham controls HEK293 for IL-6/RASGRP1 in the figures.

please include detailed description of RASGRP1 WT/mutant 3'UTR in the methods section

Response: we performed new experiments to detect IL-6/RASGRP1 in HEK293 cells (**Fig S1a and S1d**).

we provide the description about the plasmids of *RASGRP1* WT/mutant 3'UTR in the methods section (**page 20, line 430-431**) and in **Fig S1e** legend.

3.a) Fig S1q-v: I don't understand where to find this figure: I have fig S1 with graph only. fig s2 has only the RasGRP1 siRNA vs control, not the let-7a mimick experiments? Please address this.

Response: We apologize for this oversight. It would be **Fig S3 q-v**.

3.b) Please show these viability data in a supplementary figure.

Response: We have shown the viability of primary macrophages transfected with negative control, *Rasgrp1* siRNA, let-7a and *Rasgrp1* 3' UTR #1 for 3 days. (**Fig S4p**)

4) New fig. 1C: please adjust your legend, and indicate how often you have performed the western blot experiment.

Response: We have adjusted Fig. 1c legend and described the frequency of this western blot experiment in revised MS (**page 31, line 714-719**).

5) Thanks, protein turnover takes longer, so I understand your additional longer timepoint of 24 hrs. Please indicate clearly in the legend: now the WB experiment is not mentioned at all in the legend. It strengthens your point, that effect is due to mRNA level changes and not protein.

Response: We thank the reviewer's helpful suggestion. We provide the description about RasGRP1 expression-level in the Fig S3p legend.

6) Thanks. Could you please explain in the legends and/or methods how many regions of what size you analyzed (or how many cells).

Why are there 4 datapoints per condition in fig s6 when there are 3 indep. experiments indicated in the legend? Please address.

Response: we analyzed 1000 cells in each field and provide description about it in the Fig 4i and Fig 5j legends.

We apologize for this mistake, and we have corrected it in the figure S6 legend.

7) The western blot is not helpful at all for FFPE specificity. The only way is to use different control tissues known to be negative for RasGRP1 vs positive control (you can use protein atlas to find

proper control tissues). Please address.

Response: We thank the reviewer's good suggestion, and we provide a positive control for exhibiting the specificity of anti-RasGRP1 antibody (**New Fig. S14**).

Please provide additional information in the description of scoring (methods section): Have you used specific cut-off values for intensities (median intensity of a specific size area?o how do you get your intensity values?)

Regarding the cell percentages: how have you counted cells? What is the cut-off value for positive qualification?

Response: We thank the reviewer's constructive suggestions, and we provide the description about how we evaluated the scores of intensities of positive staining and how we calculated the percentages of positive-stained cells in methods section (**page 23, line 490-496**).

Regarding the last tumor related RasGRP1 protein part reply:

Indeed RasGRP1 protein has different function in different receptor responses, also depending on presence and function of other GEFs.

The reasoning that an effect of RasGRP1 on T and B cell selection affects tumor induction is weak and suggestive.

RasGRP1 ko mice and humans lacking functional RASGRP1 have severe immune deficiency (salzer et al Nat. Immunology (2016), Baars eji 2021)- and overexpression has been associated to T-ALL (hartzell et al. Science Signal 2013). this is a selection/development error. Partial decreased

RasGRP1 expression/function has been observed in autoimmunity (daley, eLife 2013, baars eji 2021)

The suggestion made here by the authors has no foundation: We are here looking at tumors, and acute inflammatory tissue responses, not at T and B cell selection during development.

I recommend removing this part from the text completely, and only refer to the epithelial cancer citation (depeille), which is of relevance here.

Response: We thank the reviewer's nice suggestion and we have removed this part from the text.

Please check/correct legends of figure 6 some of the numbers seem swapped, and please include for the wound healing assay experiment repeat numbers/quantification.

Response: We apologize for this oversight.

We provide the repeat numbers and quantification of the wound healing assay experiment (**Fig. 6e**).

One last recommendation when re-reading the manuscript: The story is quite complex and a graphical model/ abstract summarizing the mode of action of RASGRP1 mRNA versus protein, and the contexts, would help a reader to understand the manuscript better.

Response: We thank the reviewer's good suggestion and we provide a graphical model for summarizing the action mode of RasGRP1 (**new Fig. 8m**).

Response to Reviewer #2

In the revised version the authors have addressed most of my concerns either experimentally or have provided explanation. Regarding the unaltered mRNA levels of IL-6 in presence of let-7a has not been well explained and should be cited as a limitation of the study in the discussion.

Response: We thank the reviewer's nice suggestion, and we have discussed the limitation of this study in the revised MS (**page 17, line 369-371**).

Response to Reviewer #3:

Many thanks for the insightful comments and suggestions. We have revised our manuscript as suggestions and performed statistical analysis which are required to address the concerns. We hope we have addressed your concerns in our revised manuscript. Thank you for your kind consideration. If any questions exist, inform us and we will discuss and perform required experiments.

The authors addressed all issues raised by the reviewer. The manuscript has been improved.

However, there are still some points that need attention:

(1) Statistics need revision by a statistician. It does not seem that statistical testing has been applied correctly. The authors did not test for data distribution and used only t-test throughout the manuscript. In many subfigures the authors used paired testing which does not seem to be appropriate. Furthermore, when testing against a normalised value, i.e. against 1, t-test does not apply but Wilcoxon signed-rank test should be used (see e.g. Fig. 6f). Not all Figure legends are consistent in indicating the statistical test that was used.

Response: We thank the reviewer's suggestion and apologize for this oversight. We

have revised figure legends of **Fig. 5a-c**, **Fig. 5e** and **Fig. 7e-h** to consistent them with their statistical test.

Under the guidance of a statistician, we have corrected the statistical testing of **Fig. 1c-f**, **Fig. 1k**, **Fig. S1b**, **Fig. 3b-e**, **Fig. 3i-k**, **Fig. 4c**, **Fig. 4i-j**, **Fig. 5a-c**, **Fig. 5e**, **Fig. 5i-k**, **Fig 6h** and **Fig. S9**, which were performed as following:

Fig. 1c (One-way ANOVA Tukey test)

1). Analyzing the data of Fig. 1c.

a. IL-6(CDS+3'UTR); b. Ctrl mimics; c. Let-7a mimics; d. Ctrl inhibitor; e.Let-7a inhibitor																
Groups	a+b		a+c		a+d		a+e		a+b+e		a+c+e		a+b+d		a+c+d	
	Duplicate	Average	Duplicate	Average	Duplicate	Average	Duplicate	Average	Duplicate	Average	Duplicate	Average	Duplicate	Average	Duplicate	Average
Sample 1	1.23		0.62		1.52		1.24		1.27		1.54		1.14		0.78	
	1.39	1.31	0.47	0.55	1.25	1.39	1.46	1.35	1.49	1.38	1.36	1.45	1.43	1.29	0.64	0.71
	1.27		0.69		1.39		1.61		1.39		1.38		1.53		0.53	
Sample 2	1.08	1.18	0.55	0.62	1.21	1.3	1.28	1.45	1.72	1.56	1.67	1.53	1.27	1.4	0.7	0.62
	1.29		0.53		1.43		1.21		1.67		1.35		1.46		0.88	
	1.43	1.36	0.65	0.59	1.67	1.55	1.43	1.32	1.35	1.51	1.19	1.27	1.24	1.35	0.73	0.81

2). Inputting data of fig. 1c in SPSS.

VAR0000 1	VAR0000 2
1.00	1.31
1.00	1.18
1.00	1.36
2.00	.55
2.00	.62
2.00	.59
3.00	1.39
3.00	1.30
3.00	1.55
4.00	1.35
4.00	1.45
4.00	1.32
5.00	1.38
5.00	1.56
5.00	1.51
6.00	1.45
6.00	1.53
6.00	1.27
7.00	1.29
7.00	1.40
7.00	1.35
8.00	.71
8.00	.62
8.00	.81

3). Testing the homogeneity of variance of the data of Fig. 1c.

Test of Homogeneity of Variances

		Levene Statistic	df1	df2	Sig.
VAR00002	Based on Mean	1.008	7	16	.462
	Based on Median	.381	7	16	.900
	Based on Median and with adjusted df	.381	7	11.341	.896
	Based on trimmed mean	.954	7	16	.495

4). Performing one-way ANOVA Tukey test on the data of Fig. 1c.

ANOVA

VAR00002

	Sum of Squares	df	Mean Square	F	Sig.
Between Groups	2.533	7	.362	41.969	.000
Within Groups	.138	16	.009		
Total	2.671	23			

Post Hoc Tests

Multiple Comparisons

Dependent Variable: VAR00002

Tukey HSD

(I) VAR00001	(J) VAR00001	Mean Difference (I-J)	Std. Error	Sig.	95% Confidence Interval	
					Lower Bound	Upper Bound
1.00	2.00	.69667*	.07581	.000	.4342	.9591
	3.00	-.13000	.07581	.679	-.3925	.1325
	4.00	-.09000	.07581	.924	-.3525	.1725
	5.00	-.20000	.07581	.212	-.4625	.0625
	6.00	-.13333	.07581	.653	-.3958	.1291
	7.00	-.06333	.07581	.988	-.3258	.1991
	8.00	.57000*	.07581	.000	.3075	.8325

Fig. 1d (One-way ANOVA Tukey test)

1). Analyzing the data of Fig. 1d.

LPS (6h)								
Groups	Ctrl mimics		Let-7a mimics		Ctrl inhibitor		Let-7a inhibitor	
	Duplicate	Average	Duplicate	Average	Duplicate	Average	Duplicate	Average
Sample 1	1458.23	1608.29	1675.06	1541.81	1509.65	1703.65	1871.53	1717.21
	1758.34		1408.56		1897.65		1562.89	
Sample 2	1924.14	1711.68	1618.01	1543.19	1845.76	1598.47	1520.15	1627.14
	1499.22		1468.37		1351.18		1734.13	
Sample 3	1686.71	1557.47	1530.73	1688.24	1541.37	1590.98	2076.59	1852.92
	1428.22		1845.76		1640.59		1629.26	

2). Inputting data of fig. 1d in SPSS.

VAR0000 1	VAR0000 2
1.00	1608.29
1.00	1711.68
1.00	1557.47
2.00	1541.81
2.00	1543.19
2.00	1688.24
3.00	1703.65
3.00	1598.47
3.00	1590.98
4.00	1717.21
4.00	1627.14
4.00	1852.92

3). Testing the homogeneity of variance of the data of Fig. 1d.

Test of Homogeneity of Variances

		Levene Statistic	df1	df2	Sig.
VAR00002	Based on Mean	.383	3	8	.768
	Based on Median	.169	3	8	.914
	Based on Median and with adjusted df	.169	3	7.066	.914
	Based on trimmed mean	.363	3	8	.782

4). Performing one-way ANOVA Tukey test on the data of Fig. 1d.

ANOVA

VAR00002

	Sum of Squares	df	Mean Square	F	Sig.
Between Groups	33340.113	3	11113.371	1.475	.293
Within Groups	60285.306	8	7535.663		
Total	93625.419	11			

Post Hoc Tests

Multiple Comparisons

Dependent Variable: VAR00002

Tukey HSD

(I) VAR00001	(J) VAR00001	Mean Difference (I-J)	Std. Error	Sig.	95% Confidence Interval	
					Lower Bound	Upper Bound
1.00	2.00	34.73333	70.87860	.959	-192.2449	261.7116
	3.00	-5.22000	70.87860	1.000	-232.1982	221.7582
	4.00	-106.61000	70.87860	.478	-333.5882	120.3682

Fig. 1e (One-way ANOVA Tukey test)

1). Analyzing the data of Fig. 1e.

LPS (6h)								
Groups	Ctrl mimics		Let-7a mimics		Ctrl inhibitor		Let-7a inhibitor	
	Duplicate	Average	Duplicate	Average	Duplicate	Average	Duplicate	Average
Sample 1	1.42	1.29	0.68	0.77	1.41	1.35	1.79	1.69
	1.16		0.86		1.29		1.59	
Sample 2	1.11	1.25	0.9	0.81	1.01	1.11	1.71	1.82
	1.39		0.72		1.21		1.93	
Sample 3	1.31	1.43	0.74	0.65	1.3	1.18	2.13	2.01
	1.55		0.56		1.06		1.89	

2). Inputting data of fig. 1e in SPSS.

VAR0000 1	VAR0000 2
1.00	1.29
1.00	1.25
1.00	1.43
2.00	.77
2.00	.81
2.00	.65
3.00	1.35
3.00	1.11
3.00	1.18
4.00	1.69
4.00	1.82
4.00	2.01

3). Testing the homogeneity of variance of the data of Fig. 1e.

Test of Homogeneity of Variances

		Levene Statistic	df1	df2	Sig.
VAR00002	Based on Mean	.522	3	8	.679
	Based on Median	.268	3	8	.846
	Based on Median and with adjusted df	.268	3	7.190	.846
	Based on trimmed mean	.502	3	8	.691

4). Performing one-way ANOVA Tukey test on the data of Fig. 1e.

ANOVA

VAR00002

	Sum of Squares	df	Mean Square	F	Sig.
Between Groups	1.824	3	.608	42.662	.000
Within Groups	.114	8	.014		
Total	1.938	11			

Post Hoc Tests

Multiple Comparisons

Dependent Variable: VAR00002

Tukey HSD

(I) VAR00001	(J) VAR00001	Mean Difference (I-J)	Std. Error	Sig.	95% Confidence Interval	
					Lower Bound	Upper Bound
1.00	2.00	.58000*	.09747	.002	.2679	.8921
	3.00	.11000	.09747	.684	-.2021	.4221
	4.00	-.51667*	.09747	.003	-.8288	-.2045

Fig. 1f (One-way ANOVA LSD test)

1). Analyzing the data of Fig. 1f.

Groups	Let-7a mimics		
	Mock	IL-6 3'UTR	IL-6 3'UTR mut
	Average	Average	Average
Sample 1	107.32	59.28	114.96
Sample 2	114.91	72.67	105.01
Sample 3	90.89	65.75	112.13

2). Inputting data of fig. 1f in SPSS.

VAR0000 1	VAR0000 2
1.00	107.32
1.00	114.91
1.00	90.89
2.00	65.75
2.00	72.67
2.00	59.28
3.00	114.96
3.00	105.01
3.00	112.13

3). Testing the homogeneity of variance of the data of Fig. 1f.

Test of Homogeneity of Variances

		Levene Statistic	df1	df2	Sig.
VAR00002	Based on Mean	1.469	2	6	.302
	Based on Median	.564	2	6	.597
	Based on Median and with adjusted df	.564	2	3.673	.612
	Based on trimmed mean	1.394	2	6	.318

4). Performing one-way ANOVA LSD test on the data of Fig. 1f.

ANOVA

VAR00002					
	Sum of Squares	df	Mean Square	F	Sig.
Between Groups	3527.264	2	1763.632	23.846	.001
Within Groups	443.753	6	73.959		
Total	3971.017	8			

Post Hoc Tests

Multiple Comparisons

Dependent Variable: VAR00002

LSD

(I) VAR00001	(J) VAR00001	Mean Difference (I- J)	Std. Error	Sig.	95% Confidence Interval	
					Lower Bound	Upper Bound
1.00	2.00	38.47333 [*]	7.02181	.002	21.2916	55.6551
	3.00	-6.32667	7.02181	.402	-23.5084	10.8551

Fig. 1k (One-way ANOVA LSD test)

1). Analyzing the data of Fig. 1k.

Groups	Let-7a mimics		
	Mock	RasGRP1 3'UTR	RasGRP1 3'UTR mut
	Average	Average	Average
Sample 1	93.48	43.67	96.61
Sample 2	113.63	35.38	107.25
Sample 3	102.42	37.17	112.48

2). Inputting data of fig. 1k in SPSS.

VAR0000 1	VAR0000 2
1.00	93.48
1.00	113.63
1.00	102.42
2.00	43.67
2.00	35.38
2.00	37.17
3.00	96.61
3.00	107.25
3.00	112.48

3). Testing the homogeneity of variance of the data of Fig. 1k.

Test of Homogeneity of Variances

		Levene Statistic	df1	df2	Sig.
VAR00002	Based on Mean	.713	2	6	.528
	Based on Median	.483	2	6	.639
	Based on Median and with adjusted df	.483	2	5.162	.642
	Based on trimmed mean	.699	2	6	.534

4). Performing one-way ANOVA LSD test on the data of Fig. 1k.

ANOVA

VAR00002

	Sum of Squares	df	Mean Square	F	Sig.
Between Groups	8607.016	2	4303.508	69.274	.000
Within Groups	372.736	6	62.123		
Total	8979.752	8			

Post Hoc Tests

Multiple Comparisons

Dependent Variable: VAR00002

LSD

(I) VAR00001	(J) VAR00001	Mean Difference (I-J)	Std. Error	Sig.	95% Confidence Interval	
					Lower Bound	Upper Bound
1.00	2.00	64.43667*	6.43546	.000	48.6897	80.1837
	3.00	-2.27000	6.43546	.736	-18.0170	13.4770

Fig. 3b (One-way ANOVA LSD test)

1). Analyzing the data of Fig. 3b.

Groups	Primer 1#			Primer 2#		
	Vector	RasGRP1 3'UTR	RasGRP1 3'UTR mut	Vector	RasGRP1 3'UTR	RasGRP1 3'UTR mut
Sample 1	100	75.62	97.47	100	57.61	96.46
Sample 2	102.21	83.68	105.32	105.33	53.29	103.58
Sample 3	110.35	82.15	111.23	106.24	48.13	89.78

2). Inputting data of fig. 3b in SPSS (Primer 1#).

	VAR00001	VAR00002
1	1.00	100.00
2	1.00	102.21
3	1.00	110.35
4	2.00	75.62
5	2.00	83.68
6	2.00	82.15
7	3.00	97.47
8	3.00	105.32
9	3.00	111.23

3). Testing the homogeneity of variance of the data of Fig. 3b (Primer 1#).

Test of Homogeneity of Variances

		Levene Statistic	df1	df2	Sig.
VAR00002	Based on Mean	.275	2	6	.769
	Based on Median	.178	2	6	.841
	Based on Median and with adjusted df	.178	2	5.827	.841
	Based on trimmed mean	.268	2	6	.773

4). Performing one-way ANOVA Tukey test on the data of Fig. 3b (Primer 1#).

ANOVA

VAR00002

	Sum of Squares	df	Mean Square	F	Sig.
Between Groups	1147.241	2	573.620	17.985	.003
Within Groups	191.367	6	31.894		
Total	1338.608	8			

Post Hoc Tests

Multiple Comparisons

Dependent Variable: VAR00002

LSD

(I) VAR00001	(J) VAR00001	Mean Difference (I- J)	Std. Error	Sig.	95% Confidence Interval	
					Lower Bound	Upper Bound
1.00	2.00	23.70333 [*]	4.61118	.002	12.4202	34.9865
	3.00	-.48667	4.61118	.919	-11.7698	10.7965

5). Inputting data of fig. 3b in SPSS (Primer 2#).

VAR0000 1	VAR0000 2
1.00	100.00
1.00	105.33
1.00	106.24
2.00	57.61
2.00	53.29
2.00	48.13
3.00	96.46
3.00	103.58
3.00	89.78

6). Testing the homogeneity of variance of the data of Fig. 3b (Primer 2#).

Test of Homogeneity of Variances

		Levene Statistic	df1	df2	Sig.
VAR00002	Based on Mean	.433	2	6	.667
	Based on Median	.454	2	6	.655
	Based on Median and with adjusted df	.454	2	5.318	.658
	Based on trimmed mean	.436	2	6	.665

7). Performing one-way ANOVA LSD test on the data of Fig. 3b (Primer 2#).

ANOVA

VAR00002

	Sum of Squares	df	Mean Square	F	Sig.
Between Groups	4538.615	2	2269.308	83.517	.000
Within Groups	163.030	6	27.172		
Total	4701.645	8			

Post Hoc Tests

Multiple Comparisons

Dependent Variable: VAR00002

LSD

(I) VAR00001	(J) VAR00001	Mean Difference (I- J)	Std. Error	Sig.	95% Confidence Interval	
					Lower Bound	Upper Bound
1.00	2.00	50.84667*	4.25611	.000	40.4324	61.2610
	3.00	7.25000	4.25611	.139	-3.1643	17.6643

Fig. 3d (One-way ANOVA Tukey test)

1). Analyzing the data of Fig. 3d.

Groups	NC		Let-7a		C+RasGRP1 3'UTR 1		-7a+RasGRP1 3'UTR		-7a+RasGRP1 3'UTR		-7a+RasGRP1 3'UTR	
	Duplicate	Average	Duplicate	Average	Duplicate	Average	Duplicate	Average	Duplicate	Average	Duplicate	Average
Sample 1	1686.71	1552.53	1468.37	1400.47	1323.37	1481.98	1746.2	1612.39	1226.22	1332.19	1509.65	1372.2
	1418.35		1332.57		1640.59		1478.58		1438.15		1234.75	
Sample 2	1770.57	1629.72	1217.75	1313.15	1758.34	1639.25	1418.35	1518.18	1584.71	1444.93	1341.84	1520.15
	1488.67		1408.55		1520.15		1618.01		1305.15		1698.45	
Sample 3	1314.23	1460.53	1341.84	1468.79	1418.35	1570.25	1951	1730.33	1296.14	1533.35	1663.49	1488.86
	1606.83		1595.73		1722.16		1509.65		1770.57		1314.23	

2). Inputting data of fig. 3d in SPSS.

VAR0000 1	VAR0000 2
1.00	1552.53
1.00	1629.72
1.00	1460.53
2.00	1400.47
2.00	1313.15
2.00	1468.79
3.00	1481.98
3.00	1639.25
3.00	1570.25
4.00	1612.39
4.00	1518.18
4.00	1730.33
5.00	1332.19
5.00	1444.93
5.00	1533.35
6.00	1372.20
6.00	1520.15
6.00	1488.86

3). Testing the homogeneity of variance of the data of Fig. 3d.

Test of Homogeneity of Variances

		Levene Statistic	df1	df2	Sig.
VAR00002	Based on Mean	.095	5	12	.991
	Based on Median	.079	5	12	.994
	Based on Median and with adjusted df	.079	5	11.292	.994
	Based on trimmed mean	.094	5	12	.992

4). Performing one-way ANOVA Tukey test on the data of Fig. 3d.

ANOVA

VAR00002

	Sum of Squares	df	Mean Square	F	Sig.
Between Groups	112465.454	5	22493.091	2.870	.062
Within Groups	94038.258	12	7836.522		
Total	206503.712	17			

Post Hoc Tests

Multiple Comparisons

Dependent Variable: VAR00002

Tukey HSD

(I) VAR00001	(J) VAR00001	Mean Difference (I-J)	Std. Error	Sig.	95% Confidence Interval	
					Lower Bound	Upper Bound
1.00	2.00	153.45667	72.27965	.338	-89.3250	396.2383
	3.00	-16.23333	72.27965	1.000	-259.0150	226.5483
	4.00	-72.70667	72.27965	.907	-315.4883	170.0750
	5.00	110.77000	72.27965	.652	-132.0116	353.5516
	6.00	87.19000	72.27965	.826	-155.5916	329.9716
2.00	1.00	-153.45667	72.27965	.338	-396.2383	89.3250
	3.00	-169.69000	72.27965	.248	-412.4716	73.0916
	4.00	-226.16333	72.27965	.073	-468.9450	16.6183
	5.00	-42.68667	72.27965	.990	-285.4683	200.0950
	6.00	-66.26667	72.27965	.935	-309.0483	176.5150

Fig. 3e (One-way ANOVA Tukey test)

1). Analyzing the data of Fig. 3e.

Groups	NC		Let-7a		C+RasGRP1 3'UTR 1		-7a+RasGRP1 3'UTR		-7a+RasGRP1 3'UTR		-7a+RasGRP1 3'UTR	
	Duplicate	Average	Duplicate	Average	Duplicate	Average	Duplicate	Average	Duplicate	Average	Duplicate	Average
Sample 1	2.97	2.81	1.64	1.73	4.48	4.65	3.34	3.18	1.74	1.93	1.34	1.45
	2.65		1.82		4.82		3.02		2.12		1.56	
	2.18		1.74		5.01		3.47		1.58		1.63	
Sample 2	2.46	2.32	1.56	1.65	4.69	4.85	3.25	3.36	1.86	1.72	1.47	1.55
	2.72		1.69		3.93		3.09		1.65		1.27	
	2.48		1.45		4.33		4.13		2.81		2.95	

2). Inputting data of fig. 3e in SPSS.

VAR0000 1	VAR0000 2
1.00	2.81
1.00	2.32
1.00	2.60
2.00	1.73
2.00	1.65
2.00	1.57
3.00	4.65
3.00	4.85
3.00	4.13
4.00	3.18
4.00	3.36
4.00	2.95
5.00	1.93
5.00	1.72
5.00	1.81
6.00	1.45
6.00	1.55
6.00	1.36

3). Testing the homogeneity of variance of the data of Fig. 3e.

Test of Homogeneity of Variances

		Levene Statistic	df1	df2	Sig.
VAR00002	Based on Mean	2.163	5	12	.127
	Based on Median	.845	5	12	.543
	Based on Median and with adjusted df	.845	5	4.767	.573
	Based on trimmed mean	2.057	5	12	.142

4). Performing one-way ANOVA Tukey test on the data of Fig. 3e.

ANOVA

VAR00002

	Sum of Squares	df	Mean Square	F	Sig.
Between Groups	20.683	5	4.137	92.842	.000
Within Groups	.535	12	.045		
Total	21.218	17			

Post Hoc Tests

Multiple Comparisons

Dependent Variable: VAR00002

Tukey HSD

(I) VAR00001	(J) VAR00001	Mean Difference (I-J)	Std. Error	Sig.	95% Confidence Interval	
					Lower Bound	Upper Bound
1.00	2.00	.92667 [*]	.17235	.002	.3478	1.5056
	3.00	-1.96667 [*]	.17235	.000	-2.5456	-1.3878
	4.00	-.58667 [*]	.17235	.046	-1.1656	-.0078
	5.00	.75667 [*]	.17235	.009	.1778	1.3356
	6.00	1.12333 [*]	.17235	.000	.5444	1.7022
2.00	1.00	-.92667 [*]	.17235	.002	-1.5056	-.3478
	3.00	-2.89333 [*]	.17235	.000	-3.4722	-2.3144
	4.00	-1.51333 [*]	.17235	.000	-2.0922	-.9344
	5.00	-.17000	.17235	.914	-.7489	.4089
	6.00	.19667	.17235	.855	-.3822	.7756

Fig. 3i (One-way ANOVA LSD test)

1). Analyzing the data of Fig. 3i.

Groups	LPS						Poly (I,C)						CpG					
	Control		KO-1#		KO-2#		Control		KO-1#		KO-2#		Control		KO-1#		KO-2#	
	Duplicate	Average	Duplicate	Average	Duplicate	Average	Duplicate	Average	Duplicate	Average	Duplicate	Average	Duplicate	Average	Duplicate	Average	Duplicate	Average
Sample 1	354.59		268.73		298.17		145.01		135.3		90.51		40.5		31.12		33.36	
	284.05	319.32	335.46	302.09	347.29	322.73	95.67	120.34	82.71	109	159.77	125	27.47	33.98	40.79	35.96	29.04	31.2
	396.18		292.04		317.36		92.41		96.34		144.01		34.78		39.67		28.84	
Sample 2	286.03	341.1	354.59	323.31	372.22	344.79	138.14	115.28	107.63	101.99	89.88	116.95	27.67	31.22	28.84	34.25	25.63	27.24
	333.14		377.41		349.71		152.22		82.14		162.02		36.25		32.67		33.59	
	401.71	367.42	319.57	348.49	393.44	371.57	103.97	128.09	146.02	114.08	109.9	135.96	30.06	33.16	43.71	38.19	24.08	28.84

2). Inputting data of fig. 3i in SPSS (LPS).

VAR0000 1	VAR0000 2
1.00	319.32
1.00	341.10
1.00	367.42
2.00	302.09
2.00	323.31
2.00	348.49
3.00	322.73
3.00	344.79
3.00	371.57

3). Testing the homogeneity of variance of the data of Fig. 3i (LPS).

Test of Homogeneity of Variances

		Levene Statistic	df1	df2	Sig.
VAR00002	Based on Mean	.004	2	6	.996
	Based on Median	.003	2	6	.997
	Based on Median and with adjusted df	.003	2	5.988	.997
	Based on trimmed mean	.004	2	6	.996

4). Performing one-way ANOVA LSD test on the data of Fig. 3i (LPS).

ANOVA

VAR00002

	Sum of Squares	df	Mean Square	F	Sig.
Between Groups	809.801	2	404.900	.707	.530
Within Groups	3435.720	6	572.620		
Total	4245.520	8			

Post Hoc Tests

Multiple Comparisons

Dependent Variable: VAR00002

LSD

(I) VAR00001	(J) VAR00001	Mean Difference (I- J)	Std. Error	Sig.	95% Confidence Interval	
					Lower Bound	Upper Bound
1.00	2.00	17.98333	19.53834	.393	-29.8253	65.7919
	3.00	-3.75000	19.53834	.854	-51.5586	44.0586

5). Inputting data of fig. 3i in SPSS (Poly (I:C)).

VAR0000 1	VAR0000 2
1.00	120.34
1.00	115.28
1.00	128.09
2.00	109.00
2.00	101.99
2.00	114.08
3.00	125.00
3.00	116.95
3.00	135.96

6). Testing the homogeneity of variance of the data of Fig. 3i (Poly (I:C)).

Test of Homogeneity of Variances

		Levene Statistic	df1	df2	Sig.
VAR00002	Based on Mean	.346	2	6	.721
	Based on Median	.238	2	6	.795
	Based on Median and with adjusted df	.238	2	5.101	.797
	Based on trimmed mean	.339	2	6	.725

7). Performing one-way ANOVA LSD test on the data of Fig. 3i (Poly (I:C)).

ANOVA

VAR00002

	Sum of Squares	df	Mean Square	F	Sig.
Between Groups	498.528	2	249.264	4.411	.066
Within Groups	339.060	6	56.510		
Total	837.589	8			

Post Hoc Tests

Multiple Comparisons

Dependent Variable: VAR00002

LSD

(I) VAR00001	(J) VAR00001	Mean Difference (I- J)	Std. Error	Sig.	95% Confidence Interval	
					Lower Bound	Upper Bound
1.00	2.00	12.88000	6.13786	.081	-2.1388	27.8988
	3.00	-4.73333	6.13786	.470	-19.7521	10.2855

8). Inputting data of fig. 3i in SPSS (CpG).

VAR0000 1	VAR0000 2
1.00	33.98
1.00	31.22
1.00	33.16
2.00	35.96
2.00	34.25
2.00	38.19
3.00	31.20
3.00	27.24
3.00	29.18

9). Testing the homogeneity of variance of the data of Fig. 3i (CpG).

Test of Homogeneity of Variances

		Levene Statistic	df1	df2	Sig.
VAR00002	Based on Mean	.104	2	6	.903
	Based on Median	.130	2	6	.880
	Based on Median and with adjusted df	.130	2	5.864	.880
	Based on trimmed mean	.106	2	6	.901

10). Performing one-way ANOVA LSD test on the data of Fig. 3i (CpG).

ANOVA

VAR00002					
	Sum of Squares	df	Mean Square	F	Sig.
Between Groups	71.995	2	35.998	10.982	.010
Within Groups	19.667	6	3.278		
Total	91.662	8			

Post Hoc Tests

Multiple Comparisons

Dependent Variable: VAR00002

LSD

(I) VAR00001	(J) VAR00001	Mean Difference (I- J)	Std. Error	Sig.	95% Confidence Interval	
					Lower Bound	Upper Bound
1.00	2.00	-3.34667	1.47823	.064	-6.9638	.2704
	3.00	3.58000	1.47823	.052	-.0371	7.1971

Fig. 3j (One-way ANOVA LSD test)

1). Analyzing the data of Fig. 3j.

Groups	LPS						Poly (I:C)						CpG					
	Control		KO-1#		KO-2#		Control		KO-1#		KO-2#		Control		KO-1#		KO-2#	
	Duplicate	Average	Duplicate	Average	Duplicate	Average	Duplicate	Average	Duplicate	Average	Duplicate	Average	Duplicate	Average	Duplicate	Average	Duplicate	Average
Sample 1	1.21		0.61		0.66		0.44		0.22		0.34		0.19		0.12		0.14	
	1.05	1.13	0.77	0.69	0.74	0.7	0.36	0.4	0.28	0.25	0.26	0.3	0.23	0.21	0.1	0.11	0.1	0.12
Sample 2	1.01		0.76		0.86		0.45		0.28		0.31		0.2		0.09		0.09	
	1.29	1.15	0.66	0.71	0.68	0.77	0.41	0.43	0.24	0.26	0.27	0.29	0.18	0.19	0.13	0.11	0.1	0.11
Sample 3	1.06		0.71		0.71		0.39		0.25		0.27		0.16		0.08		0.13	
	1.32	1.19	0.79	0.75	0.81	0.76	0.45	0.42	0.33	0.29	0.35	0.31	0.2	0.18	0.12	0.1	0.11	0.12

2). Inputting data of fig. 3j in SPSS (LPS).

VAR0000 1	VAR0000 2
1.00	1.13
1.00	1.15
1.00	1.19
2.00	.69
2.00	.71
2.00	.75
3.00	.70
3.00	.77
3.00	.76

3). Testing the homogeneity of variance of the data of Fig. 3j (LPS).

Test of Homogeneity of Variances

		Levene Statistic	df1	df2	Sig.
VAR00002	Based on Mean	.235	2	6	.797
	Based on Median	.018	2	6	.982
	Based on Median and with adjusted df	.018	2	4.844	.982
	Based on trimmed mean	.206	2	6	.819

4). Performing one-way ANOVA Tukey test on the data of Fig. 3j (LPS).

ANOVA

VAR00002

	Sum of Squares	df	Mean Square	F	Sig.
Between Groups	.365	2	.183	165.980	.000
Within Groups	.007	6	.001		
Total	.372	8			

Post Hoc Tests

Multiple Comparisons

Dependent Variable: VAR00002

LSD

(I) VAR00001	(J) VAR00001	Mean Difference (I-J)	Std. Error	Sig.	95% Confidence Interval	
					Lower Bound	Upper Bound
1.00	2.00	.44000*	.02708	.000	.3737	.5063
	3.00	.41333*	.02708	.000	.3471	.4796

5). Inputting data of fig. 3j in SPSS (Poly (I:C)).

VAR00000 1	VAR00000 2
1.00	.40
1.00	.43
1.00	.42
2.00	.25
2.00	.26
2.00	.29
3.00	.30
3.00	.29
3.00	.31

6). Testing the homogeneity of variance of the data of Fig. 3j (Poly (I:C)).

Test of Homogeneity of Variances

		Levene Statistic	df1	df2	Sig.
VAR00002	Based on Mean	1.171	2	6	.372
	Based on Median	.273	2	6	.770
	Based on Median and with adjusted df	.273	2	4.102	.774
	Based on trimmed mean	1.081	2	6	.397

7). Performing one-way ANOVA Tukey test on the data of Fig. 3j (Poly (I:C)).

ANOVA

VAR00002

	Sum of Squares	df	Mean Square	F	Sig.
Between Groups	.037	2	.019	72.826	.000
Within Groups	.002	6	.000		
Total	.039	8			

Post Hoc Tests

Multiple Comparisons

Dependent Variable: VAR00002

LSD

(I) VAR00001	(J) VAR00001	Mean Difference (I-J)	Std. Error	Sig.	95% Confidence Interval	
					Lower Bound	Upper Bound
1.00	2.00	.15000*	.01305	.000	.1181	.1819
	3.00	.11667*	.01305	.000	.0847	.1486

8). Inputting data of fig. 3j in SPSS (CpG).

VAR0000 1	VAR0000 2
1.00	.21
1.00	.19
1.00	.18
2.00	.11
2.00	.11
2.00	.10
3.00	.12
3.00	.11
3.00	.12

9). Testing the homogeneity of variance of the data of Fig. 3j (CpG).

Test of Homogeneity of Variances

		Levene Statistic	df1	df2	Sig.
VAR00002	Based on Mean	2.400	2	6	.171
	Based on Median	.800	2	6	.492
	Based on Median and with adjusted df	.800	2	4.545	.504
	Based on trimmed mean	2.245	2	6	.187

10). Performing one-way ANOVA Tukey test on the data of Fig. 3j (CpG).

ANOVA

VAR00002

	Sum of Squares	df	Mean Square	F	Sig.
Between Groups	.013	2	.007	67.444	.000
Within Groups	.001	6	.000		
Total	.014	8			

Post Hoc Tests

Multiple Comparisons

Dependent Variable: VAR00002

LSD

(I) VAR00001	(J) VAR00001	Mean Difference (I-J)	Std. Error	Sig.	95% Confidence Interval	
					Lower Bound	Upper Bound
1.00	2.00	.08667 [*]	.00816	.000	.0667	.1066
	3.00	.07667 [*]	.00816	.000	.0567	.0966

Fig. 3k (One-way ANOVA Tukey test)

1). Analyzing the data of Fig. 3k.

Groups	LPS								Poly (LC)								CpG							
	a+c		b+c		a+d		b+d		a+c		b+c		a+d		b+d		a+c		b+c		a+d		b+d	
	Duplicate	Average	Duplicate	Average	Duplicate	Average	Duplicate	Average	Duplicate	Average	Duplicate	Average	Duplicate	Average	Duplicate	Average	Duplicate	Average	Duplicate	Average	Duplicate	Average	Duplicate	Average
Sample 1	1.17		0.66		1.53		1.13		0.45		0.26		0.73		0.35		0.21		0.11		0.35		0.16	
	1.01	1.09	0.76	0.71	1.23	1.38	0.89	1.01	0.37	0.41	0.2	0.23	0.55	0.64	0.43	0.39	0.17	0.19	0.15	0.13	0.27	0.31	0.22	0.19
Sample 2	1.04		0.72		1.66		0.91		0.52		0.19		0.54		0.39		0.24		0.12		0.26		0.19	
	1.2	1.12	0.6	0.66	1.4	1.53	1.03	0.97	0.4	0.46	0.25	0.22	0.62	0.58	0.31	0.35	0.18	0.21	0.1	0.11	0.34	0.3	0.25	0.22
Sample 3	1.06		0.79		1.57		0.96		0.43		0.28		0.59		0.44		0.17		0.11		0.36		0.19	
	1.14	1.1	0.65	0.72	1.33	1.45	1.14	1.05	0.37	0.4	0.22	0.25	0.71	0.65	0.4	0.42	0.19	0.18	0.15	0.13	0.28	0.32	0.21	0.2

a. Control. b. KO-1#. c. NC. d. RasGRP1 3'UTR 1#

2). Inputting data of fig. 3k in SPSS (LPS).

VAR0000 1	VAR0000 2
1.00	1.09
1.00	1.12
1.00	1.10
2.00	.71
2.00	.66
2.00	.72
3.00	1.38
3.00	1.53
3.00	1.45
4.00	1.01
4.00	.97
4.00	1.05

3). Testing the homogeneity of variance of the data of Fig. 3k (LPS).

Test of Homogeneity of Variances

		Levene Statistic	df1	df2	Sig.
VAR00002	Based on Mean	1.370	3	8	.320
	Based on Median	1.072	3	8	.414
	Based on Median and with adjusted df	1.072	3	4.758	.443
	Based on trimmed mean	1.355	3	8	.324

4). Performing one-way ANOVA Tukey test on the data of Fig. 3k (LPS).

ANOVA

VAR00002

	Sum of Squares	df	Mean Square	F	Sig.
Between Groups	.873	3	.291	136.924	.000
Within Groups	.017	8	.002		
Total	.890	11			

Post Hoc Tests

Multiple Comparisons

Dependent Variable: VAR00002

Tukey HSD

(I) VAR00001	(J) VAR00001	Mean Difference (I-J)	Std. Error	Sig.	95% Confidence Interval	
					Lower Bound	Upper Bound
1.00	2.00	.40667*	.03764	.000	.2861	.5272
	3.00	-.35000*	.03764	.000	-.4705	-.2295
	4.00	.09333	.03764	.138	-.0272	.2139

5). Inputting data of fig. 3k in SPSS (Poly (I:C)).

VAR00001	VAR00002
1	
1.00	.41
1.00	.46
1.00	.40
2.00	.23
2.00	.22
2.00	.25
3.00	.64
3.00	.58
3.00	.65
4.00	.39
4.00	.35
4.00	.42

6). Testing the homogeneity of variance of the data of Fig. 3k (Poly (I:C)).

Test of Homogeneity of Variances

		Levene Statistic	df1	df2	Sig.
VAR00002	Based on Mean	1.011	3	8	.437
	Based on Median	.211	3	8	.886
	Based on Median and with adjusted df	.211	3	5.853	.885
	Based on trimmed mean	.918	3	8	.475

7). Performing one-way ANOVA Tukey test on the data of Fig. 3k (Poly (I:C)).

ANOVA

VAR00002

	Sum of Squares	df	Mean Square	F	Sig.
Between Groups	.232	3	.077	78.576	.000
Within Groups	.008	8	.001		
Total	.240	11			

Post Hoc Tests

Multiple Comparisons

Dependent Variable: VAR00002

Tukey HSD

(I) VAR00001	(J) VAR00001	Mean Difference (I-J)	Std. Error	Sig.	95% Confidence Interval	
					Lower Bound	Upper Bound
1.00	2.00	.19000*	.02560	.000	.1080	.2720
	3.00	-.20000*	.02560	.000	-.2820	-.1180
	4.00	.03667	.02560	.516	-.0453	.1187

8). Inputting data of fig. 3k in SPSS (CpG).

VAR0000 1	VAR0000 2
1.00	.19
1.00	.21
1.00	.18
2.00	.13
2.00	.11
2.00	.13
3.00	.31
3.00	.30
3.00	.32
4.00	.19
4.00	.22
4.00	.20

9). Testing the homogeneity of variance of the data of Fig. 3k (CpG).

Test of Homogeneity of Variances

		Levene Statistic	df1	df2	Sig.
VAR00002	Based on Mean	.376	3	8	.773
	Based on Median	.121	3	8	.945
	Based on Median and with adjusted df	.121	3	6.914	.945
	Based on trimmed mean	.350	3	8	.790

10). Performing one-way ANOVA Tukey test on the data of Fig. 3k (CpG).

ANOVA

VAR00002

	Sum of Squares	df	Mean Square	F	Sig.
Between Groups	.053	3	.018	101.762	.000
Within Groups	.001	8	.000		
Total	.055	11			

Post Hoc Tests

Multiple Comparisons

Dependent Variable: VAR00002

Tukey HSD

(I) VAR00001	(J) VAR00001	Mean Difference (I-J)	Std. Error	Sig.	95% Confidence Interval	
					Lower Bound	Upper Bound
1.00	2.00	.07000*	.01080	.001	.0354	.1046
	3.00	-.11667*	.01080	.000	-.1513	-.0821
	4.00	-.01000	.01080	.792	-.0446	.0246

Fig. 4c (One-way ANOVA Tukey test)

1). Analyzing the data of Fig. 4c.

Groups	NC	Let-7a	RasGRP1 3'UTR 1#	Let-7a + RasGRP1 3'UTR 1#
Sample 1	141.15	77.84	120.19	114.17
Sample 2	126.31	73.15	162.24	120.99
Sample 3	134.12	82.21	152.31	116.05
Sample 4	112.14	70.91	174.86	109.35
Sample 5	109.16	83.15	154.26	103.21
Sample 6	121.37	78.88	151.03	102.13

2). Inputting data of fig. 4c in SPSS.

VAR0000 1	VAR0000 2
1.00	141.15
1.00	126.31
1.00	134.12
1.00	112.14
1.00	109.16
1.00	121.37
2.00	77.84
2.00	73.15
2.00	82.21
2.00	70.91
2.00	83.15
2.00	78.88
3.00	120.19
3.00	162.24
3.00	152.31
3.00	174.86
3.00	154.26
3.00	151.03
4.00	114.17
4.00	120.99
4.00	116.05
4.00	109.35
4.00	103.21
4.00	102.13

3). Testing the homogeneity of variance of the data of Fig. 4c.

Test of Homogeneity of Variances

		Levene Statistic	df1	df2	Sig.
VAR00002	Based on Mean	1.228	3	20	.326
	Based on Median	1.228	3	20	.326
	Based on Median and with adjusted df	1.228	3	8.356	.359
	Based on trimmed mean	1.231	3	20	.325

4). Performing one-way ANOVA Tukey test on the data of Fig. 4c.

ANOVA

VAR00002

	Sum of Squares	df	Mean Square	F	Sig.
Between Groups	17328.273	3	5776.091	41.119	.000
Within Groups	2809.425	20	140.471		
Total	20137.697	23			

Post Hoc Tests

Multiple Comparisons

Dependent Variable: VAR00002

Tukey HSD

(I) VAR00001	(J) VAR00001	Mean Difference (I-J)	Std. Error	Sig.	95% Confidence Interval	
					Lower Bound	Upper Bound
1.00	2.00	46.35167*	6.84279	.000	27.1991	65.5042
	3.00	-28.44000*	6.84279	.003	-47.5925	-9.2875
	4.00	13.05833	6.84279	.256	-6.0942	32.2109

Fig. 4i (Man-Whitney U test)

1). Analyzing the data of Fig. 4i.

Groups	NC	Let-7a	RasGRP1 3'UTR 1#	Let-7a + RasGRP1 3'UTR 1#
Sample 1	411	291	584	371
Sample 2	347	279	624	433
Sample 3	479	241	547	364
Sample 4	454	258	667	311
Sample 5	376	207	564	334
Sample 6	443	226	513	386

2). Inputting data of fig. 4i in SPSS.

VAR0000 1	VAR0000 2
1.00	411.00
1.00	347.00
1.00	479.00
1.00	454.00
1.00	376.00
1.00	443.00
2.00	291.00
2.00	279.00
2.00	241.00
2.00	258.00
2.00	207.00
2.00	226.00
3.00	584.00
3.00	624.00
3.00	547.00
3.00	667.00
3.00	564.00
3.00	513.00
4.00	371.00
4.00	433.00
4.00	364.00
4.00	311.00
4.00	334.00
4.00	386.00

3). Performing Man-Whitney U test on the data of Fig. 4i (group 1 and group2).

Mann-Whitney Test

		Ranks		
	VAR00001	N	Mean Rank	Sum of Ranks
VAR00002	1.00	6	9.50	57.00
	2.00	6	3.50	21.00
	Total	12		

Test Statistics^a

VAR00002	
Mann-Whitney U	.000
Wilcoxon W	21.000
Z	-2.882
Asymp. Sig. (2-tailed)	.004
Exact Sig. [2*(1-tailed Sig.)]	.002 ^b

4). Performing Man-Whitney U test on the data of Fig. 4i (group 1 and group3).

Mann-Whitney Test

		Ranks		
	VAR00001	N	Mean Rank	Sum of Ranks
VAR00002	1.00	6	3.50	21.00
	3.00	6	9.50	57.00
	Total	12		

Test Statistics^a

VAR00002	
Mann-Whitney U	.000
Wilcoxon W	21.000
Z	-2.882
Asymp. Sig. (2-tailed)	.004
Exact Sig. [2*(1-tailed Sig.)]	.002 ^b

5). Performing Man-Whitney U test on the data of Fig. 4i (group 1 and group4).

Mann-Whitney Test

		Ranks		
	VAR00001	N	Mean Rank	Sum of Ranks
VAR00002	1.00	6	8.33	50.00
	4.00	6	4.67	28.00
	Total	12		

Test Statistics^a

VAR00002	
Mann-Whitney U	7.000
Wilcoxon W	28.000
Z	-1.761
Asymp. Sig. (2-tailed)	.078
Exact Sig. [2*(1-tailed Sig.)]	.093 ^b

Fig. 4j (Man-Whitney U test)

1). Analyzing the data of Fig. 4j.

Groups	NC	Let-7a	RasGRP1 3'UTR 1#	Let-7a + RasGRP1 3'UTR 1#
Sample 1	43	19	162	55
Sample 2	41	32	157	45
Sample 3	46	22	136	53
Sample 4	37	15	121	42
Sample 5	52	24	117	66
Sample 6	65	28	146	37

2). Inputting data of fig. 4j in SPSS.

VAR0000 1	VAR0000 2
1.00	43.00
1.00	41.00
1.00	46.00
1.00	37.00
1.00	52.00
1.00	65.00
2.00	19.00
2.00	32.00
2.00	22.00
2.00	15.00
2.00	24.00
2.00	28.00
3.00	162.00
3.00	157.00
3.00	136.00
3.00	121.00
3.00	117.00
3.00	146.00
4.00	55.00
4.00	45.00
4.00	53.00
4.00	42.00
4.00	66.00
4.00	37.00

3). Performing Man-Whitney U test on the data of Fig. 4j (group 1 and group2).

Mann-Whitney Test

		Ranks		
	VAR00001	N	Mean Rank	Sum of Ranks
VAR00002	1.00	6	9.50	57.00
	2.00	6	3.50	21.00
	Total	12		

Test Statistics^a

VAR00002	
Mann-Whitney U	.000
Wilcoxon W	21.000
Z	-2.882
Asymp. Sig. (2-tailed)	.004
Exact Sig. [2*(1-tailed Sig.)]	.002 ^b

4). Performing Man-Whitney U test on the data of Fig. 4j (group 1 and group3).

Mann-Whitney Test

		Ranks		
	VAR00001	N	Mean Rank	Sum of Ranks
VAR00002	1.00	6	3.50	21.00
	3.00	6	9.50	57.00
	Total	12		

Test Statistics^a

VAR00002	
Mann-Whitney U	.000
Wilcoxon W	21.000
Z	-2.882
Asymp. Sig. (2-tailed)	.004
Exact Sig. [2*(1-tailed Sig.)]	.002 ^b

5). Performing Man-Whitney U test on the data of Fig. 4j (group 1 and group4).

Mann-Whitney Test

		Ranks		
	VAR00001	N	Mean Rank	Sum of Ranks
VAR00002	1.00	6	5.92	35.50
	4.00	6	7.08	42.50
	Total	12		

Test Statistics^a

VAR00002	
Mann-Whitney U	14.500
Wilcoxon W	35.500
Z	-.561
Asymp. Sig. (2-tailed)	.575
Exact Sig. [2*(1-tailed Sig.)]	.589 ^b

Fig. 5a (One-way ANOVA Tukey test)

1). Analyzing the data of Fig. 5a.

Groups	NC	Let-7a	RasGRP1 3'UTR 1#	Let-7a + RasGRP1 3'UTR 1#
Sample 1	14.2	17	12.1	14
Sample 2	14.3	17.1	13.1	14.6
Sample 3	15.1	17.5	12.5	14.8
Sample 4	15.5	16.8	13.3	15
Sample 5	15	18	12.3	15.6
Sample 6	14.5	18.2	13.3	15.5

2). Inputting data of fig. 5a in SPSS.

VAR0000 1	VAR0000 2
1.00	14.20
1.00	14.30
1.00	15.10
1.00	15.50
1.00	15.00
1.00	14.50
2.00	17.00
2.00	17.10
2.00	17.50
2.00	16.80
2.00	18.00
2.00	18.20
3.00	12.10
3.00	13.10
3.00	12.50
3.00	13.30
3.00	12.30
3.00	13.30
4.00	14.00
4.00	14.60
4.00	14.80
4.00	15.00
4.00	15.60
4.00	15.50

3). Testing the homogeneity of variance of the data of Fig. 5a.

Test of Homogeneity of Variances

		Levene Statistic	df1	df2	Sig.
VAR00002	Based on Mean	.026	3	20	.994
	Based on Median	.024	3	20	.995
	Based on Median and with adjusted df	.024	3	15.297	.995
	Based on trimmed mean	.026	3	20	.994

4). Performing one-way ANOVA Tukey test on the data of Fig. 5a.

ANOVA

VAR00002

	Sum of Squares	df	Mean Square	F	Sig.
Between Groups	65.801	3	21.934	71.816	.000
Within Groups	6.108	20	.305		
Total	71.910	23			

Post Hoc Tests

Multiple Comparisons

Dependent Variable: VAR00002

Tukey HSD

(I) VAR00001	(J) VAR00001	Mean Difference (I-J)	Std. Error	Sig.	95% Confidence Interval	
					Lower Bound	Upper Bound
1.00	2.00	-2.66667 [*]	.31907	.000	-3.5597	-1.7736
	3.00	2.00000 [*]	.31907	.000	1.1069	2.8931
	4.00	-.15000	.31907	.965	-1.0431	.7431

Fig. 5b (One-way ANOVA Tukey test)

1). Analyzing the data of Fig. 5b.

Groups	NC	Let-7a	RasGRP1 3'UTR 1#	Let-7a + RasGRP1 3'UTR 1#
Sample 1	16.6	17.8	17.3	17.4
Sample 2	16.9	18.3	17.5	17.9
Sample 3	17.3	17.1	16.4	18.3
Sample 4	17.2	18.5	16.9	17.3
Sample 5	17.5	18.1	17.4	17.9
Sample 6	17.9	18.3	17.9	18.6

2). Inputting data of fig. 5b in SPSS.

VAR0000 1	VAR0000 2
1.00	16.60
1.00	16.90
1.00	17.30
1.00	17.20
1.00	17.50
1.00	17.90
2.00	17.80
2.00	18.30
2.00	17.10
2.00	18.50
2.00	18.10
2.00	18.30
3.00	17.30
3.00	17.50
3.00	16.40
3.00	16.90
3.00	17.40
3.00	17.90
4.00	17.40
4.00	17.90
4.00	18.30
4.00	17.30
4.00	17.90
4.00	18.60

3). Testing the homogeneity of variance of the data of Fig. 5b.

Test of Homogeneity of Variances

		Levene Statistic	df1	df2	Sig.
VAR00002	Based on Mean	.041	3	20	.989
	Based on Median	.014	3	20	.998
	Based on Median and with adjusted df	.014	3	18.614	.998
	Based on trimmed mean	.034	3	20	.991

4). Performing one-way ANOVA Tukey test on the data of Fig. 5b.

ANOVA

VAR00002

	Sum of Squares	df	Mean Square	F	Sig.
Between Groups	3.195	3	1.065	4.316	.017
Within Groups	4.935	20	.247		
Total	8.130	23			

Post Hoc Tests

Multiple Comparisons

Dependent Variable: VAR00002

Tukey HSD

(I) VAR00001	(J) VAR00001	Mean Difference (I-J)	Std. Error	Sig.	95% Confidence Interval	
					Lower Bound	Upper Bound
1.00	2.00	-.78333	.28679	.057	-1.5860	.0194
	3.00	.00000	.28679	1.000	-.8027	.8027
	4.00	-.66667	.28679	.126	-1.4694	.1360

Fig. 5c (One-way ANOVA Tukey test)

1). Analyzing the data of Fig. 5c.

Groups	NC	Let-7a	RasGRP1 3'UTR 1#	Let-7a + RasGRP1 3'UTR 1#
Sample 1	6.02	7.63	4.9	5.92
Sample 2	6.23	6.79	5.14	6.43
Sample 3	6.41	7.54	4.52	6.88
Sample 4	6.43	7.33	4.63	6.81
Sample 5	5.94	8.12	5.37	7.14
Sample 6	5.51	7.28	5.31	7.4

2). Inputting data of fig. 5c in SPSS.

VAR0000 1	VAR0000 2
1.00	6.02
1.00	6.23
1.00	6.41
1.00	6.43
1.00	5.94
1.00	5.51
2.00	7.63
2.00	6.79
2.00	7.54
2.00	7.33
2.00	8.12
2.00	7.28
3.00	4.90
3.00	5.14
3.00	4.52
3.00	4.63
3.00	5.37
3.00	5.31
4.00	5.92
4.00	6.43
4.00	6.88
4.00	6.81
4.00	7.14
4.00	7.40

3). Testing the homogeneity of variance of the data of Fig. 5c.

Test of Homogeneity of Variances

		Levene Statistic	df1	df2	Sig.
VAR00002	Based on Mean	.311	3	20	.817
	Based on Median	.210	3	20	.889
	Based on Median and with adjusted df	.210	3	15.035	.888
	Based on trimmed mean	.286	3	20	.835

4). Performing one-way ANOVA Tukey test on the data of Fig. 5c.

ANOVA

VAR00002

	Sum of Squares	df	Mean Square	F	Sig.
Between Groups	19.936	3	6.645	37.103	.000
Within Groups	3.582	20	.179		
Total	23.518	23			

Post Hoc Tests

Multiple Comparisons

Dependent Variable: VAR00002

Tukey HSD

(I) VAR00001	(J) VAR00001	Mean Difference (I-J)	Std. Error	Sig.	95% Confidence Interval	
					Lower Bound	Upper Bound
1.00	2.00	-1.35833*	.24434	.000	-2.0422	-.6744
	3.00	1.11167*	.24434	.001	.4278	1.7956
	4.00	-.67333	.24434	.055	-1.3572	.0106

Fig. 5e (One-way ANOVA Tukey test)

1). Analyzing the data of Fig. 5e.

Groups	NC	Let-7a	RasGRP1 3'UTR 1#	Let-7a + RasGRP1 3'UTR 1#
Sample 1	121.17	32.14	155.26	109.74
Sample 2	135.24	35.85	176.79	112.33
Sample 3	126.86	22.33	153.34	118.4
Sample 4	108.32	30.18	175.08	154.82
Sample 5	112.97	33.82	181.16	123.03
Sample 6	119.21	31.2	173.93	113.27

2). Inputting data of fig. 5e in SPSS.

VAR0000 1	VAR0000 2
1.00	121.17
1.00	135.24
1.00	126.86
1.00	108.32
1.00	112.97
1.00	119.21
2.00	32.14
2.00	35.85
2.00	22.33
2.00	30.18
2.00	33.82
2.00	31.20
3.00	155.26
3.00	176.79
3.00	153.34
3.00	175.08
3.00	181.16
3.00	173.93
4.00	109.74
4.00	112.33
4.00	118.40
4.00	154.82
4.00	123.03
4.00	113.27

3). Testing the homogeneity of variance of the data of Fig. 5e.

Test of Homogeneity of Variances

		Levene Statistic	df1	df2	Sig.
VAR00002	Based on Mean	1.653	3	20	.209
	Based on Median	.660	3	20	.586
	Based on Median and with adjusted df	.660	3	11.189	.593
	Based on trimmed mean	1.452	3	20	.258

4). Performing one-way ANOVA Tukey test on the data of Fig. 5e.

ANOVA

VAR00002

	Sum of Squares	df	Mean Square	F	Sig.
Between Groups	60113.059	3	20037.686	149.089	.000
Within Groups	2688.015	20	134.401		
Total	62801.075	23			

Post Hoc Tests

Multiple Comparisons

Dependent Variable: VAR00002

Tukey HSD

(I) VAR00001	(J) VAR00001	Mean Difference (I-J)	Std. Error	Sig.	95% Confidence Interval	
					Lower Bound	Upper Bound
1.00	2.00	89.70833 [*]	6.69330	.000	70.9742	108.4425
	3.00	-48.63167 [*]	6.69330	.000	-67.3658	-29.8975
	4.00	-1.30333	6.69330	.997	-20.0375	17.4308

Fig. 5i (Man-Whitney U test)

1). Analyzing the data of Fig. 5i.

Groups	NC	Let-7a	RasGRP1 3'UTR 1#	Let-7a + RasGRP1 3'UTR 1#
Sample 1	9.8	3.2	13.9	7.9
Sample 2	8.3	2.5	11.9	7.3
Sample 3	7.1	2.7	13.3	5.9
Sample 4	6.9	2.4	12.4	7.5
Sample 5	7.6	2.8	13.1	7
Sample 6	9.2	2.6	12.7	6.8

2). Inputting data of fig. 5i in SPSS.

VAR0000 1	VAR0000 2
1.00	9.80
1.00	8.30
1.00	7.10
1.00	6.90
1.00	7.60
1.00	9.20
2.00	3.20
2.00	2.50
2.00	2.70
2.00	2.40
2.00	2.80
2.00	2.60
3.00	13.90
3.00	11.90
3.00	13.30
3.00	12.40
3.00	13.10
3.00	12.70
4.00	7.90
4.00	7.30
4.00	5.90
4.00	7.50
4.00	7.00
4.00	6.80

3). Performing Man-Whitney U test on the data of Fig. 5i (group 1 and group2).

Mann-Whitney Test

		Ranks		
	VAR00001	N	Mean Rank	Sum of Ranks
VAR00002	1.00	6	9.50	57.00
	2.00	6	3.50	21.00
	Total	12		

Test Statistics^a

VAR00002	
Mann-Whitney U	.000
Wilcoxon W	21.000
Z	-2.882
Asymp. Sig. (2-tailed)	.004
Exact Sig. [2*(1-tailed Sig.)]	.002 ^b

4). Performing Man-Whitney U test on the data of Fig. 5i (group 1 and group3).

Mann-Whitney Test

		Ranks		
	VAR00001	N	Mean Rank	Sum of Ranks
VAR00002	1.00	6	3.50	21.00
	3.00	6	9.50	57.00
	Total	12		

Test Statistics^a

VAR00002	
Mann-Whitney U	.000
Wilcoxon W	21.000
Z	-2.882
Asymp. Sig. (2-tailed)	.004
Exact Sig. [2*(1-tailed Sig.)]	.002 ^b

5). Performing Man-Whitney U test on the data of Fig. 5i (group 1 and group4).

Mann-Whitney Test

		Ranks		
	VAR00001	N	Mean Rank	Sum of Ranks
VAR00002	1.00	6	8.17	49.00
	4.00	6	4.83	29.00
	Total	12		

Test Statistics^a

VAR00002	
Mann-Whitney U	8.000
Wilcoxon W	29.000
Z	-1.601
Asymp. Sig. (2-tailed)	.109
Exact Sig. [2*(1-tailed Sig.)]	.132 ^b

Fig. 5j (Man-Whitney U test)

1). Analyzing the data of Fig. 5j.

Groups	NC	Let-7a	RasGRP1 3'UTR 1#	Let-7a + RasGRP1 3'UTR 1#
Sample 1	324	146	415	321
Sample 2	348	123	434	286
Sample 3	352	157	452	264
Sample 4	367	141	467	259
Sample 5	359	111	403	293
Sample 6	333	158	474	312

2). Inputting data of fig. 5j in SPSS.

VAR0000 1	VAR0000 2
1.00	324.00
1.00	348.00
1.00	352.00
1.00	367.00
1.00	359.00
1.00	333.00
2.00	146.00
2.00	123.00
2.00	157.00
2.00	141.00
2.00	111.00
2.00	158.00
3.00	415.00
3.00	434.00
3.00	452.00
3.00	467.00
3.00	403.00
3.00	474.00
4.00	321.00
4.00	286.00
4.00	264.00
4.00	259.00
4.00	293.00
4.00	312.00

3). Performing Man-Whitney U test on the data of Fig. 5j (group 1 and group2).

Mann-Whitney Test

		Ranks		
	VAR00001	N	Mean Rank	Sum of Ranks
VAR00002	1.00	6	9.50	57.00
	2.00	6	3.50	21.00
	Total	12		

Test Statistics^a

		VAR00002
Mann-Whitney U		.000
Wilcoxon W		21.000
Z		-2.882
Asymp. Sig. (2-tailed)		.004
Exact Sig. [2*(1-tailed Sig.)]		.002 ^b

4). Performing Man-Whitney U test on the data of Fig. 5j (group 1 and group3).

Mann-Whitney Test

		Ranks		
	VAR00001	N	Mean Rank	Sum of Ranks
VAR00002	1.00	6	3.50	21.00
	3.00	6	9.50	57.00
	Total	12		

Test Statistics^a

		VAR00002
Mann-Whitney U		.000
Wilcoxon W		21.000
Z		-2.882
Asymp. Sig. (2-tailed)		.004
Exact Sig. [2*(1-tailed Sig.)]		.002 ^b

5). Performing Man-Whitney U test on the data of Fig. 5j (group 1 and group4).

Mann-Whitney Test

		Ranks		
	VAR00001	N	Mean Rank	Sum of Ranks
VAR00002	1.00	6	9.50	57.00
	4.00	6	3.50	21.00
	Total	12		

Test Statistics^a

	VAR00002
Mann-Whitney U	.000
Wilcoxon W	21.000
Z	-2.882
Asymp. Sig. (2-tailed)	.004
Exact Sig. [2*(1-tailed Sig.)]	.002 ^b

Fig. 5k (Man-Whitney U test)

1). Analyzing the data of Fig. 5k.

Groups	NC	Let-7a	RasGRP1 3'UTR 1#	Let-7a + RasGRP1 3'UTR 1#
Sample 1	47	23	81	39
Sample 2	44	25	89	42
Sample 3	43	24	74	37
Sample 4	38	33	95	37
Sample 5	48	22	77	33
Sample 6	39	27	84	45

2). Inputting data of fig. 5k in SPSS.

VAR0000 1	VAR0000 2
1.00	47.00
1.00	44.00
1.00	43.00
1.00	38.00
1.00	48.00
1.00	39.00
2.00	23.00
2.00	25.00
2.00	24.00
2.00	33.00
2.00	22.00
2.00	27.00
3.00	81.00
3.00	89.00
3.00	74.00
3.00	95.00
3.00	77.00
3.00	84.00
4.00	39.00
4.00	42.00
4.00	37.00
4.00	37.00
4.00	33.00
4.00	45.00

3). Performing Man-Whitney U test on the data of Fig. 5k (group 1 and group2).

Mann-Whitney Test

		Ranks		
	VAR00001	N	Mean Rank	Sum of Ranks
VAR00002	1.00	6	9.50	57.00
	2.00	6	3.50	21.00
	Total	12		

Test Statistics^a

		VAR00002
Mann-Whitney U		.000
Wilcoxon W		21.000
Z		-2.882
Asymp. Sig. (2-tailed)		.004
Exact Sig. [2*(1-tailed Sig.)]		.002 ^b

4). Performing Man-Whitney U test on the data of Fig. 5k (group 1 and group3).

Mann-Whitney Test

		Ranks		
	VAR00001	N	Mean Rank	Sum of Ranks
VAR00002	1.00	6	3.50	21.00
	3.00	6	9.50	57.00
	Total	12		

Test Statistics^a

		VAR00002
Mann-Whitney U		.000
Wilcoxon W		21.000
Z		-2.882
Asymp. Sig. (2-tailed)		.004
Exact Sig. [2*(1-tailed Sig.)]		.002 ^b

5). Performing Man-Whitney U test on the data of Fig. 5k (group 1 and group4).

Mann-Whitney Test

		Ranks		
	VAR00001	N	Mean Rank	Sum of Ranks
VAR00002	1.00	6	8.25	49.50
	4.00	6	4.75	28.50
	Total	12		

Test Statistics^a

VAR00002	
Mann-Whitney U	7.500
Wilcoxon W	28.500
Z	-1.687
Asymp. Sig. (2-tailed)	.092
Exact Sig. [2*(1-tailed Sig.)]	.093 ^b

Fig. 6h (Wilcoxon signed-rank test)

1). Analyzing the data of Fig. 6h.

Groups	EGF 1 min		EGF 3 min		EGF 5 min	
	Ctrl	RasGRP1	Ctrl	RasGRP1	Ctrl	RasGRP1
Sample 1	3.52	2.13	6.13	2.51	10.05	3.09
Sample 2	3.59	2.39	6.39	2.79	10.31	3.42
Sample 3	3.81	2.21	6.01	2.92	9.82	3.51
Sample 4	3.32	2.18	6.52	2.58	9.91	3.28
Sample 5	3.48	2.48	6.48	2.72	10.13	3.21
Sample 6	3.11	2.41	6.67	2.41	10.19	3.32

2). Inputting data of fig. 6h in SPSS (EGF 1 min).

VAR00001	VAR00002
3.52	2.13
3.59	2.39
3.81	2.21
3.32	2.18
3.48	2.48
3.11	2.41

3). Performing Wilcoxon signed-rank test on the data of Fig. 6h (EGF 1 min).

Wilcoxon Signed Ranks Test

		Ranks		
		N	Mean Rank	Sum of Ranks
VAR00002 - VAR00001	Negative Ranks	6 ^a	3.50	21.00
	Positive Ranks	0 ^b	.00	.00
	Ties	0 ^c		
	Total	6		

a. VAR00002 < VAR00001

b. VAR00002 > VAR00001

c. VAR00002 = VAR00001

Test Statistics^a

	VAR00002 - VAR00001
Z	-2.201 ^b
Asymp. Sig. (2-tailed)	.028

4). Inputting data of fig. 6h in SPSS (EGF 3 min).

VAR0000 1	VAR0000 2
6.13	2.51
6.39	2.79
6.01	2.92
6.52	2.58
6.48	2.72
6.67	2.41

5). Performing Wilcoxon signed-rank test on the data of Fig. 6h (EGF 3 min).

Wilcoxon Signed Ranks Test

		Ranks		
		N	Mean Rank	Sum of Ranks
VAR00002 - VAR00001	Negative Ranks	6 ^a	3.50	21.00
	Positive Ranks	0 ^b	.00	.00
	Ties	0 ^c		
	Total	6		

a. VAR00002 < VAR00001

b. VAR00002 > VAR00001

c. VAR00002 = VAR00001

Test Statistics^a

	VAR00002 - VAR00001
Z	-2.201 ^b
Asymp. Sig. (2-tailed)	.028

6). Inputting data of fig. 6h in SPSS (EGF 5 min).

VAR00001	VAR00002
10.05	3.09
10.31	3.42
9.82	3.51
9.91	3.28
10.13	3.21
10.19	3.32

7). Performing Wilcoxon signed-rank test on the data of Fig. 6h (EGF 5 min).

Wilcoxon Signed Ranks Test

		Ranks		
		N	Mean Rank	Sum of Ranks
VAR00002 - VAR00001	Negative Ranks	6 ^a	3.50	21.00
	Positive Ranks	0 ^b	.00	.00
	Ties	0 ^c		
	Total	6		

a. VAR00002 < VAR00001

b. VAR00002 > VAR00001

c. VAR00002 = VAR00001

Test Statistics^a

		VAR00002 - VAR00001
Z		-2.201 ^b
Asymp. Sig. (2-tailed)		.028

Fig. S9 (One-way ANOVA LSD test)

1). Analyzing the data of Fig. S9.

Groups	LPS						Poly (I:C)						CpG					
	Control		KO-1#		KO-2#		Control		KO-1#		KO-2#		Control		KO-1#		KO-2#	
	Duplicate	Average	Duplicate	Average	Duplicate	Average	Duplicate	Average	Duplicate	Average	Duplicate	Average	Duplicate	Average	Duplicate	Average	Duplicate	Average
Sample 1	198.09		261.38		194.01		112.99		71.51		76.64		22.47		24.59		23.75	
	242.19	220.14	221.32	241.35	235.57	214.79	78.79	95.89	108.38	89.94	93.7	85.17	27.47	24.97	19.29	21.94	30.06	26.91
	296.11		229.13		247.28		86.22		116.16		82.14		29.04		24.76		32.67	
Sample 2	216.77	256.44	195.36	212.24	215.27	231.27	120.26	103.24	75.58	95.87	67.65	74.9	22.63	25.83	29.45	27.1	27.47	30.07
	224.41		233.94		286.03		106.15		120.26		108.38		26.54		31.34		19.56	
	179.77	202.09	280.14	257.04	243.88	264.95	68.59	87.37	89.26	104.76	85.63	97.01	31.12	28.83	26.72	29.03	22.32	20.94
Sample 3																		

2). Inputting data of fig. S9 in SPSS (LPS).

VAR0000 3	VAR0000 4
1.00	220.14
1.00	256.44
1.00	202.09
2.00	241.35
2.00	212.24
2.00	257.04
3.00	214.79
3.00	231.27
3.00	264.95

3). Testing the homogeneity of variance of the data of Fig. S9 (LPS).

Test of Homogeneity of Variances

		Levene Statistic	df1	df2	Sig.
VAR00004	Based on Mean	.078	2	6	.926
	Based on Median	.028	2	6	.973
	Based on Median and with adjusted df	.028	2	5.821	.973
	Based on trimmed mean	.074	2	6	.930

4). Performing one-way ANOVA LSD test on the data of Fig. S9 (LPS).

ANOVA

VAR00004					
	Sum of Squares	df	Mean Square	F	Sig.
Between Groups	229.718	2	114.859	.178	.841
Within Groups	3873.327	6	645.555		
Total	4103.045	8			

Post Hoc Tests

Multiple Comparisons

Dependent Variable: VAR00004

LSD

(I) VAR00003	(J) VAR00003	Mean Difference (I- J)	Std. Error	Sig.	95% Confidence Interval	
					Lower Bound	Upper Bound
1.00	2.00	-10.65333	20.74535	.626	-61.4154	40.1087
	3.00	-10.78000	20.74535	.622	-61.5421	39.9821

5). Inputting data of fig. S9 in SPSS (Poly (I:C)).

VAR0000 1	VAR0000 2
4.00	95.89
4.00	103.24
4.00	87.37
5.00	89.94
5.00	95.87
5.00	104.76
6.00	85.17
6.00	74.90
6.00	97.01

6). Testing the homogeneity of variance of the data of Fig. S9 (Poly (I:C)).

Test of Homogeneity of Variances

		Levene Statistic	df1	df2	Sig.
VAR00002	Based on Mean	.208	2	6	.818
	Based on Median	.187	2	6	.834
	Based on Median and with adjusted df	.187	2	5.353	.835
	Based on trimmed mean	.207	2	6	.819

7). Performing one-way ANOVA LSD test on the data of Fig. S9 (Poly (I:C)).

ANOVA

VAR00002

	Sum of Squares	df	Mean Square	F	Sig.
Between Groups	222.631	2	111.316	1.385	.320
Within Groups	482.270	6	80.378		
Total	704.901	8			

Post Hoc Tests

Multiple Comparisons

Dependent Variable: VAR00002

LSD

(I) VAR00001	(J) VAR00001	Mean Difference (I- J)	Std. Error	Sig.	95% Confidence Interval	
					Lower Bound	Upper Bound
4.00	5.00	-1.35667	7.32022	.859	-19.2686	16.5553
	6.00	9.80667	7.32022	.229	-8.1053	27.7186

8). Inputting data of fig. S9 in SPSS (CpG).

VAR0000 1	VAR0000 2
7.00	24.97
7.00	25.83
7.00	28.83
8.00	21.94
8.00	27.10
8.00	29.03
9.00	26.91
9.00	30.07
9.00	20.94

9). Testing the homogeneity of variance of the data of Fig. S9 (CpG).

Test of Homogeneity of Variances

		Levene Statistic	df1	df2	Sig.
VAR00002	Based on Mean	1.031	2	6	.412
	Based on Median	.390	2	6	.693
	Based on Median and with adjusted df	.390	2	4.986	.696
	Based on trimmed mean	.973	2	6	.431

10). Performing one-way ANOVA LSD test on the data of Fig. S9 (CpG).

ANOVA

VAR00002

	Sum of Squares	df	Mean Square	F	Sig.
Between Groups	.598	2	.299	.023	.977
Within Groups	78.080	6	13.013		
Total	78.678	8			

Post Hoc Tests

Multiple Comparisons

Dependent Variable: VAR00002

LSD

(I) VAR00001	(J) VAR00001	Mean Difference (I- J)	Std. Error	Sig.	95% Confidence Interval	
					Lower Bound	Upper Bound
7.00	8.00	.52000	2.94544	.866	-6.6872	7.7272
	9.00	.57000	2.94544	.853	-6.6372	7.7772

(2) The authors now indicate that the in vitro experiments have been performed three times. However, the mean of data that is displayed does not vary from the previous data representation. Also the standard deviation is very little. This is a little bit surprising given the natural variation of biological systems.

Response: We appreciate the reviewer's comment. To reduce the influence of natural variation of biological systems on the effect of RasGRP1, we usually separated and cultured peritoneal macrophages at the same conditions from littermate male mice and carry out the in vitro experiments also at the same conditions.

(3) Fig. 4: While there is a clear in vivo effect, it is not evident that macrophages take up RasGRP1-3' UTR for long time. There is hardly any co-staining visible in Figures 4h, S11a. It is therefore still possible that their in vivo effect is due to IFN induction.

Response: According to reviewer's first-round good suggestion, we co-stained F4/80 and Cy3 in Figure S11a. The results show that F4/80 mainly located at membrane of macrophages and a part of Cy3-labeled RasGRP1-3' UTR mainly located at cytoplasm of macrophages, which may indicate that macrophages took up a part of Cy3-labeled RasGRP1-3' UTR for three days or for longer time.

I agree with the reviewer's opinion that the in vivo effect may partly be due to IFN induction.

(4) Fig. 6+ S12: as suggested by the reviewer, the authors now present the growth of spheroids. While the images in Fig. S12 are convincing, images in Fig. 6 are not. Bright field images would be

better to see that these are spheroids. Furthermore, why does the size of spheroid differ so much between Fig. S12 and Fig. 6? A quantification of spheroid size would strengthen the author's statements.

Response: We appreciate the reviewer's suggestion and provide bright field images of spheroids (**New Fig. S13**).

We quantify the spheroid size (**New Fig. 6b**), and we think that lentivirus or green fluorescent protein may affect the growth of spheroid.

(5) Language needs revision. There are also several typos in the new text.

Response: We apologize for the typos. We have checked carefully to correct the typos and revised the language with the help of native English-speaking editors at SNAS.

REVIEWER COMMENTS

Reviewer #1 (Remarks to the Author):

Thank you authors for providing once again additional information and figures.

Most points have now been addressed, apart from one which is a rather essential point that has not been properly answered. This entails the control for the functionality of the used RASGRP1 protein staining by antibody of the patient tissues in Figure 7.

I have pointed the authors towards tools they can use to find proper control tissues, however, to my disappointment, the figure included (Fig S14) has merely 1 image of an IHC staining of mouse intestine. This is most definitely not a suitable control for human tissues, it is a different species, and one image compared to nothing else does not provide a good control.

Since the entire figure 7 weighs on this, and we have thus far never been able to find a good antibody for FFPE for human RASGRP1, a proper control is necessary to avoid reporting of an artifact here. Many antibodies will give some sort of staining, but that does not give any clue regarding actual specificity.

I suggested in my previous comments, that the authors perform a proper control, and could use the human protein atlas to search for differential RASGRP1 expressing tissues, which for this control require to be human, so that they may compare RASGRP1 negative and positive tissues, and it is best to show of multiple tissues at least several examples of staining.

More preferably, the authors can use human organoids, where RASGRP1 is knocked out versus Wildtype control, of which the KO should show no staining, and the WT should, using the exact same staining protocol as for the tissues in figure 7 (after embedding in FFPE and stained in the same antigen retrieval, blocking, staining, imaging protocol) .

Reviewer #3 (Remarks to the Author):

In the revised version (R2) of their manuscript entitled "RasGRP1 promotes acute inflammatory responses and restricts inflammation-contributed cancer cell growth", Wang and colleagues have addressed all issues raised by the reviewer.

The statistical methods have been revised and seem to be appropriate now.

As for the in vivo effect of RasGRP1-3' UTR (Fig. 4, S11) there are still some concerns that the observed in vivo effects are at least partly due to IFN induction, as the co-localisation data of Cy3-RasGRP1-3'UTR and F4/80 are not fully convincing. The others should therefore include one or two sentences that take-up of RasGRP1-3' UTR in vivo might not be very efficient and that the observed effects might at least in part be due to induction of an IFN response.

Overall this is an intellectually appealing study which has been very well elaborated.

Point-to-point responses to Reviewers (NCOMMS-21-07341B)

Response to Reviewer #1

Most points have now been addressed, apart from one which is a rather essential point that has not been properly answered. This entails the control for the functionality of the used RASGRP1 protein staining by antibody of the patient tissues in Figure 7.

I have pointed the authors towards tools they can use to find proper control tissues, however, to my disappointment, the figure included (Fig S14) has merely 1 image of an IHC staining of mouse intestine. This is most definitely not a suitable control for human tissues, it is a different species, and one image compared to nothing else does not provide a good control.

Since the entire figure 7 weighs on this, and we have thus far never been able to find a good antibody for FFPE for human RASGRP1, a proper control is necessary to avoid reporting of an artifact here.

Many antibodies will give some sort of staining, but that does not give any clue regarding actual specificity.

I suggested in my previous comments, that the authors perform a proper control, and could use the human protein atlas to search for differential RASGRP1 expressing tissues, which for this control require to be human, so that they may compare RASGRP1 negative and positive tissues, and it is best to show of multiple tissues at least several examples of staining.

More preferably, the authors can use human organoids, where RASGRP1 is knocked out versus Wildtype control, of which the KO should show no staining, and the WT should, using the exact same staining protocol as for the tissues in figure 7 (after embedding in FFPE and stained in the

same antigen retrieval, blocking, staining, imaging protocol) .

Response: We thank the reviewer's good suggestion and apologies for the unresolved concern at last time. According to reviewer's suggestion, we analyzed human protein atlas and human colon cancer single cell atlas and found that RasGRP1 has a low expression in gastric tissue and has a high expression in T cells, B cells, NK cells and innate lymphoid cells of colon tissue. So, we performed new experiments to detect the specificity of anti-RasGRP1 antibody in human colon and gastric tissue by immunohistochemistry staining using the same staining protocol as for the tissues in fig.7 (**new Fig S14**).

Culturing human organoids is a very good suggestion but it has a bit difficult for our Lab to perform organoid experiments. We knocked-out RasGRP1 in HepG2 cells and collected these cells via centrifugation for analyzing the specificity of anti-RasGRP1 antibody by IHC staining. Because the protein expression level of RasGRP1 is very low in HepG2 cells, we could not detect RasGRP1 in both wild-type HepG2 cells and RasGRP1-KO HepG2 cells until we over-expressed RasGRP1 in RasGRP1-KO HepG2 cells (**Fig 0, see below, not show in revised manuscript**).

We hope these experiments have addressed your concerns. Thank you for your kind consideration. If any questions exist, inform us and we will discuss and perform required experiments.

Fig. 0 Anti-RasGRP1 antibody immunohistochemistry (IHC) staining of HepG2 cells with wild-type RasGRP1 (left), knocking-out RasGRP1 (middle) and over-expressing RasGRP1 (right).

[REDACTED]

[REDACTED]

Response to Reviewer #3:

In the revised version (R2) of their manuscript entitled "RasGRP1 promotes acute inflammatory responses and restricts inflammation-contributed cancer cell growth", Wang and colleagues have addressed all issues raised by the reviewer.

The statistical methods have been revised and seem to be appropriate now.

As for the in vivo effect of RasGRP1-3' UTR (Fig. 4, S11) there are still some concerns that the observed in vivo effects are at least partly due to IFN induction, as the co-localisation data of Cy3-RasGRP1-3'UTR and F4/80 are not fully convincing. The others should therefore include one or two sentences that take-up of RasGRP1-3' UTR in vivo might not be very efficient and that the observed effects might at least in part be due to induction of an IFN response.

Overall this is an intellectually appealing study which has been very well elaborated.

Response: We thank the reviewer's good suggestion and we have described the limitation of the in vivo effects of RasGRP1-3'UTR in the revised MS (**page 13, line 272-274**).

REVIEWER COMMENTS

Reviewer #1 (Remarks to the Author):

I appreciate the included controls, it will greatly support your data. I would like to congratulate all authors with their manuscript, and have no further comments.

Point-to-point responses to Reviewers (NCOMMS-21-07341C)

Response to Reviewer #1

I appreciate the included controls, it will greatly support your data. I would like to congratulate all authors with their manuscript and have no further comments.

Response: Many thanks for all good suggestions. Special thanks to reviewer for your valuable time and expertise to review our manuscript.